# RUFY1 binds Arl8b and mediates endosome-to-TGN CI-M6PR retrieval for cargo sorting to lysosomes

Shalini Rawat[1], Dhruba Chatterjee[1], Rituraj Marwaha[1], Gitanjali Charak[1], Gaurav Kumar[2], Shrestha Shaw[1], Divya Khatter[1], Sheetal Sharma[2], Cecilia de Heus[3], Nalan Liv[3], Judith Klumperman[3], Amit Tuli[2], and Mahak Sharma[1]

**Arl8b, an Arf-like GTP-binding protein, regulates cargo trafficking and positioning of lysosomes. However, it is unknown whether Arl8b regulates lysosomal cargo sorting. Here, we report that Arl8b binds to the Rab4 and Rab14 interaction partner, RUN and FYVE domain-containing protein (RUFY) 1, a known regulator of cargo sorting from recycling endosomes. Arl8b determines RUFY1 endosomal localization through regulating its interaction with Rab14. RUFY1 depletion led to a delay in CI-M6PR retrieval from endosomes to the TGN, resulting in impaired delivery of newly synthesized hydrolases to lysosomes. We identified the dynein-dynactin complex as an RUFY1 interaction partner, and similar to a subset of activating dynein adaptors, the coiled-coil region of RUFY1 was required for interaction with dynein and the ability to mediate dynein-dependent organelle clustering. Our findings suggest that Arl8b and RUFY1 play a novel role on recycling endosomes, from where this machinery regulates endosomes to TGN retrieval of CI-M6PR and, consequently, lysosomal cargo sorting.**

## Introduction

The endolysosomal system is a dynamic network of membrane-bound compartments that includes early endosomes, recycling endosomes, late endosomes, and lysosomes. Proteins and other cellular cargo internalized at the cell surface or from intracellular locations such as the TGN traffic through these compartments on their way to their functional location, as well as to be degraded in the endolysosomal compartments (Huotari and Helenius, 2011; Saftig and Klumperman, 2009). The small GTP-binding (G) proteins of the Ras superfamily—Rabs, Arfs, and Arf-like (Arl) GTPases—regulate vesicular transport by orchestrating the recruitment of their effectors on specific organelles/endosomes, which then mediate the subsequent steps of vesicle budding, motility, tethering, and finally, fusion with the target compartment. Specific guanine exchange factors (GEFs) regulate the transition of small G proteins from their cytosolic GDP-bound form to an active, membrane-localized, GTP-bound form, while specific GTPase activating proteins (GAPs) catalyze the hydrolysis of GTP to GDP for the inactivation of small G proteins (Hutagalung and Novick, 2011).

Arl8a and Arl8b are members of the Arl subfamily that localize to late endosomes/endolysosomes/lysosomes (hereafter referred to as "lysosomes") and regulate microtubule-based lysosome motility and fusion with other membrane-bound compartments such as late endosomes and phagosomes (Hofmann and Munro, 2006; Khatter et al., 2015b). The multisubunit BLOC-1-related complex (BORC) mediates the recruitment of Arl8 paralogs to lysosomes (Pu et al., 2015). Arl8b recruits its effectors PLEKHM2/SKIP, which binds to the kinesin-1 motor protein and drives anterograde lysosomal motility, and PLEKHM1 and HOPS complex, which mediate tethering and fusion of late endosomes/autophagosomes with lysosomes (Garg et al., 2011; Khatter et al., 2015a; Marwaha et al., 2017; Rosa-Ferreira and Munro, 2011). PLEKHM1 and PLEKHM2, previously identified Arl8b effectors, are RUN domain-containing proteins that interact with Arl8b through their RUN domains (Marwaha et al., 2017; Rosa-Ferreira and Munro, 2011). The RUN domain (named after the proteins RPIP8, UNC-14, and NESCA) is present in proteins that interact with small G proteins and motor proteins. These proteins regulate processes like vesicular transport and fusion, cell migration, signaling, etc (Callebaut et al., 2001; Yoshida et al., 2011).

To find new Arl8b interaction partners, we searched the literature for RUN domain-containing proteins that have similar localization and/or functional phenotypes to Arl8b. In this context, we found a previous study that described Rabip4′, a RUN domain-containing protein that interacts with the adaptor

[1]Department of Biological Sciences, Indian Institute of Science Education and Research Mohali (IISERM), Punjab, India; [2]Division of Cell Biology and Immunology, CSIR-Institute of Microbial Technology (IMTECH), Chandigarh, India; [3]Section Cell Biology, Center for Molecular Medicine, University Medical Center Utrecht, Utrecht, Netherlands.

Correspondence to Mahak Sharma: msharma@iisermohali.ac.in

R. Marwaha's present address is Tata Institute of Fundamental Research (TIFR), Hyderabad, India.

protein complex AP-3 and regulates lysosomal spatial distribution (Ivan et al., 2012). Rabip4′ and Rabip4 are, respectively, the longer and shorter isoforms encoded by the *rufy1* gene (Fouraux et al., 2004). RUFY1 is a member of the RUFY (RUN and FYVE domain-containing proteins) protein family, which has four members in mammals: RUFY1, RUFY2, RUFY3, and RUFY4 (Char and Pierre, 2020; Kitagishi and Matsuda, 2013). Notably, two recent studies have shown that Arl8b interacts with and regulates the lysosomal localization of RUFY3 and RUFY4 (Keren-Kaplan et al., 2022; Kumar et al., 2022). Surprisingly, the Arl8b-binding site in RUFY3 is the C-terminal coiled-coil region rather than the RUN domain (Keren-Kaplan et al., 2022; Kumar et al., 2022).

RUFY1 (referring here to both isoforms) interacts with multiple early endosomal Rabs, including Rab4, Rab5, and Rab14 (Fouraux et al., 2004; Vukmirica et al., 2006). Despite this, only Rab14 was shown to be essential for RUFY1 localization to early/sorting endosomes (Yamamoto et al., 2010). Previous research has shown that RUFY1 is involved in receptor recycling and cargo sorting from early endosomes. For example, RUFY1 regulates transferrin receptor and integrin recycling from early endosomes (Cormont et al., 2001; Fouraux et al., 2004; Vukmirica et al., 2006; Yamamoto et al., 2010). RUFY1 association with early endosomes was increased following ligand-mediated activation of the epidermal growth factor receptor (EGFR), and RUFY1 knockdown resulted in prolonged retention of EGFR in early endosomal compartments, implying that it may regulate EGFR sorting from early endosomes to late endosomes/lysosomes (Gosney et al., 2018). A recent study has demonstrated that RUFY1, along with Rabenosyn-5, Rab4, and AP-3, regulates early endosomal sorting of melanosomal cargo towards the maturing melanosomes (a type of lysosome-related organelle; Nag et al., 2018).

Here, we report that Arl8b interacts with RUFY1 on a subset of endosomes that resemble early/recycling but not late endosomes. The cytosolic distribution of Arl8b-binding-defective mutants of RUFY1 and of endogenous RUFY1 in Arl8b-depleted cells revealed that binding to Arl8b was required for RUFY1 endosomal localization. Colocalization and knockdown studies showed that RUFY1 regulates endosomes to TGN retrieval of cation-independent mannose-6-phosphate (M6P) receptor (CI-M6PR). As a result, pro-cathepsins transport to late endosomes was delayed in RUFY1 depletion. The mass spectrometric-based identification of the RUFY1 interactome revealed the dynein-dynactin complex as a relevant hit. The binding of RUFY1 to dynein was required to restore CI-M6PR distribution in RUFY1-depleted cells. These results suggest that RUFY1 regulates CI-M6PR sorting to TGN and, consequently, facilitates the delivery of M6P-tagged hydrolases from TGN to lysosomes.

## Results

### RUFY1 isoforms interact with Arl8b and localize on non-acidic compartments marked by Rab14 and EEA1

Rabip4 and Rabip4′ are shorter and longer RUFY1 isoforms that differ in the first 108 amino acids at the N-terminus (Fig. 1 A). We discovered that the longer RUFY1 isoform (molecular weight

~80 kD) is more abundant in HeLa cells than the shorter isoform (molecular weight ~70 kD), whereas HEK293T cells showed nearly equal protein expression of both isoforms (Fig. 1 B). The disappearance of the 80 and 70 kD band signals in cell lysates treated with siRNA that targets both isoforms confirmed the RUFY1 antibody's specificity (Fig. 1 B). Next, we used recombinantly purified GST or GST-tagged Arl8b loaded with either GTP or GDP as bait to pull down FLAG-tagged longer and shorter RUFY1 isoforms from transfected HEK293T cell lysates to investigate the interaction of both isoforms with Arl8b. As shown in Fig. 1 C, Arl8b interacted with both RUFY1 isoforms in the presence of GTP but not GDP, implying that RUFY1 binds with the active or membrane-bound form of Arl8b. In line with the GST pulldown experiments, we found that RUFY1 (longer isoform) co-immunoprecipitated with Arl8b WT (wild-type) and Arl8b Q75L (putative GTP-bound form) point mutants, but not with Arl8b T34N (putative GDP-bound form) point mutants (Fig. 1 D). Because previous assays used RUFY1 and Arl8b overexpression, we wanted to verify whether RUFY1 interacts with Arl8b at physiological expression levels. RUFY1 was co-immunoprecipitated with Arl8b from HeLa cell lysates under endogenous conditions but not with Rab8 used as a negative control (Fig. 1 E and Fig. S1 A). As a positive control, we also probed the IP eluates for PLEKHM1, a known interaction partner for Arl8b, which was immunoprecipitated as expected (Fig. 1 E).

A recent study has shown that, of the four mammalian RUFY proteins, only RUFY3 and RUFY4 interact with Arl8b (Keren-Kaplan et al., 2022). We compared RUFY1 and RUFY3 binding to Arl8b to determine whether differential binding could explain the absence of RUFY1 detection in this previous study (Keren-Kaplan et al., 2022). Densitometric quantification of immunoblots from GST pulldown assays showed that pulldown of RUFY3 with GST-Arl8b (GTP-loaded) was ~twofold more than with RUFY1, indicating that a greater proportion of RUFY3 was bound to Arl8b than that of RUFY1 (Fig. S1, B and C). In agreement with our results that both RUFY1 and RUFY3 interact with Arl8b, a recent study has identified both proteins as significant hits in the proximal interaction network of Arl8a and Arl8b (Li et al., 2022).

As described in a previous study (Yamamoto et al., 2010), we also found that RUFY1 localizes to a subset of Rab14-positive endosomes (Fig. S1 D). Indeed, a modest colocalization of RUFY1 endosomes was also observed with early endosomal proteins EEA1 and SNX1 (Fig. S1, E and F). We did not observe colocalization of endogenous RUFY1 with LAMP1 or with Lysotracker dye; the latter accumulates and marks the acidic compartments in the cells (Fig. S1, G and H). Notably, RUFY1 endosomes were also positive for the CI-M6PR, which traffics from the TGN to early endosomes en route to late endosomes before recycling back to the TGN (Fig. S1 I). RUFY1 was also colocalized with a subset of endosomes positive for Vps26, a subunit of the retromer complex that regulates cargo recycling from endosomes to the TGN (Fig. S1 J; Gallon and Cullen, 2015). In summary, RUFY1 localizes to endosomes that include early endosomal proteins, with Rab14 showing the most apparent colocalization (see Pearson Correlation Coefficient [PCC] and Mander's Overlap Coefficient [MOC] in Fig. S1, K and L, respectively).

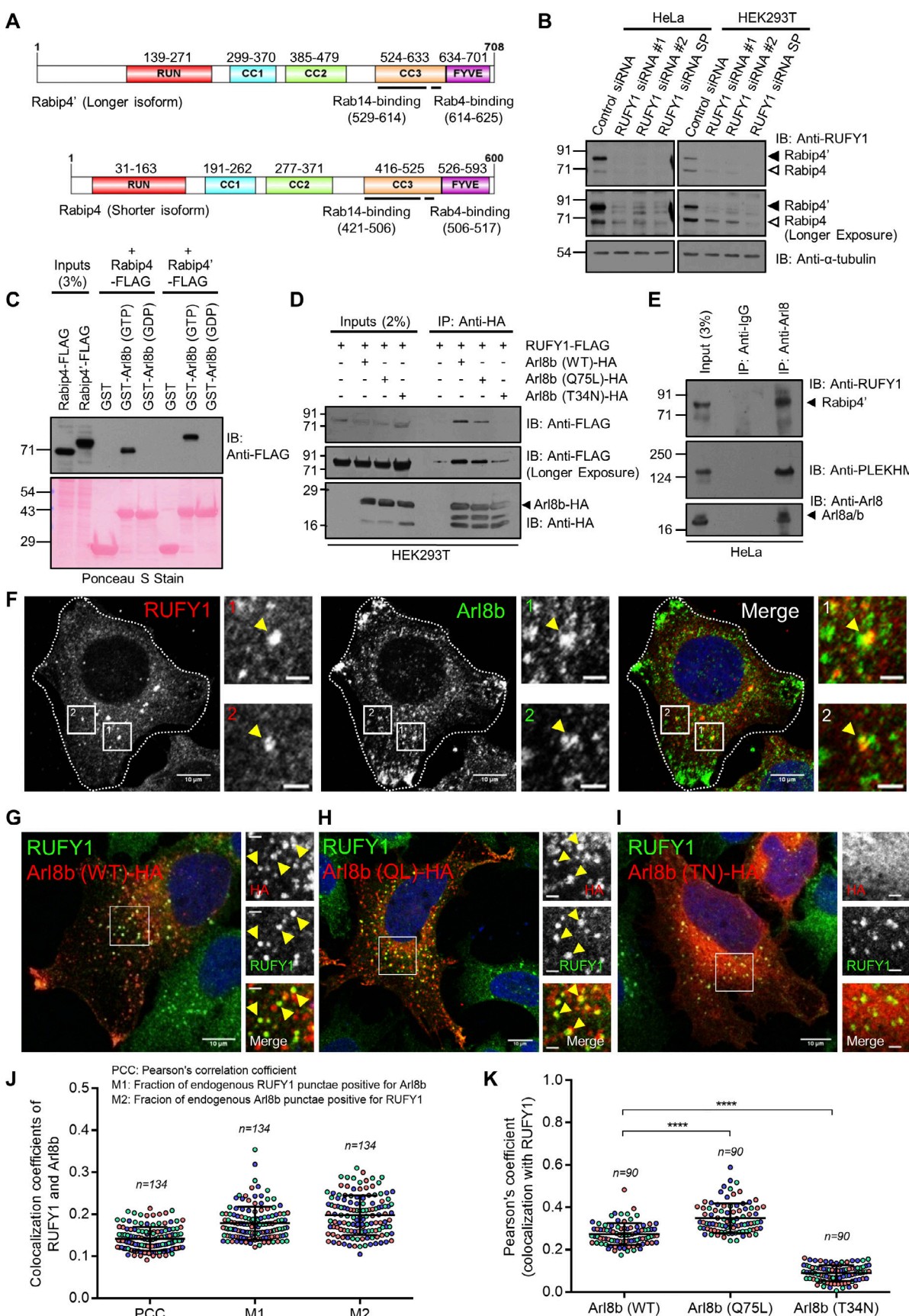

Figure 1. **RUFY1 isoforms interact with the GTP-bound form of Arl8b. (A)** Schematic representation of the domain architecture of RUFY1 isoforms. **(B)** Lysates of HeLa and HEK293T cells treated with indicated siRNA (control siRNA, RUFY1 siRNA #1, #2, and SP [SMARTpool]) were immunoblotted (IB) with

anti-RUFY1 antibody for assessing the specificity of the antibody and with anti-α-tubulin antibody as a loading control. Arrowheads indicate the two isoforms (Rabip4 and Rabip4′). **(C)** Recombinant GST and GST-Arl8b proteins were immobilized on glutathione-coated-agarose beads and loaded with either GTP or GDP and then incubated with HEK293T cell lysates expressing Rabip4′-FLAG (longer isoform) or Rabip4-FLAG (shorter isoform). The precipitates were IB with anti-FLAG antibody, and Ponceau S staining was done to visualize the purified proteins. **(D)** RUFY1-FLAG (longer isoform) was co-transfected with vector or with different forms of Arl8b-HA into HEK293T cells, and the lysates were immunoprecipitated with anti-HA antibodies-conjugated-agarose beads. The precipitates were IB with the indicated antibodies. **(E)** HeLa cell lysates were immunoprecipitated with anti-Arl8a/b antibodies-conjugated-agarose beads. The precipitates were IB with the indicated antibodies. **(F)** Representative confocal micrograph of HeLa cells immunostained for both endogenous RUFY1 and Arl8b. **(G–I)** Representative confocal micrographs of HeLa cells transfected with Arl8b (WT)-HA, Arl8b (Q75L)-HA, and Arl8b (T34N)-HA, followed by immunostaining with anti-HA and anti-RUFY1 antibodies. For F–I, arrowheads in the insets mark the colocalized pixels. Bars: (main) 10 µm; (insets) 2 µm. **(J)** Pearson's and Mander's colocalization coefficient quantification of endogenous RUFY1 with endogenous Arl8b. **(K)** Pearson's correlation coefficient quantification of endogenous RUFY1 with different forms of transfected Arl8b-HA. For graphs (J and K), the values plotted are the mean ± SD from three independent experiments. Experiments are color-coded, and each dot represents the individual data points from each experiment. The total number of cells analyzed is indicated on the top of each data set (****P < 0.0001; unpaired two-tailed *t* test). Source data are available for this figure: SourceData F1.

Since Arl8b is predominantly localized to lysosomes (marked by LAMP1), we questioned whether or not RUFY1 and Arl8b would colocalize. Indeed, under endogenous conditions and upon transfection of an epitope-tagged Arl8b (WT) construct, we observed that RUFY1 colocalized with only a subset of juxtanuclear punctae of Arl8b and not with the characteristic peripheral pool of Arl8b that generally marks the lysosomes (Fig. 1, F and G; quantification is shown in Fig. 1 J [for both endogenous proteins] and Fig. 1 K [Arl8b [WT]-HA with endogenous RUFY1]). Consistent with the evidence that RUFY1 preferentially binds to Arl8b in its GTP-bound state, colocalization between RUFY1 and Arl8b was increased in cells expressing a constitutively GTP-bound Arl8b mutant (Q75L), compared to cells expressing Arl8b in its WT form (Fig. 1, H and K). In contrast, colocalization was not observed with the cytosolic constitutively GDP-bound Arl8b mutant (T34N; Fig. 1, I and K).

To examine the characteristics of the compartment where RUFY1 colocalizes with Arl8b, we co-transfected HeLa cells with RUFY1 (longer isoform) and Arl8b and processed these cells for immuno-electron microscopy. Notably, colocalization of RUFY1 and Arl8b was identified on vesicles juxtaposed to compartments that morphologically resembled early endosomes and multivesicular bodies (MVB)/late endosomes (Fig. 2 A). To identify these vesicles, we immunostained cells co-expressing tagged Arl8b and RUFY1 constructs with a variety of markers. We noted that Arl8b and RUFY1 endosomes were mostly Rab14-positive (Fig. 2 B, quantification shown in Fig. 2, F and G). Partial colocalization was also observed with EEA1 and CI-M6PR, but late endosomal/lysosomal markers Rab7 and Lysotracker dye showed substantially less overlap (Fig. 2, C–G and Fig. S2 A). We also examined the colocalization of RUFY1 and Arl8b and the compartment-specific marker using structured illumination microscopy (SIM), which can resolve objects separated by 100–150 nm. Here, we observed two populations of Arl8b, one of which was positive for the late endosomal marker Rab7 but negative for RUFY1 (Fig. S2 B). A second Arl8b population, in contrast, contained RUFY1 and was positive for Rab14 (Fig. S2 C), EEA1 (Fig. S2 D), and CI-M6PR (Fig. S2 E). These findings suggest that, in addition to lysosomes, Arl8b localizes to non-acidic endosomes and binds to the Rab14 effector RUFY1 on these endosomes.

**Arl8b directly binds to the RUN domain of RUFY1**
Previous research has shown that the N-terminal RUN domain of PLEKHM1 and PLEKHM2 is necessary for Arl8b binding

(Marwaha et al., 2017; Rosa-Ferreira and Munro, 2011). We observed that the RUFY1 RUN domain contains a set of conserved arginine residues (R206 and R208, located within the RxRAWL motif [Fig. 3 A]), which have been demonstrated to be required for the interaction of PLEKHM1 and PLEKHM2 RUN domain with Arl8b (Marwaha et al., 2017). To determine whether the RUN domain is essential for Arl8b binding, we created a deletion mutant of RUFY1 (longer isoform) without the RUN domain-containing region (272–708 a.a.; RUFY1 [ΔRUN]) and examined its interaction with Arl8b using a GST pulldown assay. As bait, GST-Arl8b was able to pulldown WT but not RUFY1 (ΔRUN) from transfected HEK293T cell lysates (Fig. 3, B and C). Next, using site-directed mutagenesis, the R206 and R208 residues of RUFY1 (the longer isoform) were mutated to alanine, and interaction with Arl8b was evaluated. Compared to WT, the RUFY1 (R206/R208A; RR→A) mutant exhibited drastically reduced Arl8b binding (Fig. 3, B and C). To investigate if the RUFY1 RUN domain was sufficient for interaction with Arl8b, a purified protein-protein interaction assay was performed using GST-RUFY1 RUN domain (1–302 a.a.) as bait to pulldown His-tagged Arl8b (WT), Arl8b (Q75L) and Arl8b (T34N) purified proteins. As depicted in Fig. 3, D and E, the RUFY1 RUN-domain containing fragment exhibited preferential binding with the WT and Q75L forms of Arl8b as compared to the T34N form.

Next, in order to comprehend the relevance of Arl8b binding, we investigated the localization of ΔRUN and R206/R208A (RR→A) mutants of RUFY1 that were defective in binding to Arl8b. In contrast to RUFY1 WT, both ΔRUN and R206/R208A (RR→A) RUFY1 mutants were predominantly cytosolic and did not colocalize with Arl8b-positive endosomes (Fig. 3, F–H and PCC and MOC quantification shown in Fig. 3, I and J). These findings imply that Arl8b-binding to the RUN domain may be necessary for RUFY1 membrane localization.

**Arl8b regulates RUFY1 endosomal localization and promotes RUFY1 and Rab14 interaction**
Next, we depleted Arl8b from HeLa cells to ascertain if its expression is required for RUFY1 membrane localization. To this end, we first confirmed that the knockdown efficiency was >90% using a siRNA-based approach (Fig. S2 F). As with Arl8b-binding-defective mutants, we observed a striking redistribution of RUFY1 from endosomes to the cytosol in Arl8b siRNA-treated cells (compare Fig. 4, A and B; quantification of the number of RUFY1 endosomes is shown in Fig. 4 D). These

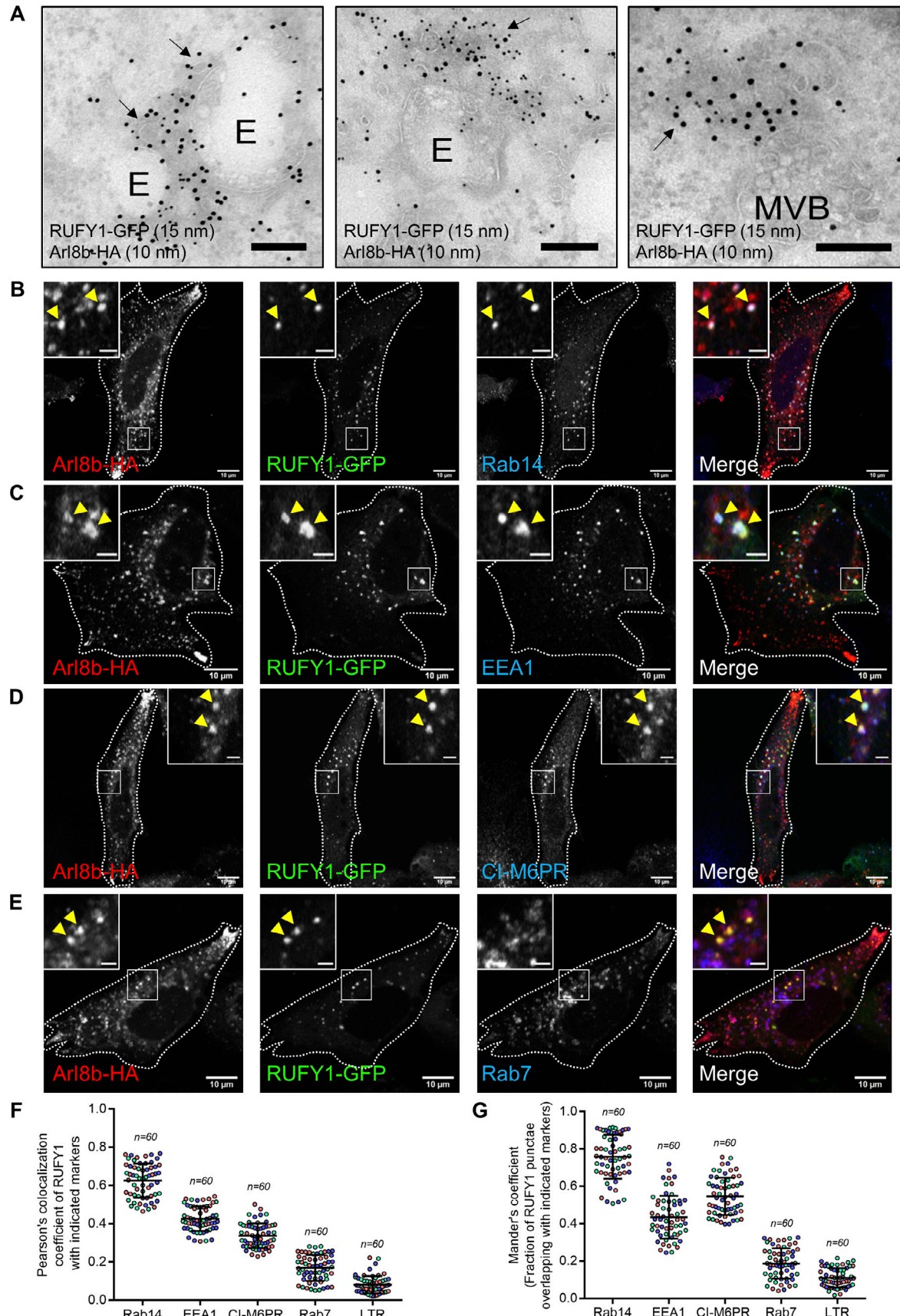

Figure 2. **Arl8b and RUFY1 colocalize on a subset of early endosomes positive for Rab14. (A)** Immuno-electron micrographs of ultrathin cryosections of HeLa cells transfected with Arl8b-HA (immunolabeled with 10 nm gold) and RUFY1-GFP (immunolabeled with 15 nm gold). Colocalization of RUFY1 and Arl8b

was observed on smaller vesicles juxtaposed to endosomes (E, left and middle panels) and late endosomes/MVBs (right panel). Bars: 200 nm. **(B–E)** Representative confocal micrographs of HeLa cells transfected with Arl8b-HA and RUFY1-GFP and stained for various endocytic markers (B) Rab14, (C) EEA1, (D) CI-M6PR, and (E) Rab7. In the insets, the arrowhead marks the colocalized pixels. Bars: (main) 10 μm; (insets) 2 μm. **(F and G)** Pearson's and Mander's colocalization coefficient quantification of RUFY1-GFP and Arl8b-HA-positive compartments with indicated endocytic markers. The values plotted are the mean ± SD from three independent experiments. Experiments are color-coded, and each dot represents the individual data points from each experiment. The total number of cells analyzed is indicated on the top of each data set.

cells were co-stained with EEA1 and Rab14, both of which continued to localize to membranes in cells treated with Arl8b siRNA (Fig. 4, A and B). In cells transfected with a siRNA-resistant Arl8b rescue construct, endosomal localization of RUFY1 was restored, demonstrating that the phenotype was caused by Arl8b depletion and not an off-target effect of the siRNA oligo (Fig. 4, C and D). We noted that few RUFY1 punctae were still present in Arl8b-depleted cells, which colocalized with Rab14 and EEA1 (see inset in Fig. 4, A and B). This suggests that a subset of RUFY1 endosomes is not dependent on Arl8b expression. The Arl8b CRISPR knockout cells also showed a striking, but partial redistribution of RUFY1 to the cytosol (as observed upon Arl8b siRNA treatment), while the early endosomal marker EEA1 continued to localize to endosomes (Fig. S2, F–I). Notably, in Arl8b overexpressing cells, we observed the opposite phenotype, i.e., the size of RUFY1 endosomes and the brightness of individual punctae were increased relative to the untransfected cells (Fig. S2, J and K).

To validate the immunofluorescence findings, we isolated membrane and cytosol fractions of control and Arl8b-depleted cells and analyzed RUFY1 levels in each fraction (Fig. 4 E). Prior research showed that RUFY1 was enriched in the membrane fraction and that this association was maintained even after solubilization with 1% Triton X-100 (Mari et al., 2001). Consistent with this, we also found that RUFY1 was exclusively present in the Triton X-100 insoluble fraction (Fig. 4 E, the band just below ~91 kD marks RUFY1, as shown in Fig. S2 L, is greatly diminished in RUFY1-depleted lysates. The band below ~71 kD marks a non-specific band observed with anti-RUFY1 antibody). GAPDH was used as a positive control for the cytosol fraction. We noted that the RUFY1 signal in Triton X-100 insoluble fractions (normalized to input) was significantly reduced upon Arl8b depletion, indicating that Arl8b determines RUFY1 membrane association (Fig. 4, E and F).

Given the substantial colocalization of RUFY1 to Rab14-positive compartments, it is conceivable that the GTP-bound form of Rab14 recruits RUFY1 to endosomes, and the presence of Arl8b on these endosomes facilitates RUFY1 stable membrane attachment. To distinguish between the roles of Arl8b and Rab14 in mediating RUFY1 endosomal localization, we employed an approach recently described for the in vivo identification of small G protein interaction partners (Gillingham et al., 2019). Herein, we expressed the GTP-locked forms of both Arl8b (Q75L) and Rab14 (Q70L) with a mitochondrial targeting sequence and assessed recruitment of the putative effector, i.e., RUFY1, to mitochondria in the transfected cells. We also investigated the role of Rab4 and Rab5 (two small G proteins shown to interact with RUFY1 (Fouraux et al., 2004) in this experiment by expressing their active versions with a

mitochondrial targeting sequence. We first confirmed that mito-tag versions of these G proteins localized to mitochondria by visualizing their colocalization with mitochondrial marker Tom20 (Fig. S3, A–D). As shown in Fig. S3, E and F, while Mito-Arl8b (Q75L) was able to recruit its known effector SKIP/PLEKHM2 to mitochondria, RUFY1 retained its endosomal distribution and was not relocalized to mitochondria upon expression of Mito-Arl8b (Q75L). Indeed, of the four G proteins investigated in this assay, only Rab14 (Q70L) was able to recruit RUFY1 to mitochondrial membranes (Fig. S3, G–I; PCC quantification shown in Fig. S3 J). In accordance with these findings and as reported in the literature (Yamamoto et al., 2010), RUFY1 was completely cytosolic in Rab14 siRNA-treated cells (Fig. S3, K and L).

Our findings indicate that RUFY1 is not an Arl8b effector, so how does Arl8b depletion impair RUFY1 localization on Rab14-positive recycling endosomes? We noted that even when Rab14 was overexpressed in Arl8b knockdown cells, a significant proportion of RUFY1 remained cytosolic, despite Rab14's endosomal distribution (see Video 1 and quantification from a single frame of the live-cell imaging videos in Fig. S3 M), suggesting a reduced association of RUFY1 with Rab14. This was reflected by a reduced co-immunoprecipitation of RUFY1 and Rab14 in Arl8b-depleted cells (Fig. 4, G and H). We then analyzed Arl8b's role in regulating RUFY1 and Rab14 interaction, which in turn would influence RUFY1 endosomal localization. To test this, the three proteins were recombinantly isolated, and RUFY1 was incubated with GTP-loaded Rab14 in the presence of increasing amounts of His-tagged Arl8b protein. As shown in Fig. 4 I, the pulldown of full-length RUFY1 with Rab14 increased with increasing amounts of Arl8b, although no such increase was observed in the presence of His-tagged Rab7, used as a control. As predicted by the formation of a tripartite Rab14-RUFY1-Arl8b complex, Arl8b was pulled down with Rab14 only in the presence of RUFY1 and no interaction was observed without RUFY1. Rab7 was not pulled down in this RUFY1 and Rab14 complex, demonstrating the experiment's selectivity (Fig. 4 I).

Interestingly, in a yeast-two hybrid assay, we observed that deletion of the RUN domain-containing region (NΔRUN RUFY1) enhanced RUFY1 interaction with Rab14 (Fig. S3 N). Thus, it is plausible that the RUN domain of RUFY1 plays an autoinhibitory role in binding to Rab14 and that interaction with Arl8b relieves this autoinhibition, thereby promoting efficient binding of RUFY1 to Rab14.

## RUFY1 depletion leads to enlarged lysosomes with features suggestive of lysosome dysfunction
Previous studies have shown that RUFY1 regulates cargo sorting and recycling from early/recycling endosomes (Fouraux et al.,

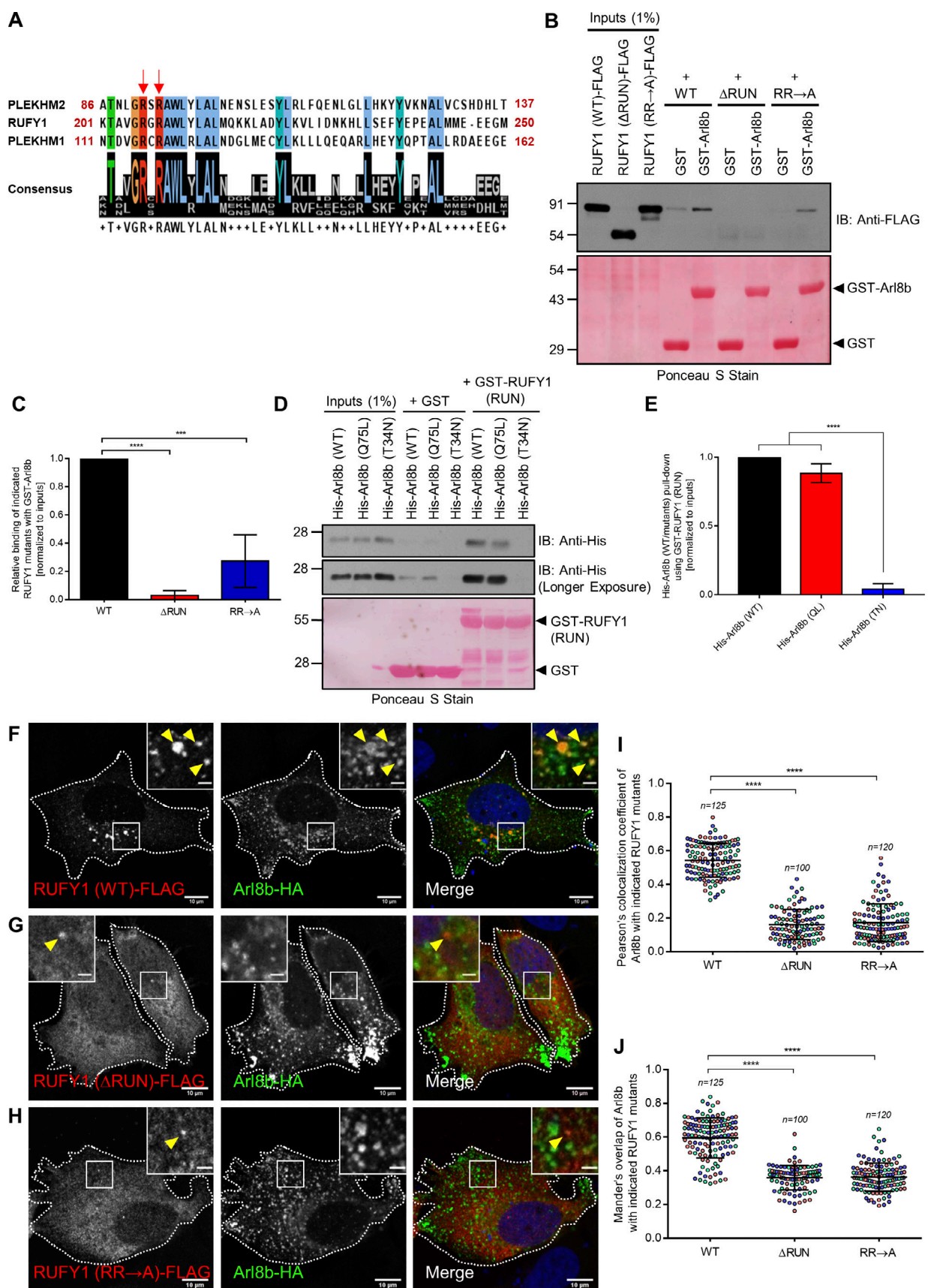

Figure 3. **The RUN domain-containing region of RUFY1 is essential and sufficient for binding to Arl8b. (A)** Schematic showing the Clustal Omega alignment of a short stretch of protein sequence within the RUN domain of PLEKHM2, RUFY1 (longer isoform), and PLEKHM1. The red arrows show conserved

arginine residues (present in the context of the RxRAWL motif) in the RUN domains of all three proteins. In this study, R206 and R208 residues of RUFY1 were mutated to alanine. The color-coding of amino acids is based on their physiochemical properties, as listed on the Clustal Omega homepage. **(B)** HEK293T cell lysates expressing RUFY1 (WT)-FLAG or indicated RUFY1 mutants were incubated with recombinant GST and GST-Arl8b proteins immobilized on glutathione-coated-agarose beads. The precipitates were immunoblotted (IB) with anti-FLAG antibodies, and Ponceau S staining was done to visualize the purified proteins. **(C)** Densitometric analysis of band intensity of GST pulldown normalized to input RUFY1-FLAG signal (WT or mutants). The values plotted are the mean ± SD from four independent experiments (****P < 0.0001; ***P < 0.001; unpaired two-tailed *t* test). **(D)** Recombinant GST and GST-RUFY1 (RUN; 1–302 a.a.) proteins were immobilized on glutathione-coated-agarose beads and incubated with His-Arl8b (WT), His-Arl8b (Q75L), and His-Arl8b (T34N). The precipitates were IB with anti-His antibodies, and Ponceau S staining was done to visualize the purified proteins. Notably, His-Arl8b (Q75L) showed more non-specific binding (as observed in the GST lane) than His-Arl8b (WT). **(E)** Densitometric analysis of band intensity of GST pulldown normalized to input signal (His-Arl8b WT or mutants). The values plotted are the mean ± SD from three independent experiments (****P < 0.0001; unpaired two-tailed *t* test). **(F–H)** Representative confocal micrographs of HeLa cells co-transfected with Arl8b-HA and RUFY1 (WT)-FLAG (F), RUFY1 (ΔRUN)-FLAG (G) or RUFY1 (RR→A)-FLAG (H). The arrowhead marks the colocalized pixels. Bars: (main) 10 µm; (insets) 2 µm. **(I and J)** Pearson's and Mander's colocalization coefficient quantification of Arl8b-HA with indicated RUFY1 mutants. The values plotted are the mean ± SD from three independent experiments. Experiments are color-coded, and each dot represents the individual data points from each experiment. The total number of cells analyzed is indicated on the top of each data set (****P < 0.0001; unpaired two-tailed *t* test). Source data are available for this figure: SourceData F3.

---

2004; Vukmirica et al., 2006; Yamamoto et al., 2010). However, the role of RUFY1 in regulating lysosomal function has not been investigated, although RUFY1 depletion was shown to cause a change in lysosome positioning (Ivan et al., 2012). To determine whether RUFY1 regulates the composition and/or function of late endocytic compartments, we treated cells with a single siRNA (siRNA #1) or a pool of four oligos (SMARTpool; SP) and immunostained them for markers of late endosome/lysosomal compartments. Both SP and the single oligo efficiently reduced expression of the two RUFY1 isoforms (see Fig. 1 B). In RUFY1-depleted cells, we observed a striking enlargement of LAMP1-positive compartments with the appearance of several ring-like or vacuolated LAMP1-positive endosomes (Fig. 5, A–C [see inset], and quantification of lysosome [LAMP1+] area in Fig. 5 E). In cells expressing siRNA-resistant RUFY1 (longer isoform) construct, lysosome area was similar to control, indicating that this phenotype is specifically due to RUFY1 depletion (Fig. 5, D and E). This change in lysosome area was better visualized by LAMP1 immuno-electron microscopy on ultrathin cryosections of RUFY1-depleted cells, where a twofold increase in lysosome area was observed on average when compared to control cells (Fig. 5, F and G).

Lysotracker staining also revealed a ~1.5–2-fold higher signal in RUFY1-depleted cells compared to control siRNA-treated cells, indicating an expansion of acidic compartments (Fig. S4, A–D). Rab7 immunostaining similarly revealed enlarged structures in RUFY1-depleted cells, although the phenotype was less obvious than observed with LAMP1 immunostaining (Fig. S4, E–G). Notably, we did not observe peripheral positioning of lysosomes in RUFY1-depleted cells, as previously described (Ivan et al., 2012). HEK293T cells employed in this prior work have a different RUFY1 isoform expression pattern than HeLa cells (see Fig. 1 B), in addition to the morphological differences between the two cell types, which could also account for this difference. As a control, we detected no significant differences in the size and distribution of sorting endosomes labeled by EEA1 in RUFY1-knockdown cells (Fig. S4, H–J).

To evaluate the pH of acidic compartments in RUFY1-depleted cells, we employed the Lysosensor Yellow/Blue DND-160 dye, which permits ratiometric detection of the intra-organelle pH of acidic organelles (Ma et al., 2017). The average lysosomal pH of RUFY1-depleted cells (6.00 ± 0.17) was higher than that of

control cells in three independent experiments, although the difference was not statistically significant (5.58 ± 0.22, as measured over three independent experiments (Fig. 5 H). This result suggests that lysosomes are probably less degradative upon RUFY1 depletion.

To determine cargo degradation in lysosomes, we treated control and RUFY1-depleted cells with BODIPY-BSA, an endocytic cargo similar to DQ-BSA that fluoresces upon proteolytic cleavage in lysosomes (Marwaha and Sharma, 2017). We found a modest but consistent decrease in BODIPY-BSA signal after RUFY1 knockdown, implying that the enlarged lysosomes observed in RUFY1 likely accumulate cargo substrates due to their lower degradative potential (Fig. S4 K). Next, we analyzed degradation of another lysosomal cargo, EGFR, which is endocytosed and transported to lysosomes upon EGF stimulation. RUFY1-depleted cells showed a significant delay in the loss of EGFR fluorescence signal intensity at different time points of the chase after EGF-stimulation, confirming RUFY1's role in regulating lysosomal cargo degradation (Fig. 5 I).

Previous research has demonstrated that lysosomal enzymes such as cathepsins are elevated in response to lysosome dysfunction, possibly as a compensatory mechanism to degrade cellular cargo (Feng et al., 2020; Napolitano and Ballabio, 2016; Stoka et al., 2016; Werner et al., 2020). To investigate this, we measured the cumulative immunofluorescence intensity of cathepsin D in control and RUFY1 knockdown cells. As demonstrated in Fig. 5, J–O, RUFY1 depletion caused a twofold increase in cathepsin D levels, which were restored in siRNA-resistant RUFY1 transfected cells, showing the specificity of siRNA treatment. Using immunoblotting (Fig. S4 L), anti-cathepsin D antibodies detected an increase in pro-cathepsin D and mature cathepsin D levels in RUFY1-depleted cells. Additionally, there was a slight increase in LAMP1 levels in RUFY1-depleted cells (Fig. S4 L). Taken together, depletion of RUFY1 results in an increase in the size and pH of lysosomes, an increase in cathepsin levels, and impairment in cargo breakdown. Several of these characteristics are regarded as hallmarks of lysosome dysfunction in aging, neurodegenerative, and lysosomal storage diseases (Bonam et al., 2019; Stoka et al., 2016). Consequently, these findings suggest that RUFY1 plays a novel role in maintaining the normal degradative activity of lysosomes.

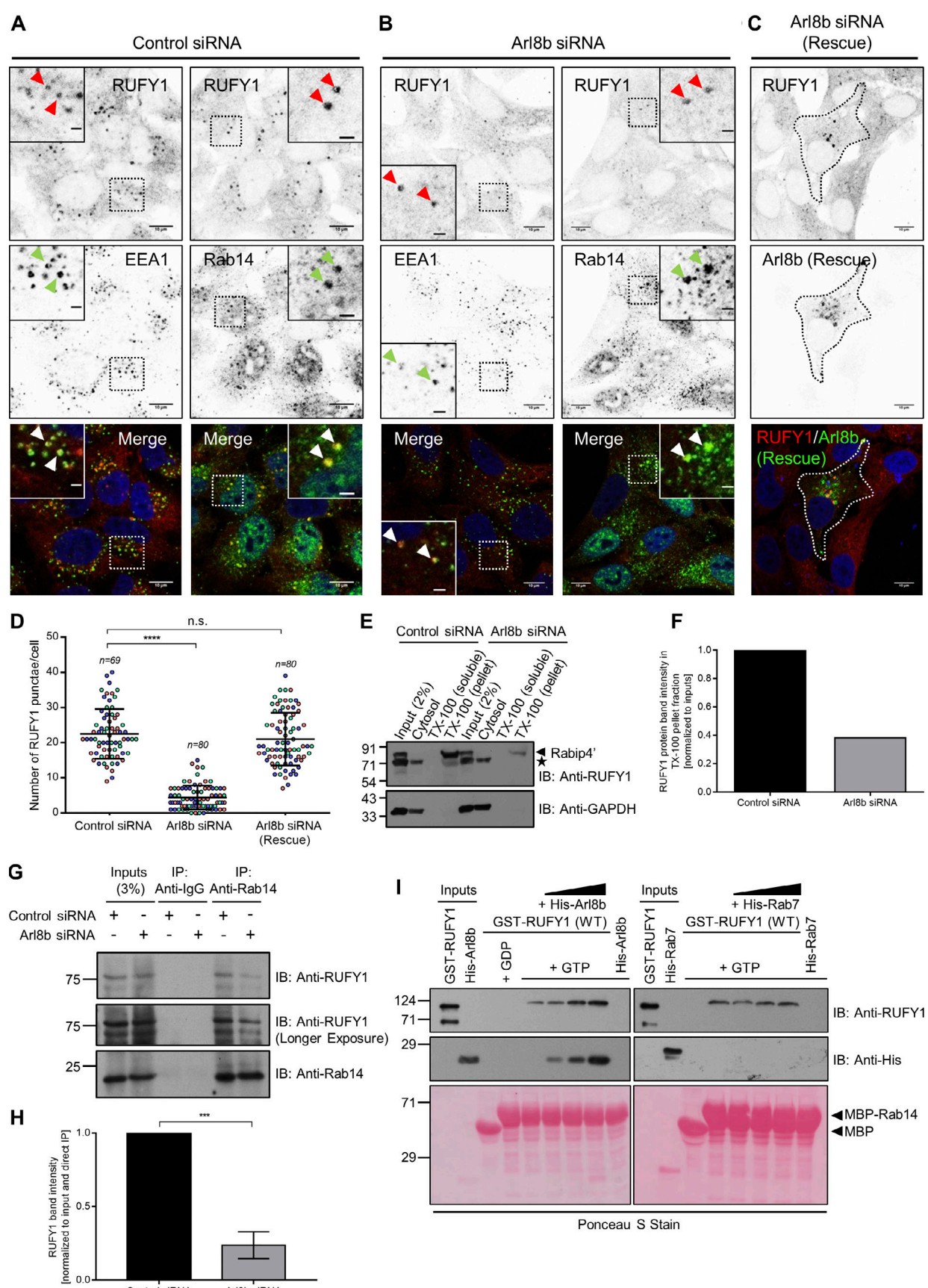

Figure 4. **Arl8b regulates RUFY1 endosomal localization and promotes the interaction of RUFY1 and Rab14. (A and B)** Representative confocal micrographs of HeLa cells treated with the indicated siRNA, followed by immunostaining for endogenous proteins (as labeled). **(C)** Representative confocal

micrograph of HeLa cells treated with Arl8b siRNA and transfected with the untagged-Arl8b (Rescue) construct followed by immunostaining for RUFY1 and Arl8b. Single-channel images of RUFY1, EEA1, and Rab14 are shown as inverted images. Non-specific nuclear staining was observed with anti-Rab14 antibodies. Arrowheads (red for the RUFY1 channel and green for the EEA1/Rab14 channel) mark the colocalized pixels. Bars: (main) 10 µm; (insets) 2 µm. **(D)** Quantification of the number of RUFY1 punctae in HeLa cells upon different siRNA treatments as indicated. The values plotted are the mean ± SD from three independent experiments. Experiments are color-coded, and each dot represents the individual data points from each experiment. The total number of cells analyzed is indicated on the top of each data set (****P < 0.0001; n.s., not significant; unpaired two-tailed t test). **(E)** HeLa cells treated with either control or Arl8b siRNA were homogenized and subjected to ultracentrifugation to separate membrane and cytosol fractions. The supernatant is referred to as cytosol, and the pellet fraction was further treated with 1% Triton X (TX)-100 followed by ultracentrifugation to separate TX-100-insoluble membranes obtained as pellets and supernatant as TX-100-soluble fractions. Cytosol, TX-100 soluble and insoluble pellets were separated by SDS-PAGE followed by immunoblotting with indicated antibodies. Note: "*" marks the non-specific band observed at ~71 kD upon immunoblotting with anti-RUFY1 antibody. The detection of a non-specific band at ~71 kD by this antibody was further confirmed by RUFY1 siRNA as shown in Fig. S2 L. **(F)** Densitometric analysis of RUFY1 band signal in TX-100 insoluble pellet normalized to the input signal. The values plotted are the averages from two independent experiments. **(G)** Lysates of HeLa cells treated with indicated siRNA were immunoprecipitated with anti-Rab14 antibodies, and the precipitates were IB with the indicated antibodies. **(H)** Densitometric analysis of RUFY1 band intensity normalized to input and to direct IP of Rab14. The values plotted are the mean ± SD from three independent experiments (***P < 0.001; unpaired two-tailed t test). **(I)** Recombinant GST-RUFY1 (WT) protein was incubated with MBP alone or GDP/GTP-loaded MBP-Rab14, immobilized on amylose resin, in the presence of increasing amounts of His-Arl8b (WT) or His-Rab7 (WT). The precipitates were IB with the indicated antibodies, and Ponceau S staining was done to visualize the purified proteins. Source data are available for this figure: SourceData F4.

## RUFY1 regulates retrieval of CI-M6PR from endosomes to the TGN

Following the observation that a subpopulation of CI-M6PR colocalizes with RUFY1/Rab14 (Fig. S1 I) and that lysosome dysfunction is observed upon impairment in the CI-M6PR trafficking pathway (Allison et al., 2017; Cui et al., 2019), we evaluated CI-M6PR localization upon RUFY1 depletion. CI-M6PR facilitates the transport of mannose-6-phosphate (M6P)-tagged soluble hydrolases from the TGN to late endosomes. The acidic pH of late endosomes leads to cargo-CI-M6PR complex dissociation, and the receptor (CI-M6PR) is sorted back to the TGN from endosomes (Saftig and Klumperman, 2009).

We measured CI-M6PR colocalization with Golgi and endosomal markers in control and RUFY1-depleted cells to investigate steady-state CI-M6PR distribution (Fig. 6, A–D). In control siRNA-treated cells, CI-M6PR was enriched in the perinuclear region and colocalized with the Golgi marker, Giantin, as well as a population of peripheral CI-M6PR punctate structures colocalized with the endosomal marker SNX1 (Fig. 6, A, C, F, and G). RUFY1 knockdown resulted in a more endosomal and less perinuclear distribution of CI-M6PR, as well as an increase in the size of CI-M6PR punctae (see also CI-M6PR intensity profile distribution in Fig. 6 E). As predicted by an altered CI-M6PR distribution, colocalization of CI-M6PR with Giantin was significantly reduced, whereas it was markedly increased with SNX1 and Rab14 in RUFY1-depleted cells (Fig. 6, B, D, F, and G; and Fig. S4, M–O). By generating a siRNA-resistant rescue construct of RUFY1 (longer isoform), the normal perinuclear distribution of CI-M6PR was restored, indicating that RUFY1 is essential for CI-M6PR Golgi localization (Fig. 6, H–J). We observed no significant change in steady-state CI-M6PR colocalization with retromer subunit Vps35 between control and RUFY1-knockdown cells (Fig. S4, P–R). These results indicate that RUFY1 facilitates sorting of CI-M6PR from a Rab14/SNX1 compartment to the TGN.

Next, we evaluated whether the change in steady-state CI-M6PR distribution observed in RUFY1-depleted cells is a result of a block or delay in CI-M6PR trafficking from endosomes to the TGN. To this end, we utilized the anti-CD8α-mediated internalization assay (Seaman, 2004; Shi et al., 2018), wherein the CD8α-CI-M6PR reporter construct (chimera wherein the cytoplasmic tail of CD8α is replaced with the cytoplasmic tail of CI-M6PR) was transfected into control and RUFY1 siRNA-treated cells, followed by incubation with antibodies against CD8α to label the surface CD8α-CI-M6PR population and chase for various time points (Fig. 6 K). At 5 min, the CD8α-CI-M6PR population was localized to EEA1-positive compartments in both control and RUFY1-depleted cells, with little or no colocalization with Giantin at this time point (Fig. 6 L). At 60 min of chase in control cells, the bulk of CD8α-CI-M6PR cargo vesicles had sorted from early endosomes and localized to the perinuclear area together with Giantin (Fig. 6, M, N, and Q). In contrast, upon RUFY1 depletion, CD8α-CI-M6PR colocalization with Golgi was greatly reduced, and numerous CD8α-CI-M6PR remained localized to early endosomes (Fig. 6, O–Q). Thus, our data imply that RUFY1 is required for optimal CI-M6PR exit from SNX1- and Rab14-positive endosomes to the TGN.

## RUFY1 depletion impairs sorting of lysosomal hydrolases to late endosomes and lysosomes

To evaluate the effects of the delayed retrieval of CI-M6PR from endosomes to TGN upon RUFY1 depletion, we used the RUSH (Retention Using Selective Hooks) method to examine the trafficking of CI-M6PR cargo, cathepsin Z, in control and RUFY1-depleted cells (Fig. 7 A; Boncompain et al., 2012; Niu et al., 2019). Before the addition of biotin, cathepsin Z linked to a streptavidin-binding peptide displayed ER localization because it was maintained there by the luminal streptavidin fused to the ER retention signal KDEL (Hook; Fig. 7 B). A cathepsin Z signal was detected in Golgi and vesicles originating from Golgi in control cells 40 min after the addition of biotin (Fig. 7 C). We detected little or no colocalization with LAMP1-GFP at this time period (Fig. 7, C and J). Approximately 180 min after biotin addition and till 320 min, the majority of the cathepsin Z signal was confined to the LAMP1-GFP compartment (Fig. 7, D, E, and J). In RUFY1-depleted cells, cathepsin Z trafficking to the Golgi appeared comparable to control (Fig. 7, F and G); however, there was a noticeable delay in trafficking from the Golgi to lysosomes, as indicated by the decreased colocalization of cathepsin Z with LAMP1-GFP 180 min after biotin addition (Fig. 7, H and J).

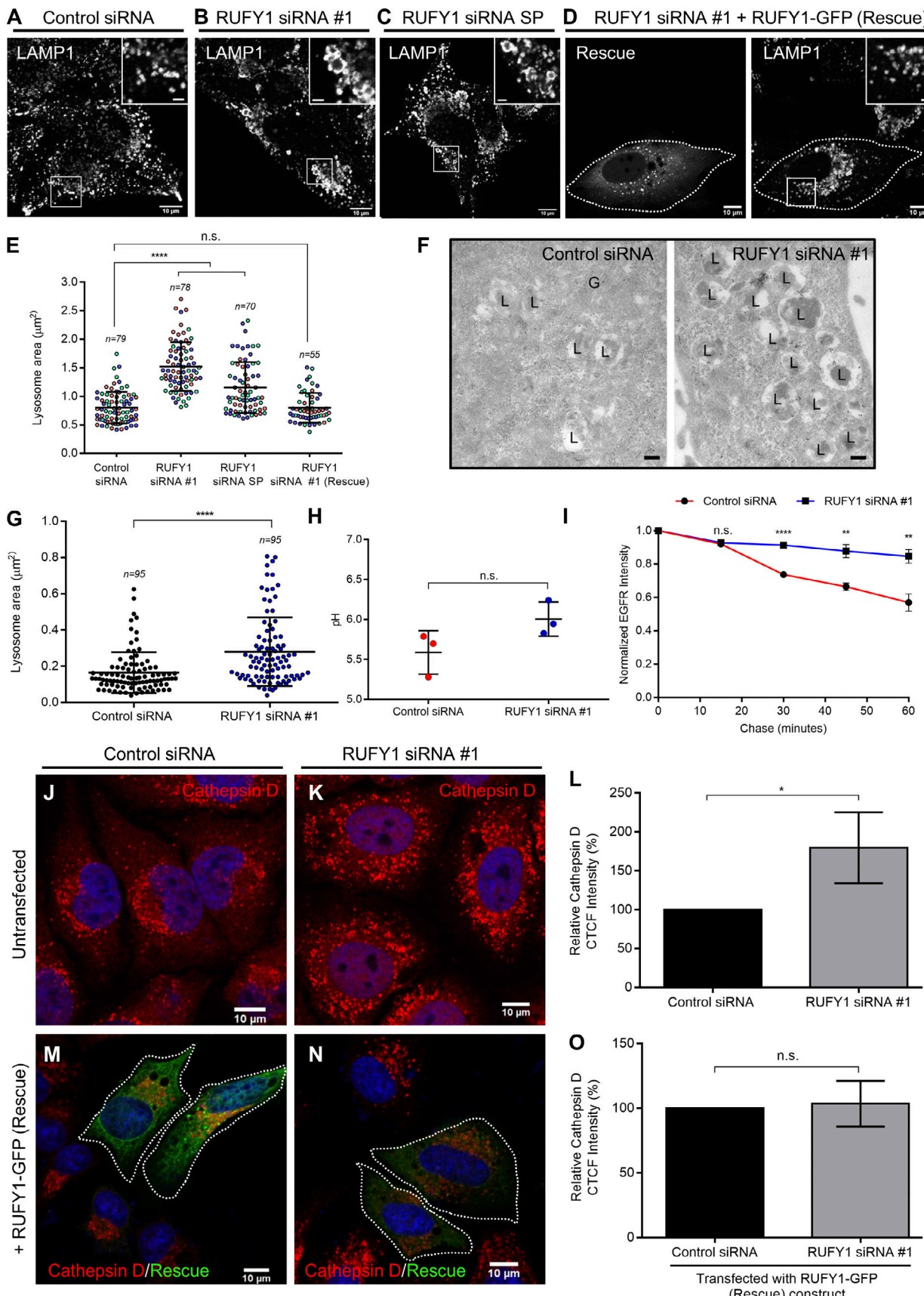

Figure 5. **Depletion of RUFY1 leads to enlarged lysosomes with features suggestive of lysosome dysfunction. (A–C)** Representative confocal micrographs of HeLa cells treated with the indicated siRNA, then immunostained with anti-LAMP1 antibody. **(D)** Representative confocal micrograph of HeLa cells

treated with RUFY1 siRNA #1 and transfected with the RUFY1-GFP (Rescue) construct, followed by immunostaining with anti-LAMP1 antibody. Bars: (main) 10 μm; (insets) 2 μm. **(E)** Quantification of the average area of the lysosomes per cell (LAMP1-positive compartments) upon indicated siRNA treatments. The graph shows the average area of lysosomes per cell. The values plotted are the mean ± SD from three independent experiments. In each experiment, 20–30 cells were analyzed (****P < 0.0001; n.s., not significant; unpaired *t* test). **(F)** Electron micrographs of ultrathin cryosections of HeLa cells treated with either control or RUFY1 siRNA. Cells were fixed and immunogold labeled for endogenous LAMP1 (10 nm gold). Bars: 200 nm. **(G)** Quantification of the lysosome area (LAMP1-positive compartments) from electron micrographs of HeLa cells treated with the indicated siRNA. The error bars represent the mean ± SD of 95 compartments per condition (****P < 0.0001; unpaired two-tailed *t* test). **(H)** Ratiometric measurement of Lysosensor Yellow/Blue DND-160 dye fluorescence in HeLa cells treated with either control or RUFY1 siRNA to assess change in the pH of lysosomes. The data plotted are the mean ± SD from three independent experiments (n.s., not significant; unpaired two-tailed *t* test). **(I)** Serum-starved control siRNA or RUFY1-siRNA-treated HeLa cells were pulsed with EGF (100 ng/ml) for 10 min and chased in complete medium for 15, 30, 45 and 60 min. EGFR degradation was evaluated from immunofluorescence images by normalizing the residual mean EGFR fluorescence intensity at various chase times to the mean EGFR fluorescence intensity in the pulse-only sample. The values plotted are the mean ± SD from three independent experiments with 50–60 cells analyzed per time point in every experiment (****P < 0.0001; **P < 0.01; n.s., not significant; unpaired two-tailed *t* test). **(J and K)** Confocal images of HeLa cells treated with control or RUFY1 siRNA, followed by immunostaining with an anti-cathepsin D antibody. Bars: 10 μm. **(L)** Measurement of Correlated Total Cell Fluorescence (CTCF) values of the cathepsin D signal in HeLa cells treated with the indicated siRNA using ImageJ. Data represent mean ± SD from three independent experiments with 50 cells analyzed per experiment (*P < 0.05; unpaired two-tailed *t* test). **(M and N)** Confocal micrographs of HeLa cells treated with the indicated siRNA and transfected with the RUFY1-GFP (Rescue) construct, followed by immunostaining with an anti-cathepsin D antibody. Bars: 10 μm. **(O)** Measurement of CTCF values of cathepsin D signal in HeLa cells treated with the indicated siRNA and transfected with the RUFY1-GFP (Rescue) construct. Data represents mean ± SD from three independent experiments with 35–50 cells analyzed per experiment (n.s., not significant; unpaired two-tailed *t* test).

Between 260 and 320 min, we observed colocalization of cathepsin Z signal with LAMP1-GFP in RUFY1 knockdown cells, indicating that RUFY1 depletion delays, but does not block, cathepsin Z transit to lysosomes (Fig. 7, I and J). It's reported that impaired recycling of CI-M6PR from endosomes to the TGN results in improper sorting and, consequently, secretion of the precursor form of lysosomal hydrolases (Ghosh et al., 2003). Indeed, we also observed a modest increase in pro-cathepsin D levels in the extracellular media following RUFY1 knockdown (Fig. 7, K and L). By mediating CI-M6PR recycling from endosomes back to the TGN, RUFY1 regulates sorting of CI-M6PR cargo, namely lysosomal hydrolases, to late endosomes and lysosomes.

## RUFY1 interacts with the dynein-dynactin complex and promotes perinuclear organelle clustering in a dynein-dependent manner

Next, we sought to identify putative RUFY1 interaction partners in order to get insight into how it regulates CI-M6PR trafficking. To achieve this goal, we utilized tandem affinity pulldown using TAP-tagged RUFY1 (longer isoform) and mass spectrometry to identify probable hits. Dynein heavy chain (DHC) was the second most abundant hit in the TAP eluate, second only to RUFY1 (Table S1). We also found other subunits of the dynein motor complex, including dynein light intermediate chain 1 (LIC1), dynein light intermediate chain 2 (LIC2), and dynein intermediate chain (DIC) 2, as well as subunits of the dynactin complex, including p50 (dynamitin) and p150^glued (dynactin; Table S2). Importantly, two previous studies have also reported dynein subunits as interaction partners of RUFY1, although the significance of this interaction was not explored (Ivan et al., 2012; Redwine et al., 2017). Noticeably, we did not find peptides belonging to Rab14 or Arl8b in this TAP pulldown, most likely due to the low affinity of these interactions.

In accordance with the mass spectrometry results, we detected RUFY1 interaction with the dynein-dynactin subunits-DIC and p150^glued under endogenous conditions (Fig. 8 A). As a control, we also performed immunoprecipitation of an unrelated cytosolic protein to demonstrate the specificity of RUFY1

interaction with the dynein-dynactin complex (Fig. S5 A). RUFY1, notably, possesses a long central coiled-coil domain (∼300 amino acids in length), which is a characteristic property of the activating class of dynein-dynactin adaptors (henceforth referred to as adaptors) such as BICD2 and Hook proteins (Schroeder et al., 2014; Schroeder and Vale, 2016; Fig. S5 B shows the prediction from AlphaFold (Jumper et al., 2021; Varadi et al., 2022) for human proteins RUFY1, BICD2, and HOOK3). Similar to these adaptors, we found that RUFY1 interacts with the C-terminal region of LIC1, which is utilized as a bait protein in pulldown assays (Fig. 8 B). This interaction was dependent on the RUN domain-containing region of RUFY1, since the binding of the RUFY1 (ΔRUN) mutant to LIC1 was significantly reduced as compared to RUFY1 (WT; Fig. 8 B). According to recent studies, the C-terminal amphipathic helix of LIC1 contains highly conserved phenylalanine residues (F447 and F448 in human LIC1) that insert into the hydrophobic pocket of the adaptors (Lee et al., 2020; Lee et al., 2018). Using a GST pulldown assay, we discovered that mutation of these conserved F447/F448 to alanine (FFAA) residues in LIC1 significantly reduced the binding between RUFY1 and LIC1, indicating that RUFY1 and LIC1 interact in a manner similar to that of other activating dynein adaptors (Fig. 8, C and D).

Next, we investigated whether the dynein-dynactin complex governs RUFY1 distribution. Indeed, RUFY1 distribution was drastically altered and displaced toward the cell periphery in cells treated with dynein heavy chain (DHC) siRNA (Fig. S5, C–E). Immunostaining with anti-CI-M6PR antibody revealed that RUFY1 endosomes found in DHC siRNA were also positive for CI-M6PR (Fig. S5, C, D, and F). As expected, Rab14 distribution also showed a similar displacement towards the cell periphery in DHC siRNA (Fig. S5, G–I), implying that dynein is required for the typical perinuclear distribution of endosomes containing Rab14, RUFY1 and CI-M6PR.

Next, we utilized the FRB-FKBP rapamycin-induced heterodimerization strategy to induce RUFY1 localization to mitochondria in order to determine whether RUFY1 was sufficient to drive mitochondrial perinuclear distribution in a dynein-dependent manner, i.e., whether RUFY1 acts as an organelle-

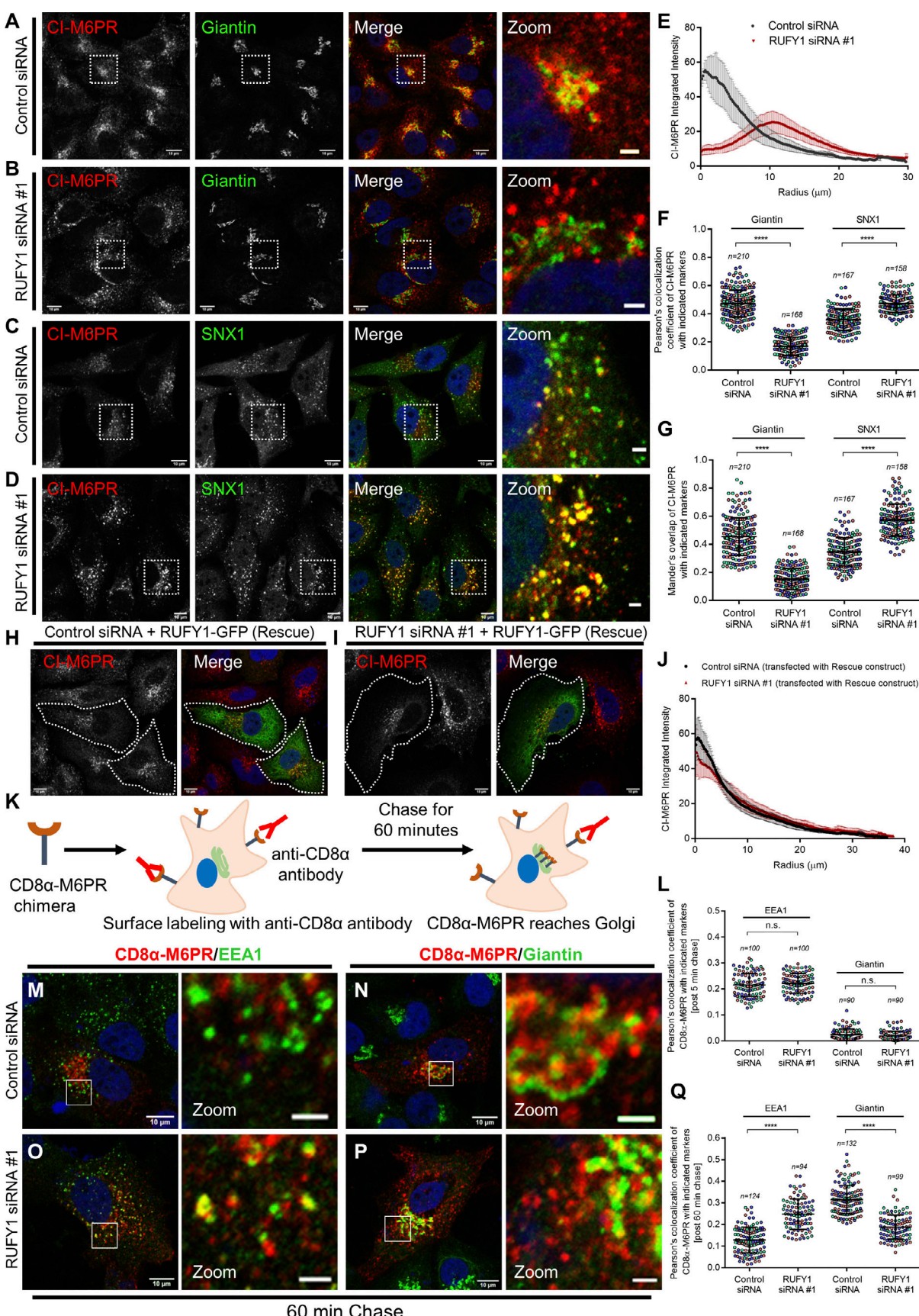

Figure 6. **RUFY1 regulates retrieval of CI-M6PR from endosomes to the TGN. (A–D)** Representative confocal images of HeLa cells treated with the indicated siRNAs, followed by immunostaining for CI-M6PR and co-stained with Giantin or SNX1 (as labeled). Bars: (main) 10 µm; (insets) 2 µm. **(E)** Radial

profile plot of the CI-M6PR intensity distribution in HeLa cells treated with indicated siRNAs. The values plotted are the mean ± SD from three independent experiments, with 30–40 cells analyzed per experiment. **(F and G)** Pearson's and Mander's colocalization coefficient quantification of CI-M6PR with Giantin and SNX1 (as indicated in the graph) in control siRNA and RUFY1 siRNA-treated HeLa cells. The values plotted are the mean ± SD from three independent experiments. Experiments are color coded, and each dot represents the individual data points from each experiment. The total number of cells analyzed is indicated on the top of each data set (****P < 0.0001; unpaired two-tailed *t* test). **(H and I)** Representative confocal micrographs of HeLa cells treated with the indicated siRNA and transfected with the RUFY1-GFP (Rescue) construct, followed by immunostaining with an anti-CI-M6PR antibody. Cells expressing rescue construct are marked with a white boundary. Bars: 10 μm. **(J)** Radial profile plot of the CI-M6PR intensity distribution in HeLa cells treated with the indicated siRNAs and transfected with the RUFY1-GFP (Rescue) construct. The values plotted are the mean ± SD from three independent experiments, with 30–40 cells analyzed per experiment. **(K)** Schematic illustrating CD8α-M6PR trafficking assay. The anti-CD8 primary antibody was used for labeling the cell surface receptor population, followed by a chase of the internalized CD8α-M6PR for either 5 min or 60 min. Cells were fixed and immunostained with secondary antibodies. **(L)** Pearson's colocalization coefficient quantification of endocytosed CD8α-M6PR with EEA1 and Giantin markers after 5 min of chase. Cells were transfected with CD8α-M6PR chimera construct after 60 h of the indicated siRNA treatments, followed by labeling cell surface CD8α-M6PR receptors and chase for 5 min post internalization. Cells were fixed and immunostained for EEA1 and Giantin. **(M–P)** Representative confocal images of endocytosed CD8α-M6PR at 60 min post internalization. Cells were fixed and immunostained for EEA1 and Giantin. Bars: (main) 10 μm; (insets) 2 μm. **(Q)** Pearson's colocalization coefficient quantification of endocytosed CD8α-M6PR with EEA1 and Giantin markers at 60 min of the chase in HeLa cells treated with indicated siRNA. For L and Q, the values plotted are the mean ± SD from three independent experiments. Experiments are color coded, and each dot represents the individual data points from each experiment. The total number of cells analyzed is indicated on the top of each data set (****P < 0.0001; n.s., not significant; unpaired two-tailed *t* test).

specific dynein adaptor. The mitochondrial localization of FKBP12-GFP (vector-transfected) and FKBP12-GFP-RUFY1 in rapamycin-treated cells co-expressing FRB-Tom-70p (Mito-FRB) served as initial confirmation that the heterodimerization strategy was working (compare Fig. 8, E and G to Fig. 8, F and H). Significantly, RUFY1 localization led to a substantial mitochondrial clustering in the perinuclear area, whereas vector-transfected cells had a typical mitochondrial distribution (compare Fig. 8 F to Fig. 8 H, see intensity profile distribution in Fig. 8 I). Next, we determined if RUFY1-mediated mitochondrial clustering requires dynein-driven transport to the perinuclear area. To do this, we depleted DHC and evaluated mitochondrial distribution in rapamycin-treated FKBP12-GFP-RUFY1 transfected cells (Fig. 8, J–L). Indeed, mitochondria failed to cluster in RUFY1-transfected cells when dynein was depleted (Fig. 8, K and L), indicating that RUFY1 recruits dynein to transport organelles/endosomes towards the perinuclear area.

### RUFY1 coiled-coil domains CC1 and CC2 are required for perinuclear organelle clustering and for interaction with the dynein-dynactin complex

Previous studies have shown that the long coiled-coil segment of ~350 A° length of the activating dynein adaptor proteins runs parallel to the dynactin filament and serves as a binding surface for dynein tails. As a result, engaging dynactin and forming a stable tripartite dynein-dynactin-adaptor complex requires a minimum length of the coiled-coil segment (Grotjahn et al., 2018; Schroeder and Vale, 2016; Urnavicius et al., 2015). We created FKBP12-GFP-tagged domain deletion mutants of RUFY1 with progressive deletion of its C-terminal end to test if RUFY1 requires a minimal length of its coiled-coil region for perinuclear organelle clustering (Fig. 9 A). As illustrated in Fig. 9, B, C, and J, FKBP12-GFP-RUFY1 (1–500 amino acids containing the first and second coiled-coil regions) promoted mitochondrial perinuclear clustering similar to or sometimes better than RUFY1 (WT). However, a RUFY1 deletion mutant lacking a region upwards of the second coiled-coil region (1–400 and 1–300 amino acids) resulted in a significant decrease in mitochondrial perinuclear clustering when compared to RUFY1 (WT; Fig. 9, D, E, and J).

We then evaluated the importance of the first two coiled-coil regions by creating internal deletions of RUFY1 that were missing either the first (RUFY1 [ΔCC1]; lacking 300–400 amino acids) or the second (RUFY1 [ΔCC2]; lacking 400–500 amino acids) or both (RUFY1 [ΔCC1 + ΔCC2]; lacking 300–500 amino acids) coiled-coil regions. We first confirmed that all the internal domain-deletion FKBP12-GFP-RUFY1 mutants showed colocalization with Rab14, indicating that the mutants were properly folded and the C-terminal Rab14-binding site was in the proper conformation (Fig. S5, J–M). In the presence of Mito-FRB expression and rapamycin-treatment, we observed that the RUFY1 mutant lacking either the second coiled-coil (ΔCC2) or both coiled-coil regions (ΔCC1 + ΔCC2) had a more severe loss of mitochondrial perinuclear clustering ability than the mutant lacking the first coiled-coil region (ΔCC1; Fig. 9, F–H and J). Similarly, a RUFY1 (ΔRUN) mutant that does not interact with LIC1, was unable to promote perinuclear mitochondrial clustering (Fig. 9, I and J), indicating that LIC1 interaction is required for RUFY1 to drive dynein-dependent organelle dispersal. The results from the organelle-clustering assay were supported by an observed decrease in binding of the RUFY1 mutants with LIC1, as compared to RUFY1 (WT; Fig. 9, K and L).

### RUFY1 interaction with the dynein-dynactin complex is required for CI-M6PR retrieval from endosomes to the TGN

Finally, we analyzed whether RUFY1 influences the retrograde motility of CI-M6PR-containing vesicles and whether RUFY1 binding to dynein is required for CI-M6PR retrieval back to the TGN. To this end, we first investigated the motility of CI-M6PR vesicles from the cell surface to the TGN in the absence of RUFY1. We observed that CD8α-M6PR localized to the SNX1 compartment within 10–15 min of endocytosis and to the Golgi at ~30 min (Fig. 10, A and C). Several CD8α-M6PR vesicles were closely juxtaposed to RUFY1 endosomes within 10–15 min of endocytosis and remained so for the length of the imaging, indicating that CD8α-M6PR traffics to the RUFY1 compartment at a similar time point as SNX1 (Fig. 10 B). SNX1-positive vesicles were also persistently connected with RUFY1 endosomes, and SNX1-positive tubules occasionally emerged from RUFY1 endosomes (Video 2).

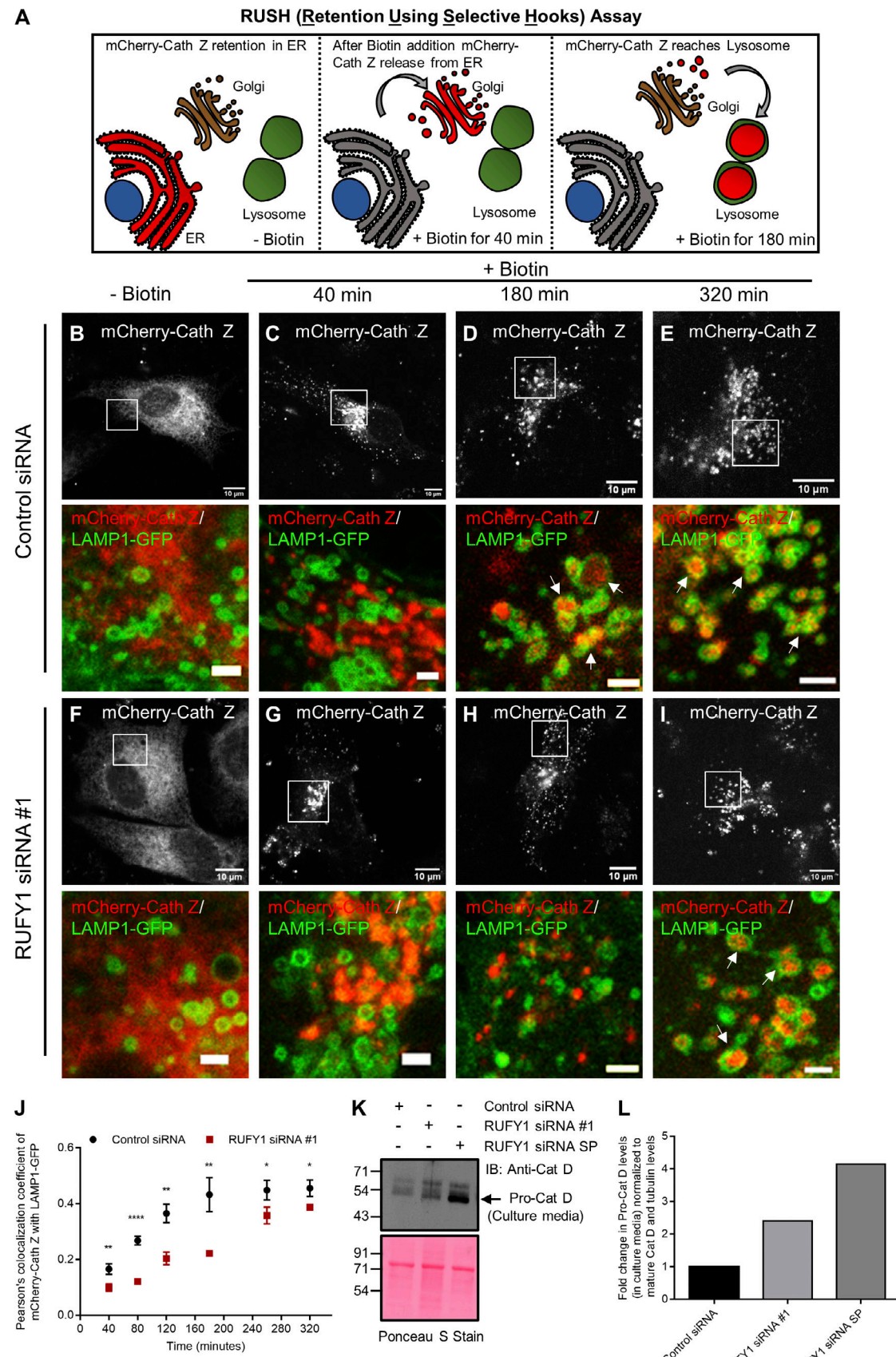

**Figure 7. RUFY1 depletion impairs CI-M6PR cargo, cathepsin Z, delivery to lysosomes. (A)** Schematic representation of the RUSH assay to study the trafficking of CI-M6PR cargo, mCherry-Cathepsin Z (Cath Z), from the endoplasmic reticulum (ER) to the lysosome. SBP-mCherry-CathZ is retained in the ER

via its interaction with the hook Str-KDEL. Upon biotin addition, mCherry-Cath Z releases from the ER and over time reaches the Golgi and subsequently to the lysosomes. **(B–I)** Representative confocal images of a RUSH experiment performed in HeLa cells treated with control or RUFY1 siRNA. Following 60 h of indicated siRNA treatment; cells were co-transfected with Str-KDEL-IRES-SBP-mCherry-Cath Z and LAMP1-GFP constructs. Different transfected control and RUFY1 depleted cells were imaged at 37°C prior to biotin addition and at various time points post biotin addition. Representative images of different cells at 40, 180, and 320 min post biotin addition are shown. Bars: (main) 10 μm; (insets) 2 μm. **(J)** Pearson's colocalization coefficient was quantified for the cargo, mCherry-Cath Z, with LAMP1-GFP at different time points after the addition of biotin by drawing different ROIs in cells. Data represents mean ± SD from 30 to 40 cells in total from three independent experiments, and in each cell, 2–3 ROI were selected for analysis (****$P < 0.0001$; **$P < 0.01$; *$P < 0.05$; unpaired two-tailed $t$ test). **(K)** Immunoblot of pro-cathepsin D secretion assay performed in HeLa cells treated with the indicated siRNA. TCA precipitated proteins from cell culture media of siRNA-treated cells were immunoblotted with an anti-cathepsin D antibody. Ponceau S stain was done to visualize equal loading of proteins. **(L)** Densitometric analysis of pro-cathepsin D band intensity in culture media normalized to mature cathepsin and α-tubulin band intensity in total cell lysates. The averaged values from two independent experiments are plotted. Source data are available for this figure: SourceData F7.

We evaluated the 2D trajectories of CD8α-M6PR endocytosed from the plasma membrane using a semi-automated, custom-written particle-tracking program (Mohan et al., 2019). We analyzed ~200 tracks in control cells and found that 28% of CD8α-M6PR tracks were "processive," 27% were "non-processive," and the remainder of tracks were "diffusive" (Fig. 10 D; see the Materials and methods section for a definition; Videos 3, 4, and 5 feature examples of the three different types of tracks). In contrast, upon RUFY1 depletion, the majority of vesicle tracks were non-processive, whereas processive and diffusive tracks dramatically decreased, indicating that the majority of M6PR endosomes displayed non-processive motility (Fig. 10 D). Analysis of vesicle track lengths also revealed a reduction in CD8α-M6PR vesicles with run lengths of 4–5 μm and an increase in those with run lengths of 1–2 μm upon RUFY1 depletion (Fig. 10 E). Run lengths of >2 μm are indicative of dynein-based motility, hence the general trend of lower processive run lengths in RUFY1-depleted cells is noteworthy (Flores-Rodriguez et al., 2011).

To test whether dynein binding is required for RUFY1's role in mediating CI-M6PR sorting from endosomes to the TGN, we transfected either RUFY1 (WT) or RUFY1 (ΔCC2) siRNA-resistant construct in RUFY1 knockdown cells to rescue the CI-M6PR localization defect. As illustrated in Fig. 10, F–I, unlike the WT construct, RUFY1 (ΔCC2) was not able to rescue the CI-M6PR localization defect, as demonstrated by a reduced colocalization of CI-M6PR with Giantin. We conclude that RUFY1 is required for dynein-dependent motility and sorting of CI-M6PR from endosomes to the TGN.

## Discussion

Like other small G proteins of the Rab, Arf, and Arl family, Arl8b plays multiple roles at the compartment it primarily localizes to, i.e., late endosomes/lysosomes. These functions include positioning, assembly of core machinery for fusion with other vesicles, tubulation, and exocytosis of lysosomes (Khatter et al., 2015b). Arl8b recruits effectors and interaction partners with distinct roles, such as motor adaptor (PLEKHM2), tethering factor (HOPS complex), and fusion assembly adaptor (PLEKHM1; Garg et al., 2011; Khatter et al., 2015a; Marwaha et al., 2017; Rosa-Ferreira and Munro, 2011).

In this study, we have identified a new interaction partner of Arl8b, RUFY1, that has been previously shown to mediate cargo sorting from early/recycling endosomes (Nag et al., 2018;

Vukmirica et al., 2006; Yamamoto et al., 2010). RUFY1 colocalizes with Arl8b on Rab14-positive recycling endosomes (Fig. 10 J). Rab14 localizes to recycling endosomes, Golgi, and the TGN compartments, with earlier studies depicting that active or GTP-bound Rab14 is endosomal while GDP-bound Rab14 is Golgi-localized (Junutula et al., 2004). Rab14, like Rab4, is most likely found on sorting endosomes where recycling cargo is separated from cargo bound for TGN and/or storage vesicles (Leto and Saltiel, 2012; Linford et al., 2012). Our study demonstrates that in addition to its well-known localization to late endosomes/lysosomes, a subpopulation of Arl8b localizes to recycling/sorting endosomes.

We anticipate that functionally, Arl8b residing on this subpopulation might regulate lysosome cargo sorting along with RUFY1 and its interaction partner, AP-3, a well-established regulator of cargo sorting to late endosomes and lysosomes (Dell' Angelica, 2009). For instance, one of the cargos traversing this route is the lysosomal glycoprotein LAMP1 (an AP-3-dependent cargo [Peden et al., 2004]). It will be relevant to determine whether sorting of lysosomal glycoproteins is also dependent on Arl8b and RUFY1 interaction. Similarly, the glycoprotein Progranulin, which is converted into granulin peptides in lysosomes, is one of the cargoes for the sortilin receptor that like CI-M6PR traffic through the endosome-TGN pathway (Hu et al., 2010; Talbot et al., 2018). Loss-of-function mutations in progranulin lead to lysosome malfunction and are associated with frontotemporal lobar degeneration, a neurodegenerative illness (Paushter et al., 2018). Notably, several of the lysosomal abnormalities described in progranulin gene knockout animals, such as enlargement of acidic compartments, increased cathepsin levels, and decreased cargo degradation (Paushter et al., 2018), are also observed in RUFY1-depleted cells. Future research should determine whether Arl8b and RUFY1 regulate progranulin trafficking to lysosomes and whether the lysosomal dysfunction found in RUFY1 depletion is associated with progranulin processing defects. Furthermore, whether Arl8b regulates cargo sorting to other storage compartments is an intriguing question for future research. Indeed, RUFY1 and Rab14 are both involved in the insulin-dependent sorting of the glucose transporter Glut4 into Glut4 storage vesicles (GSVs), also RUFY1 regulates cargo sorting to melanosomes (Mari et al., 2006; Nag et al., 2018; Reed et al., 2013; Sadacca et al., 2013).

Our findings show that Arl8b regulates RUFY1 localization by directly promoting RUFY1-Rab14 interaction. We observed that removing the RUFY1 RUN domain-containing region increased

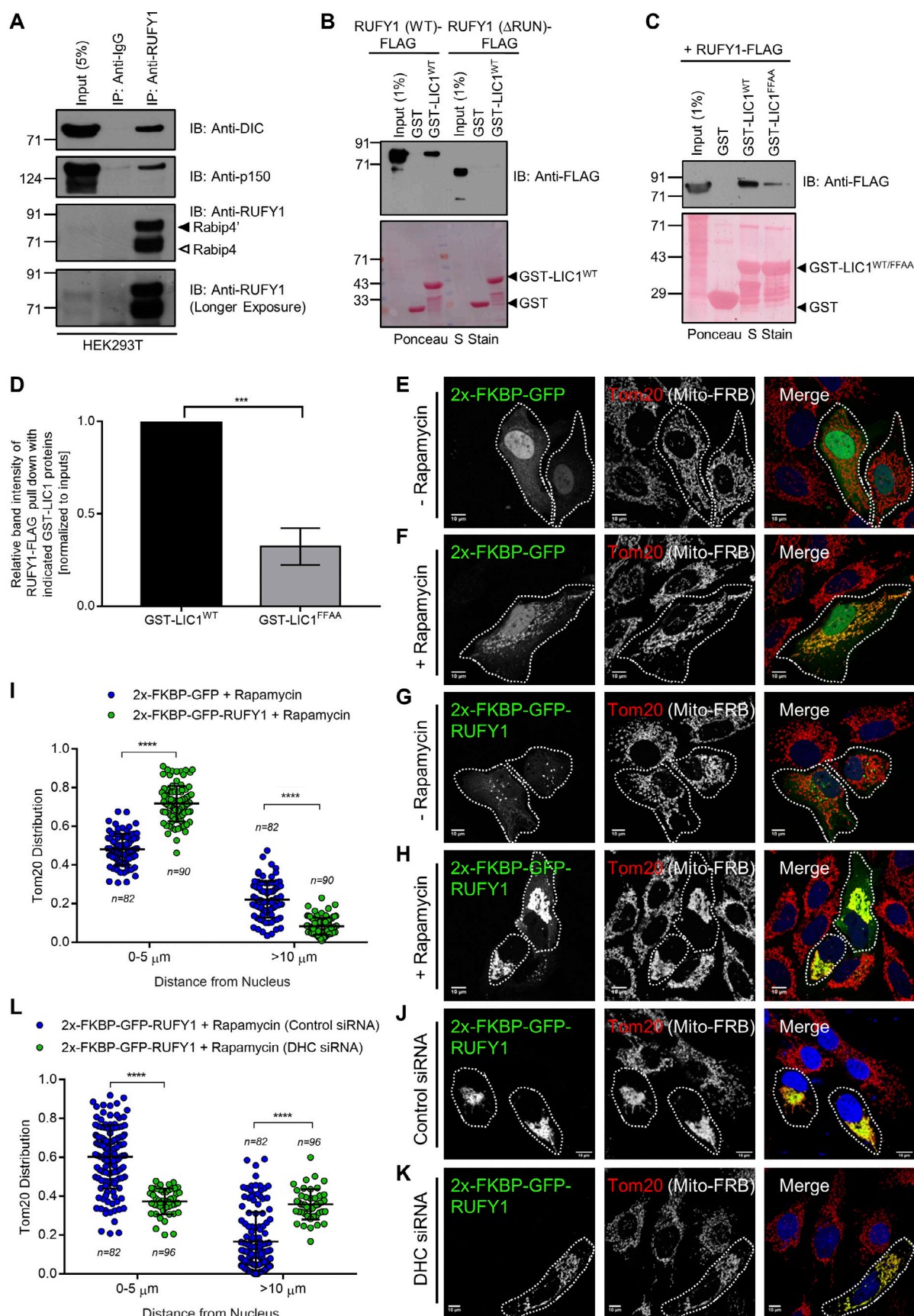

Figure 8. **RUFY1 interacts with the dynein-dynactin complex and acts as a dynein cargo adaptor. (A)** HEK293T cell lysates were immunoprecipitated using anti-RUFY1 antibody bound to Protein-A/G beads, and the precipitates were immunoblotted (IB) with indicated antibodies. **(B)** Recombinant GST and

GST-LIC1$^{WT}$ (389–523 a.a.) proteins were immobilized on glutathione-coated-agarose beads and incubated with HEK293T cell lysates expressing RUFY1 (WT)-FLAG or RUFY1 (ΔRUN)-FLAG. The precipitates were IB with anti-FLAG antibody and Ponceau S staining was done to visualize the purified proteins. **(C)** Recombinant GST, GST-LIC1$^{WT}$, and GST-LIC1$^{FFAA}$ (F447A/F448A) proteins were immobilized on glutathione-coated-agarose beads and incubated with HEK293T cell lysates expressing RUFY1 (WT)-FLAG. The precipitates were IB with anti-FLAG antibody and Ponceau S staining was done to visualize the purified proteins. **(D)** Densitometric analysis of RUFY1 (WT) pulldown using the indicated GST-fusion proteins (normalized to input signals) is shown. The values plotted are the mean ± SD from three independent experiments (***P < 0.001; unpaired two-tailed t test). **(E–H)** Representative confocal images of HeLa cells transiently expressing FRB-Tom70p with 2x-FKBP-GFP (E and F) or 2x-FKBP-GFP-RUFY1 (G and H) treated with or without rapamycin, followed by immunostaining with anti-Tom20 antibodies to visualize mitochondria. A white boundary marks the co-transfected cells. Bars: 10 µm. **(I)** The Tom20 signal intensity profile was quantified with respect to distance from the nucleus of HeLa cells expressing indicated FRB-FKBP fusion proteins upon addition of rapamycin. The values plotted are the mean ± SD from three independent experiments. Experiments are color-coded, and each dot represents the individual data points from each experiment. The total number of cells analyzed is indicated on the top or bottom of each data set (****P < 0.0001; unpaired two-tailed t test). **(J and K)** Representative confocal micrographs of HeLa cells treated with the indicated siRNAs and co-transfected with FRB-Tom70p and 2x-FKBP-GFP-RUFY1 constructs, followed by 2 h treatment with rapamycin before fixation. Cells were immunostained with an anti-Tom20 antibody to visualize mitochondria. A white boundary marks the co-transfected cells. Bars: 10 µm. **(L)** The Tom20 signal intensity profile was quantified with respect to distance from the nucleus of HeLa cells treated with indicated siRNA and expressing FRB-FKBP fusion proteins in the presence of rapamycin. The values plotted are the mean ± SD from three independent experiments. Experiments are color coded, and each dot represents the individual data points from each experiment. The total number of cells analyzed is indicated on the top or bottom of each data set (****P < 0.0001; unpaired two-tailed t test). Source data are available for this figure: SourceData F8.

binding to Rab14, implying that this N-terminal region has an autoinhibitory role. Based on these findings, we propose a model in which Arl8b binding to this region relieves autoinhibition and promotes a conformational change in RUFY1 that allows Rab14 binding. Interestingly, a previous study found that Arl8b plays a similar role in relieving autoinhibition of its effector SKIP in order to promote SKIP interaction with the anterograde motor kinesin-1 (Keren-Kaplan and Bonifacino, 2021).

Our findings suggest that the presence of RUFY1 on Rab14 endosomes is required for the dynein-dependent retrograde transport of cargo from endosomes towards TGN or storage compartments (Fig. 10 J). Regarding how RUFY1 governs the dynein-based motility of cargo-containing endosomes, it will be important to determine whether RUFY1 functions as an activating adaptor of dynein, connecting the dynein motor to the dynactin complex in a configuration suitable for processive motility. Akin to other activating adaptors, the coiled-coil region of RUFY1 is ~300 amino acids, which is an appropriate length to run along the dynactin filament (Olenick and Holzbaur, 2019; Reck-Peterson et al., 2018). Our findings show that reducing the length of the coiled-coil region, and specifically deleting the second coiled-coil region, impairs RUFY1's ability to interact with dynein and promote dynein-dependent organelle clustering.

In addition, RUFY1, like a subset of activating adaptors, has a binding site for the C-terminal adaptor-binding region of LIC1 that is upstream to the coiled-coil domains (Lee et al., 2020). Excitingly, recent research has demonstrated that RUFY3 and RUFY4 are Arl8 effectors that localize to late endosomes and lysosomes and interact with the dynein-dynactin complex to facilitate dynein-dependent retrograde motility of lysosomes (Keren-Kaplan et al., 2022; Kumar et al., 2022). We hypothesize that the RUFY family of proteins are a new class of activating dynein cargo adaptors that interact with Arl8b on distinct membranes (i.e., recycling endosomes [for RUFY1 and RUFY2] or lysosomes [for RUFY3 and RUFY4]) and mediate processive motility of the target compartments.

In conclusion, our findings suggest an unexpected collaboration between Arl8b and Rab14 to recruit their common interaction partner RUFY1 on endosomes, which mediates cargo sorting at this endosome towards the Golgi and/or storage compartments. It is unknown whether the two pools of Arl8b on early/recycling endosomes and lysosomes, respectively, are mutually exclusive or whether the Rab14 compartment represents a transit point in Arl8b's journey towards the newly formed lysosomes.

## Materials and methods
### Cell culture and RNAi
HeLa and HEK293T (from ATCC) were cultured in DMEM (Gibco) supplemented with 10% FBS (Gibco) in a humidified chamber with 5% $CO_2$ at 37°C. Arl8b$^{-/-}$ KO HeLa cells used in this study were described previously (Marwaha et al., 2017). Briefly, Arl8b$^{-/-}$KO HeLa cell line was generated using CRISPR/Cas9 methodology using sgRNA target sequence: 5′-GATGGAGCTGACGCTCG-3′. All the cell lines were subcultured for not more than 18 passages and regularly screened for the absence of mycoplasma contamination by the MycoAlert Mycoplasma Detection Kit (Lonza).

The siRNA oligos for gene silencing studies were purchased from Dharmacon (Horizon Discovery) and prepared according to the manufacturer's instructions. Transient transfection of siRNAs was performed with DharmaFECT 1 reagent (Dharmacon) according to the manufacturer's instructions. Following siRNA oligos were used in the study: Control siRNA, 5′-TGGTTTACATGTCGACTAATT-3′; RUFY1 siRNA #1, 5′-CATCAGATATAGCGCTAGTT-3′; RUFY1 siRNA #2: 5′-ATAAACATCTCTTAAGCGATT-3′; RUFY1 siRNA SMARTpool (SP), ON-TARGETplus SMARTpool (L-016355-00-0005); Arl8b siRNA, 5′-AGGTAACGTCACAATAAAGAT-3′; Rab14 siRNA, 5′-CAACTACTCTTACATCTTTTT-3′; and DHC siRNA, 5′-GAGAGGAGGTTATGTTTAATT-3′.

To generate HEK293T cells stably expressing N-term TAP-tagged-RUFY1, lentiviral transduction was performed as described previously (Garg et al., 2011). Briefly, for lentiviral transduction, HEK293T cells were plated at 100,000/well in 6-well plates (Corning) in complete media containing 8 µg/ml Polybrene (Sigma-Aldrich) and transduced by addition of 100 µl viral supernatant. 24 h later, media was replaced and fresh

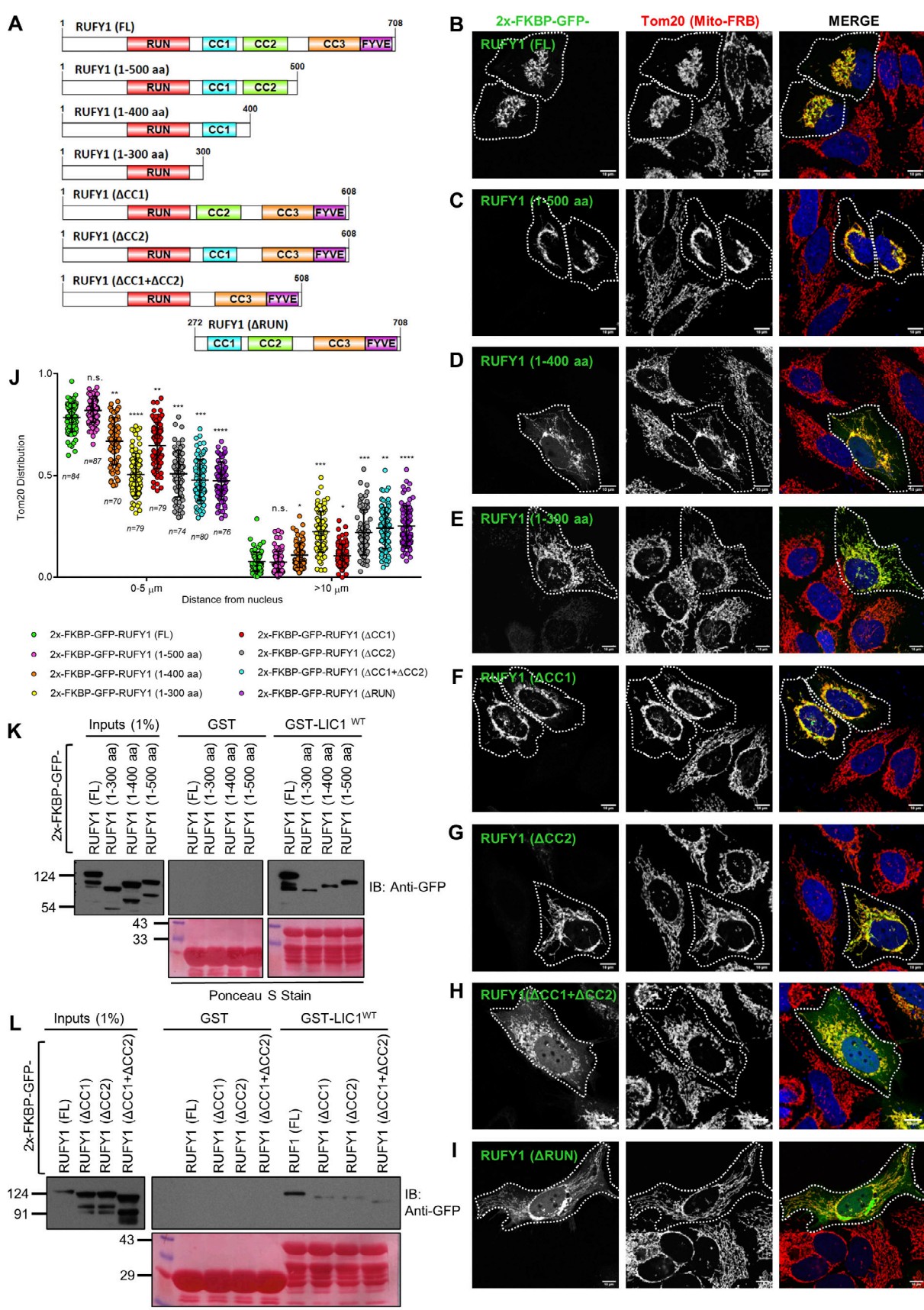

Figure 9. **The first two coiled-coil domains of RUFY1 are required for its function as a dynein cargo adaptor and interaction with dynein subunit LIC1.** **(A)** Schematic representation of the domain architecture of RUFY1 (full-length; FL) and mutants with progressive deletion of its C-terminus and internal

deletions lacking either the first (300–400 a.a.) or second (400–500 a.a.) or both coiled-coil regions (300–500 a.a.). The RUFY1 (ΔRUN) mutant lacks the 271 a.a. from the N-terminal. **(B–I)** Representative confocal images of HeLa cells transiently expressing FRB-Tom70p with 2x-FKBP-GFP-RUFY1 (FL) or mutants (as described in A) and treated without rapamycin, followed by immunostaining with an anti-Tom20 antibody to visualize mitochondria. A white boundary marks the co-transfected cells. Bars: 10 μm. **(J)** The Tom20 signal intensity profile was quantified with respect to distance from the nucleus of HeLa cells expressing indicated FRB-FKBP fusion proteins upon addition of rapamycin. The values plotted are the mean ± SD from three independent experiments. Experiments are color-coded, and each dot represents the individual data points from each experiment. The total number of cells analyzed is indicated on the bottom of each data set (*P < 0.05, **P < 0.01, ***P < 0.001 ****P < 0.0001; n.s., not significant; unpaired two-tailed *t* test). **(K and L)** Recombinant GST and GST-LIC1$^{WT}$ proteins were immobilized on glutathione-coated-agarose beads and incubated with HEK293T cell lysates expressing 2x-FKBP-GFP-RUFY1 (FL) or mutants (as indicated). The precipitates were immunoblotted (IB) with an anti-GFP antibody and Ponceau S staining was done to visualize the purified proteins. Source data are available for this figure: SourceData F9.

media containing 3 μg/ml Puromycin (Sigma-Aldrich) was added to select transductants and experiments were performed on days 5–21 following transduction after verifying the expression of TAP-RUFY1 by immunofluorescence and Western blotting.

### Plasmids

All the mammalian, yeast and bacterial expression plasmids and primer sequences used in this study are listed in Table S2.

### Antibodies and chemicals

All the antibodies used in this study are listed in Table S3. Lysotracker red DND-99, DAPI and EGF were purchased from Invitrogen. Imidazole, Glutathione, IPTG (isopropylthio-β-galactoside), Polybrene, Puromycin, and Rapamycin were purchased from Sigma-Aldrich.

### Transfection, immunofluorescence and live-cell imaging

Cells were grown on glass coverslips (VWR), and desired mammalian expression plasmid transfections were performed using XtremeGene HP (Roche). Post 16–18 h of transfection, cells were fixed with 4% paraformaldehyde (PFA; w/v) in PHEM buffer (60 mM PIPES, 10 mM EGTA, 25 mM HEPES, and 2 mM MgCl$_2$, final pH 6.8) at room temperature (RT) for 10 min. Immunostaining was performed as described previously (Marwaha et al., 2017). Briefly, fixed cells were incubated with blocking solution (PHEM + 0.2% saponin + 5% FBS) for 30 min at RT followed by incubation with primary antibodies diluted in PHEM + 0.2% saponin for 2 h. After washing cells three times with 1X PBS, cells were incubated with appropriate secondary antibodies diluted in PHEM + 0.2% saponin for 30 min at RT. Finally, coverslips were washed thrice with 1X PBS and mounted on glass slides in Fluoromount-G (Southern Biotech). Single-plane confocal images were acquired using LSM710 confocal microscope using 63×/1.4 NA oil immersion objective, and Zen Black 2012 software (ZEISS) was used to acquire images. All the representative confocal images were adjusted for brightness and contrast using ImageJ (National Institutes of Health) and Adobe Photoshop CS6 software.

For live-cell imaging experiments, cells were seeded on glass-bottom live-cell imaging dishes (Eppendorf and Ibidi), followed by transfection of indicated plasmids. After 12 h of transfection, the media of cells was replaced by phenol red-free DMEM (Gibco), and the dish was placed in a live-cell imaging chamber maintained at 37°C with a 5% CO$_2$ supply. Time-lapse imaging was performed on LSM 710 confocal microscope with 63×/1.4 NA oil immersion objective. Live-cell imaging videos were acquired using Zen Black 2012 software (ZEISS), and final adjustments for brightness and contrast were done using ImageJ.

### Structured illumination microscopy (SIM)

To carry out SIM imaging, cells were fixed and immunostained as described above. SIM images were captured with Zeiss Elyra 7 (Lattice SIM technology) using either Plan Apo 40×/1.40 oil or Plan Apo 63×/1.40 oil objective and sCMOS camera (PCO Edge). All the single-plane 2D images were captured using Lattice SIM acquisition mode in which 15 phase images were acquired with 1,280 × 1,280 pixel resolution and SIM processing done by ZEN Black (ZEISS) software to give the final super-resolved image. All the representative SIM images presented in the figures were adjusted for brightness and contrast using Adobe Photoshop CS6 software. For live-cell imaging experiments in SIM mode, cells were seeded on a glass-bottom live-cell imaging dish and transfected with indicated plasmids for 12 h. Following incubation, time-lapse imaging was performed in phenol red-free DMEM in a live-cell imaging chamber maintained at 37°C with 5% CO$_2$ and 95% humidity using Plan Apo 40×/1.40 oil objective.

### Image analysis and quantification
#### Colocalization analysis

For calculating colocalization, images were opened in ImageJ and max entropy thresholding was applied; Pearson's correlation coefficient (PCC) and Mander's overlap (M1 and M2) were measured using the JaCOP plugin.

#### Analysis of LAMP1-positive vesicles

For measuring the size of LAMP1-positive compartments, a cell was selected using a freehand selection tool, and surrounding cells were removed using the "clear outside" option from the "Edit function" of ImageJ and then "Max Entropy" threshold was applied. Further, the lysosome area was measured using the Analyze particles function of ImageJ software.

#### Quantification of corrected total cell fluorescence (CTCF)

Cathepsin D CTCF values in control and RUFY1 siRNA #1 treated cells were calculated using the formula CTCF = Integrated density—(area of the selected cell X mean fluorescence of background). Images were imported into ImageJ software, and the parameters required to calculate CTCF were derived using the "Measure" option in the "Analyze" function of ImageJ software.

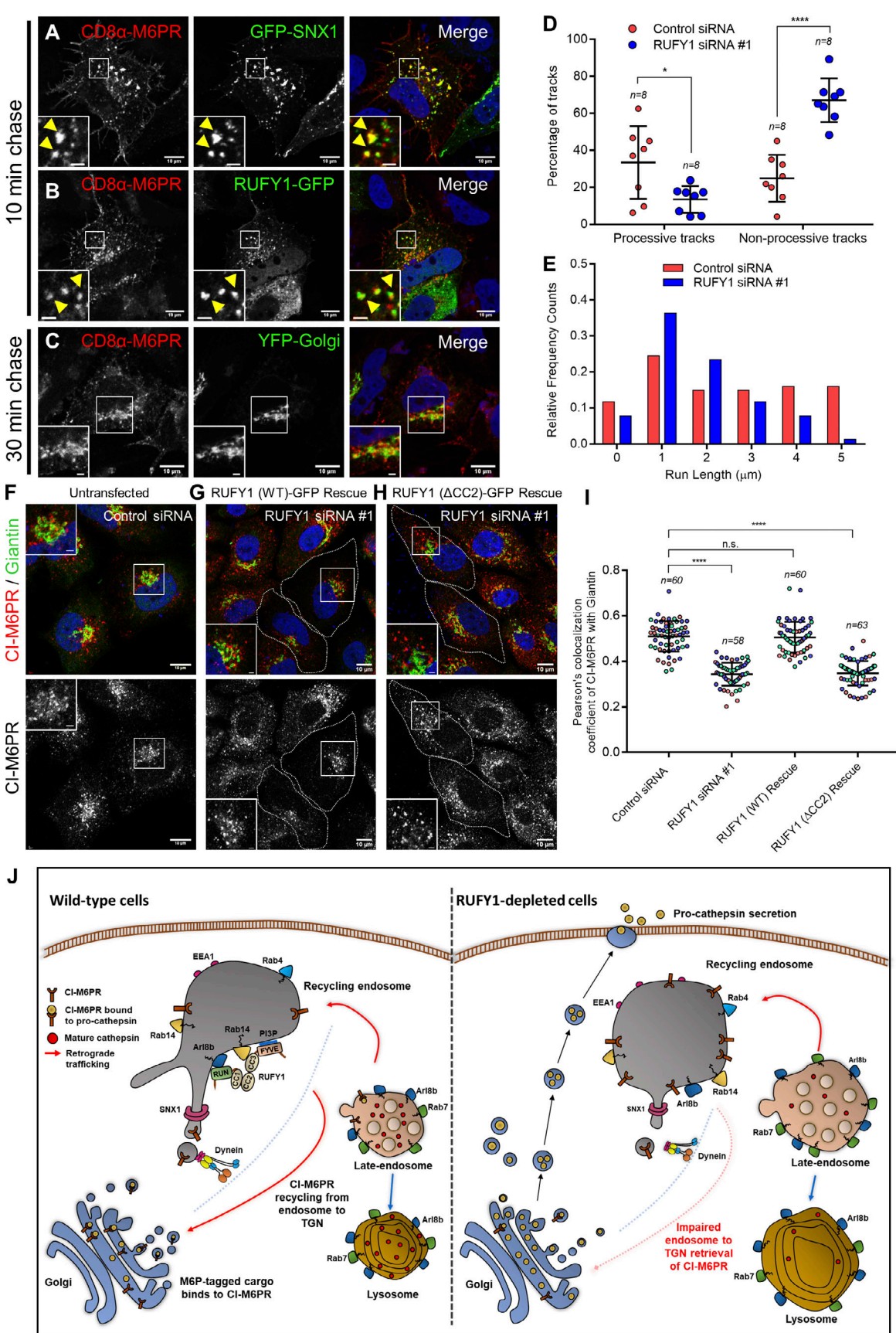

Figure 10. **RUFY1 interaction with the dynein-dynactin complex is required for CI-M6PR retrieval from endosomes to the TGN. (A–C)** Confocal images of HeLa cells co-transfected with CD8α-M6PR and GFP-SNX1 (A) or RUFY1-GFP (B) or YFP-Golgi (C). Surface labeling of CD8α-M6PR was performed as

described in Fig. 6 K and cells were fixed at the indicated time points, followed by immunostaining with secondary antibodies to anti-CD8 antibodies. **(D)** Quantification of the percentage of processive and non-processive tracks of CD8α-M6PR endosomes in HeLa cells treated with control or RUFY1 siRNA. Data represents mean ± SD analyzed from live-cell imaging experiments, and each dot represents one cell (****P < 0.0001; *P < 0.05; unpaired two-tailed t test). **(E)** A histogram displaying the distribution of run lengths obtained from particle tracking of CD8α-M6PR endosomes in HeLa cells treated with either control or RUFY1 siRNA. For both D and E, around 200 tracks were analyzed from 8 cells. **(F–H)** Representative confocal micrographs of HeLa cells treated with the indicated siRNA and transfected with RUFY1 (WT)-GFP (Rescue) or RUFY1 (ΔCC2)-GFP (Rescue) constructs and immunostained for CI-M6PR and Giantin. Cells expressing the rescue construct are marked with a white boundary. Bars: (main) 10 μm; (insets) 2 μm. **(I)** Pearson's colocalization coefficient quantification of CI-M6PR with Giantin in control siRNA and RUFY1 siRNA-treated HeLa cells and in cells transfected with either RUFY1 (WT)-GFP (Rescue) or RUFY1 (ΔCC2)-GFP (Rescue) constructs. The values plotted are the mean ± SD from three independent experiments. Experiments are color-coded, and each dot represents the individual data points from each experiment. The total number of cells analyzed is indicated on the top of each data set (****P < 0.0001; n.s., not significant; unpaired two-tailed t test). **(J)** Proposed role of RUFY1 in regulating cargo sorting from endosomes to the TGN: RUFY1's binding to Arl8b and Rab14 regulates its endosomal localization. RUFY1 recruits the dynein-dynactin complex on these endosomes and mediates CI-M6PR retrieval from the endosome to the TGN. In cells depleted of RUFY1, retrograde motility of CI-M6PR endosomes is reduced, resulting in a defect in sorting of pro-cathepsins to lysosomes. Alternatively, sorting of CI-M6PR into tubular endosomes might be affected, eventually leading to a defect in CI-M6PR retrieval from endosomes to the TGN.

### Radial profile analysis

Images were analyzed for distribution of RUFY1, CI-M6PR and Rab14 in cells using ImageJ plugin "Radial Profile." Normalized integrated intensity signals were obtained by drawing a circle around the entire volume of the cell with the nucleus as center point.

### Immuno-electron microscopy

Cells were fixed in 2% PFA (Electron Microscopy Sciences) and 0.2% Glutaraldehyde (Sigma-Aldrich) in 0.1 M Phosphate Buffer (PB) for 2 h at RT. The fixative was replaced with 1% PFA in 0.1 M PB, and samples were stored until further processing as described before (Slot and Geuze, 2007). Briefly, samples were rinsed in PBS, blocked with 0.15% glycine in PBS, scraped in 1% gelatin in PBS, pelleted, and embedded in 12% gelatin. Small blocks of the pellet were cryoprotected with 2.3 M sucrose, mounted on aluminum pins and plunge frozen in liquid nitrogen. Ultrathin cryosections were cut at –120°C, placed on copper carrier grids, thawed, and immunolabeled. Sections were incubated with blocking buffer containing fish skin gelatin (Sigma-Aldrich) and acetylated bovine serum albumin (BSA, Aurion) and immunolabeled with biotinylated mouse anti-LAMP1 CD107a antibody (555798; BD Pharmingen) at 1:150 dilution, followed by rabbit anti-mouse antibody (Rockland; 610-4120) at 1:250 dilution. Subsequently, sections were incubated with Protein A-conjugated to 10 nm gold particles (Cell Microscopy Core, University Medical Center Utrecht, The Netherlands), stained with uranyl acetate followed by a methylcellulose-uranyl acetate mixture, and examined in a Tecnai12 (FEI, ThermoFisher Scientific) transmission electron microscope. Quantification of the lysosomal area was done by ImageJ software using the freehand selection tool.

### Lysate preparation and immunoblotting

For preparing lysates, cells were collected by trypsinization followed by two washes with DPBS. Lysis was done in ice-cold RIPA buffer (10 mM Tris-Cl, pH 8.0, 1% Triton X-100, 0.1% SDS, 140 mM NaCl, 0.1% sodium deoxycholate, 1 mM EDTA, 0.5 mM EGTA) containing protease inhibitor cocktail (Sigma-Aldrich) for 30 min on ice. The lysates were then centrifuged at 16,000 × g for 10 min, and the supernatant was collected for protein estimation by the bicinchoninic acid assay (Sigma-Aldrich). Protein samples were prepared by boiling samples with 4X sample loading buffer and loaded onto SDS-PAGE for further analysis by immunoblotting. Briefly, protein samples separated on SDS-PAGE were transferred onto polyvinylidene fluoride (PVDF) membranes (Bio-Rad), followed by overnight blocking with 10% skim milk (BD Difco) prepared in 1X PBS containing 0.05% Tween 20 (Sigma-Aldrich). After washing, the blot was incubated with the primary antibody solution prepared in 0.05% PBS-Tween 20 for 2 h at RT. The membranes were washed for 10 min thrice with 0.05% PBS-Tween 20 and further incubated with HRP-conjugated secondary antibody solution prepared in 0.05% PBS-Tween 20 for 1 h at RT. After the secondary antibody step, membranes were washed thrice for 10 min with 0.3% PBS-Tween 20. Blots were developed using a chemiluminescence-based method (ECL Prime Western Blotting System, Cytiva) using x-ray films (Carestream). ImageJ software was used to perform the densitometry analysis on immunoblots.

### Co-immunoprecipitation (Co-IP) assay

HEK293T or HeLa cells were transfected with desired mammalian expression construct followed by lysis with ice-cold TAP lysis buffer (20 mM Tris, pH 8.0, 150 mM NaCl, 0.5% NP-40, 1 mM MgCl$_2$, 1 mM Na$_3$VO$_4$, 1 mM NaF, 1 mM PMSF, and protease inhibitor cocktail) on rotation (Hula Mixer, Thermo Scientific) for 30 min at 4°C. The cell lysate was centrifuged at 13,000 rpm for 10 min at 4°C, and the post-nuclear supernatant (PNS) was collected. The PNS was incubated with indicated antibody conjugated-agarose beads at 4°C rotation for 3 h, followed by four washes with TAP wash buffer (20 mM Tris, pH 8.0, 150 mM NaCl, 0.1% NP-40, 1 mM MgCl$_2$, 1 mM Na$_3$VO$_4$, 1 mM NaF, and 1 mM PMSF). For endogenous co-IP experiments, desired antibodies were first bound with protein A/G beads (Invitrogen) overnight at 4°C on rotation followed by incubation with cell lysates for 8–10 h and finally washing as mentioned above. Protein complexes are eluted by boiling the beads in 2X sample loading buffer at 100°C for 10 min. The samples were then loaded on SDS-PAGE for Western blotting and densitometric analysis as described above.

### Recombinant protein purification and purified protein interaction assay

All bacterial protein expression vectors encoding GST-, His-, or MBP-tagged proteins were transformed into E. coli strain BL21 (DE3; Invitrogen) except for GST-RUFY1 (WT), which was

transformed into Rosetta (DE3) strain. Primary culture was set up by inoculating a single transformed colony in Luria Bertani (LB) broth (Difco) containing appropriate antibiotics followed by incubation for 12 h at 37°C. The secondary culture was set up by inoculating 1% of primary culture in Super Broth media (Hi-media) containing desired antibiotics as described above, and the culture was allowed to grow aerobically at 37°C until the $OD_{600}$ reaches ~0.4. Protein expression was induced by adding 0.5 mM IPTG (Sigma-Aldrich) in secondary culture and incubated overnight at 18°C. Following incubation, the induced culture was pelleted at 4,000 rpm for 15 min and washed with 1X PBS.

The pelleted culture was resuspended in sonication buffer (20 mM Tris and 150 mM NaCl, pH 8.0 [GST-, MBP-tagged proteins]; 50 mM Tris and 150 mM NaCl, pH 8.0 [His-tagged proteins]) containing protease inhibitor tablet (Roche) and 1 mM PMSF (Sigma-Aldrich). Bacterial cells were lysed by sonication, followed by centrifugation at 12,000 rpm for 15 min at 4°C. The clear supernatants were incubated with glutathione-conjugated-agarose resin (G Biosciences), cobalt resin (Thermo Fisher Scientific) or amylose resin (New England Biolabs) on rotation for 2–3 h at 4°C to facilitate the binding of proteins. The beads were washed a minimum of six times with wash buffer (20 mM Tris and 150 mM NaCl, pH 8.0 [GST-, MBP-tagged proteins]; 50 mM Tris, pH 8.0, 300 mM NaCl, 10 mM Imidazole, pH 8.0 [His-tagged proteins]) to remove non-specific proteins. For the purified protein interaction assay, His-tagged proteins were eluted from the cobalt resin using elution buffer (50 mM Tris, pH 8.0, 300 mM NaCl, 250 mM Imidazole) and GST-tagged proteins were eluted from glutathione-conjugated-agarose resin using elution buffer (50 mM Tris and 100 mM NaCl, pH 8.0, 10 mM glutathione). Eluted proteins were further concentrated using Millipore Amicon Ultra Centrifugal Filter Unit (Millipore).

In the purified protein interaction assay of GST and GST-RUFY1 (RUN) with His-Arl8b (WT or mutants), GST-fusion proteins (5 µg) immobilized on glutathione-conjugated-agarose beads were incubated with 2.5 µg of His-tagged proteins in TAP lysis buffer (20 mM Tris, pH 8.0, 150 mM NaCl, 0.5% NP-40, 1 mM $MgCl_2$, 1 mM $Na_3VO_4$, 1 mM NaF, 1 mM PMSF, and protease inhibitor cocktail) for 2 h at 4°C rotation, followed by two washes with TAP lysis buffer containing 0.3% NP-40. Protein complexes were eluted by boiling samples in 2X sample loading buffer at 100°C for 10 min and loaded onto SDS-PAGE gel followed by immunoblotting as described earlier.

For protein-protein interaction assay of MBP and MBP-tagged Rab14 with GST-RUFY1 (WT), MBP-fusion proteins (5 µg) immobilized on amylose beads were loaded with 5 mM GTP (Jena Biosciences) or GDP (Sigma-Aldrich) at 4°C on rotation in TAP lysis buffer containing 10 mM $MgCl_2$ and protease inhibitor for 30 min. After loading with GTP/GDP, MBP-fusion proteins were incubated with 3 µg of GST-RUFY1(WT) protein in the absence or presence of increasing amounts of His-Arl8b or His-Rab7 (0.25, 0.5 and 0.75 µg) in the same buffer at 4°C for 1 h followed by two washes with TAP lysis buffer containing 0.3% NP-40. Protein complexes were eluted by boiling samples in 2X sample loading buffer at 100°C for 10 min and loaded onto SDS-PAGE gel followed by immunoblotting with the indicated antibodies.

## GST pull-down assay

For GST pull-down assays, HEK293T cells were transfected with indicated constructs and lysed in ice-cold TAP lysis buffer (20 mM Tris, pH 8.0, 150 mM NaCl, 0.5% NP-40, 1 mM $MgCl_2$, 1 mM $Na_3VO_4$, 1 mM NaF, 1 mM PMSF, and protease inhibitor cocktail). The lysates were prepared as described above and were incubated with GST or GST-fusion proteins immobilized on glutathione-conjugated-agarose resin for 2 h at 4°C on rotation. The resin was washed three times with the wash buffer (20 mM Tris, pH 8.0, 150 mM NaCl, 0.3% NP-40, 1 mM $MgCl_2$, 1 mM NaF, 1 mM $Na_3VO_4$). Protein complexes were eluted by boiling samples in 2X sample loading buffer at 100°C for 10 min and loaded onto SDS-PAGE gel followed by immunoblotting as described earlier.

For GST pull-down assays with GTP/GDP loading, GST-Arl8b fusion proteins immobilized on glutathione-conjugated-agarose beads were first loaded with 20 mM GTP or GDP in TAP lysis buffer containing 10 mM $MgCl_2$ for 30 min at 4°C on rotation. After loading with GTP/GDP, beads were pelleted down followed by incubation with HEK293T cell lysates transfected with indicated constructs for 2 h at 4°C on rotation followed by three washes with wash buffer (20 mM Tris, pH 8.0, 150 mM NaCl, 0.3% NP-40, 1 mM $MgCl_2$, 1 mM NaF, 1 mM $Na_3VO_4$).

## Tandem affinity purification (TAP) and mass spectrometry analysis

To study the interactome of RUFY1, a TAP assay was performed using the Interplay TAP Purification Kit (Agilent Technologies). In brief, ~50 million HEK293T cells stably expressing TAP-RUFY1 were lysed according to the protocol provided by the manufacturer and subjected to TAP protocol using streptavidin and calmodulin resins. Lysates were first incubated with streptavidin resin to allow purification using the SBP tag. Streptavidin resin was washed to remove any unbound proteins, and further bound protein complexes were eluted with elution buffer containing biotin. Streptavidin eluate was then applied to calmodulin resin at 4°C on rotation for the second round of purification using the CBP tag. Purified protein complexes bound on calmodulin resin were given three washes to remove contaminants and eluted by heating samples in the 4X sample loading buffer. Eluted samples were resolved on 10% SDS-PAGE and subjected to mass spectrometry analysis at the Taplin MS Facility (Harvard Medical School, Boston, MA). The proteins interacting with RUFY1 were filtered using CRAPOME tools (available at https://reprint-apms.org/) and are listed in Table S1.

## Membrane-cytosol fractionation

Membrane-cytosol fractionation was performed as described previously (Mari et al., 2001) with minor modifications. Following siRNA treatment, HeLa cells (one confluent 100 mm cell culture dish) were homogenized with Dounce homogenizer in 1 ml of homogenization buffer (Tris-Cl, pH 7.4, 1 mM EDTA, 250 mM Sucrose and 125 mM KCl) containing protease inhibitor cocktail (Sigma-Aldrich). Homogenates were centrifuged at 800 × $g$ for 10 min at 4°C. The cell pellet was discarded, and the PNS was ultracentrifuged at 108,000 × $g$ for 1 h at 4°C (Hitachi)

to separate cytosol and total membrane pellet. The total membrane pellet was incubated with 1 ml of 1% Triton X-100 prepared in homogenization buffer for 1 h at 4°C with rotation. Triton X-100 suspension was ultracentrifuged at 100,000 × *g* for 1 h to separate soluble and insoluble fractions. Cytosol and soluble fractions were TCA (Sigma-Aldrich) precipitated followed by two washes with acetone. The precipitated cytosol and Triton X-100 soluble fraction along with Triton X-100 insoluble pellet were resuspended in 2X sample loading buffer by heating at 100°C for 10 min followed by SDS-PAGE and immunoblotting with indicated antibodies as described above.

## Yeast two-hybrid assay

The yeast two-hybrid assay was performed as described previously (Sharma et al., 2021). Briefly, bait proteins in fusion with Gal4 DNA-binding domain (cloned in pGBKT7 vector) and prey proteins in fusion with Gal4 activation domain (cloned in pGADT7 vector) were co-transformed in Y2H Gold strain of *S. cerevisiae* (Takara Bio, Inc.). The transformants were then plated onto double-dropout plates lacking tryptophan and leucine (–Leu/–Trp) and allowed to grow at 30°C for 3 d to select yeast cells containing both bait and prey plasmids. The co-transformant yeast colonies obtained after incubation were dissolved in sterile water and OD600 was measured. For each bait and prey combination, OD600 of yeast cell suspension was normalized to 0.5 and fivefold serial dilutions were prepared in sterile water. Each dilution along with starting undiluted suspension was spotted on double-dropout (–Leu/–Trp) and quadruple-dropout plates lacking leucine, tryptophan, histidine, and adenine (-Leu/-Trp/-His/-Ade) to assess the interaction.

## Measurement of lysosome pH

To measure the lysosome's pH, LysoSensor Yellow/Blue DND-160 (Invitrogen) was used as described previously (Ma et al., 2017). To determine the pH of lysosomes, a pH calibration curve was first generated for HeLa cells. Briefly, cells were trypsinized and incubated with LysoSensor Yellow/Blue DND-160 (final concentration of 2 μM) in phenol red-free DMEM (Gibco) for 3 min at 37°C. Following incubation, cells were washed twice with DPBS and further incubated with a set of isotonic pH calibration buffers (143 mM KCl, 5 mM Glucose, 1 mM $MgCl_2$, 1 mM $CaCl_2$, 20 mM MES, 10 μM Nigericin, and 5 μM monensin) with pH ranging from 4 to 6. Each pH calibration buffer was pre-adjusted to its final pH value using 1 N NaOH or 1 N HCl. Cells (10,000/well) were transferred into a black 96-well plate (Thermo Fisher Scientific), and fluorescence readings were recorded simultaneously for two excitation wavelengths (340 and 380 nm) at 37°C. Finally, the ratio of fluorescence intensity of emission at 340–380 nm against respective pH values was plotted to generate the pH calibration curve. Using this curve, the pH value of lysosomes for HeLa cells treated with indicated siRNA treatment was extrapolated.

## Lysotracker red uptake analysis by flow cytometry

For Lysotracker uptake assay, HeLa transfected with indicated siRNA were incubated with Lysotracker Red DND-99 (Invitrogen) dye (final concentration of 100 nM) in phenol red-free

DMEM (Gibco) for 2 h at 37°C. Post-incubation period, Lysotracker-containing media was removed, and cells were trypsinized, washed and resuspended in ice-cold 1X PBS and analyzed by flow cytometry. Sample acquisition was performed with BD FACS Aria Fusion Cytometer using BD FACS Diva software version 8.0.1 (BD Biosciences). Data analysis was done using BD FlowJo version 10.0.1.

## BODIPY-BSA trafficking assay

For BODIPY-BSA trafficking, HeLa cells treated with indicated siRNA were loaded with BODIPY-BSA (BODIPY FL-conjugated, BioVision) at a final concentration of 20 μg/ml in phenol red-free DMEM and incubated for 7 h at 37°C. Post-incubation period, media was removed, and cells were trypsinized, washed and resuspended in ice-cold 1X PBS and analyzed by flow cytometry. Sample acquisition was performed with BD FACS Aria Fusion Cytometer using BD FACS Diva software version 8.0.1 (BD Biosciences).

## EGFR degradation assay

After 60 h of indicated siRNA treatment, cells were starved in serum-free DMEM (Gibco) for 1 h, following which unlabeled EGF (Invitrogen) pulse (at a final concentration of 100 ng/ml) was given for 10 min at 37°C. Further, cells were chased in serum containing complete media at 37°C for the indicated time points, after which the cells were fixed with 4% PFA and immunostained with anti-EGFR antibody (Invitrogen). Single plane confocal images were acquired using an LSM-710 confocal microscope (ZEISS), and EGFR intensity was measured using ImageJ software. The Corrected Total Cell Fluorescence (CTCF) values were calculated using the formula: CTCF = Integrated density—(area of selected cell X mean fluorescence of background).

## CD8α-M6PR trafficking assay

CD8α-M6PR trafficking assay was performed as described previously (Seaman, 2004; Shi et al., 2018) with minor modifications. Briefly, HeLa cells grown on glass coverslips (VWR) were transfected with CD8α-CI-M6PR expressing plasmid. After 12 h of transfection, cells were incubated with the complex of the anti-CD8α primary antibody with Alexa Fluor 568/488-conjugated secondary antibody for 30 min on ice followed by two washes with citric acid buffer, pH 4.5 (0.1 M citric acid anhydrous and 0.1 M tri-sodium citrate dihydrate). After washing, cells were incubated with pre-warmed media at 37°C for indicated time points, followed by fixation with 4% PFA in PHEM buffer for 15 min at 4°C. For siRNA studies, transfection of CD8α-CI-M6PR construct was done after 60 h of siRNA treatment followed by the trafficking protocol mentioned above.

## RUSH assay

The RUSH assay was performed as previously described (Niu et al., 2019). HeLa cells grown on a glass-bottom live-cell imaging dish were subjected to indicated siRNA treatment for 60 h followed by co-transfection of Str-KDEL-IRES-SBP-mCherry-CTSZ and LAMP1-GFP plasmids. After 12 h of transfection, media of the cells was replaced by phenol red-free

DMEM containing 10% FBS, and single plane confocal images of live cells were acquired at 37°C using LSM-710 confocal microscope (ZEISS) before and after the addition of biotin (final concentration 40 µM, Sigma-Aldrich) for the indicated time points.

## Pro-cathepsin D secretion assay

Pro-cathepsin D secretion in media was assayed as previously described (Hao et al., 2013). HeLa cells were grown in complete culture media and transfected with the desired siRNA. After 60 h of siRNA treatment, media was replaced with reduced serum media, Opti-MEM (Gibco), and incubated for 16 h. Opti-MEM media was collected and centrifuged at 1,600 × g for 10 min to remove cell debris. Clarified supernatant obtained was treated with sodium deoxycholate (final concentration v/v, 0.02%) on ice for 30 min, followed by precipitation of proteins using ice-cold TCA (Sigma-Aldrich) at a final concentration of 15% v/v overnight at 4°C on rotation (Hula Mixer). Precipitated proteins were pelleted by centrifugation at 13,000 rpm for 15 min, followed by washes with ice-cold acetone. Supernatants were discarded, and pellets were dried and resuspended by heating them with 2X sample loading buffer for 15 min at 100°C followed by analysis by Western blotting as described previously.

## FKBP-FRB heterodimerization assay

HeLa cells grown on glass coverslips were co-transfected with 2xFKBP-GFP vector or 2xFKBP-GFP-RUFY1 (WT/mutants) with FRB-Tom-70p for 12 h, followed by rapamycin treatment (final concentration, 100 nM; Sigma-Aldrich) of 2 h. Cells were fixed with 4% PFA in PHEM buffer for 10 min at RT, and immunostaining was performed as described earlier. For visualization of mitochondria, cells were stained with anti-Tom20 antibody. Single-plane confocal images were acquired using LSM710 Confocal microscope (ZEISS), followed by the analysis of mitochondrial distribution using ImageJ. Cells expressing both FRB-Tom70p with 2x-FKBP-GFP alone or 2x-FKBP-GFP-RUFY1 (WT/mutants) were selected using a freehand selection tool. The mitochondrial intensity was quantified by drawing circles at an increment of 5 µm starting from nuclear rim to cell periphery, followed by normalization of intensities at every distance by total cell intensity. Normalized Tom20 Intensity values obtained were plotted as a relative function of distance from the nucleus.

## CD8α-CI-M6PR particle motility assay

HeLa cells treated with indicated siRNA were transfected with CD8α-CI-M6PR expressing plasmid. After 12 h post-transfection, cells were incubated on ice with a complex of anti-CD8 primary antibody bound to Alexa Fluor-488 conjugated secondary antibody for 30 min for surface labeling of the receptor. The cells were shifted to warm phenol red-free DMEM for 10 min, followed by live-cell imaging as described previously. Post data acquisition, the time-lapse video files were analyzed by a semi-automated particle tracking software (Mohan et al., 2019). The particle tracking analysis tool provides 2D trajectories (x and y values for each time point or frame) for moving vesicles. The particles moving for a minimum of 25 frames were selected for analysis. A custom-written MATLAB script was employed to determine the active and passive phases from the 2D particle trajectories (Mohan et al., 2019; Verdeny-Vilanova et al., 2017). Further, the trajectories were analyzed for the time dependence of the mean square displacement (MSD) to calculate the diffusion coefficient (α; Mohan et al., 2019; Ruthardt et al., 2011). The power-law exponents calculated from MSD were used to categorize the processive and diffusive behavior of the tracks (Martin et al., 2002; Mohan et al., 2019). The exponents were calculated by fitting the MSD curves to the power-law log $[MSD(\Delta t)] = \alpha \log(\Delta t) + C$. Tracks with $\alpha > 1.45$ were categorized as processive. Tracks with $1 \le \alpha \le 1.45$ were categorized as diffusive, and for $\alpha < 1$, tracks were categorized as non-processive. The run-length was also computed using the same custom-written program after the active/passive categorization of the trajectories as previously described (Mohan et al., 2019).

## Statistical analysis

All graphs report the mean ± SD, unless otherwise specified. P values were calculated using a two-tailed unpaired Student's t test from three independent biological replicates. Data distribution was assumed to be normal, but this was not formally tested.

## Online supplemental material

Fig. S1 shows RUFY1 interaction with Arl8b is specific and both RUFY1 and RUFY3 interact with Arl8b and the subcellular localization of RUFY1 to early and recycling endosomes. Fig. S2 shows that RUFY1 and Arl8b colocalize on endosomes positive for early and recycling markers, but not on late endocytic compartments, and Arl8b regulates RUFY1 endosomal localization. Fig. S3 shows that Rab14 is essential and sufficient to target RUFY1 recruitment to membranes. Fig. S4 shows that RUFY1 depletion leads to lysosome enlargement and reduced cargo degradative ability and also defects in endosome to TGN retrieval of CI-M6PR. Fig. S5 shows that RUFY1 interaction with dynein intermediate chain is specific and dynein is required for normal perinuclear positioning of RUFY1 and Rab14 endosomes. Video 1 shows the dynamics of RUFY1 and Rab14 in control and Arl8b-depleted cells. Video 2 shows the dynamics of Rabip4 (RUFY1 shorter isoform) and SNX1. Videos 3, 4, and 5 show examples of processive, non-processive, and diffusive tracks of CD8α-M6PR-positive endosomes, respectively. Table S1 provides a list of RUFY1 interacting proteins identified from HEK293T cell lysates by TAP-pulldown assay followed by mass spectrometric analysis. Table S2 provides a list of DNA constructs and primer sequences used in this study. Table S3 provides a list of antibodies used in this study.

## Data availability

All data are contained within the manuscript.

## Acknowledgments

The authors acknowledge Dr. Nitin Mohan (IIT Kanpur) for sharing the particle tracking software and MATLAB script and

for instructions on the methodology, and for providing inputs on particle tracking data analysis. The authors acknowledge Prateek Arora (IISER Mohali FACS Facility) for technical help in flow cytometry and Ross Tomaino (Taplin MS Facility, Harvard Medical School) for mass spectrometry analysis.

S. Rawat and D. Khatter acknowledge fellowship support from University Grants Commission, and D. Chatterjee, R. Marwaha, G. Charak, S. Sharma acknowledges fellowship support from IISER Mohali, G. Kumar acknowledges fellowship support from Council of Scientific and Industrial Research. This work was supported by the Department of Biotechnology (DBT)/Wellcome Trust India Alliance Senior Fellowship (IA/S/19/1/504270) to M. Sharma and Intermediate Fellowship (IA/I/14/2/501543) to A. Tuli. N. Liv is supported by a ZonMW TOP grant (40-00812-98-16006 to J. Klumperman). The electron microscopy within this work is part of the research program National Roadmap for Large-Scale Research Infrastructure (NEMI; project number 184.034.014 to J. Klumperman), financed by the Dutch Research Council (NWO). The funders had no role in study design, data collection, interpretation, or the decision to submit the work for publication. Open Access funding provided by IISER Mohali.

The authors declare no competing financial interests.

Author contributions: S. Rawat, A. Tuli, and M. Sharma conceived and designed the project. S. Rawat performed the majority of the experiments, analyzed the results, and prepared the figures. D. Chatterjee had performed the protein-protein interaction experiments and yeast two-hybrid assays. R. Marwaha had performed the lysosome size quantification and protein-protein interaction assays. G. Charak generated the TAP-RUFY1 expressing cells and performed the TAP-pulldown experiments. G. Kumar and S. Sharma contributed to carrying out protein-protein interaction assays. S. Shaw performed the cathepsin secretion assays and standardized and assisted in the RUSH assays. D. Khatter had standardized the Arl8b and RUFY1 colocalization and provided critical molecular biology reagents. C. de Heus had performed the immuno-EM experiments, and C. de Heus, N. Liv and J. Klumperman had analyzed the EM data and provided inputs on manuscript text. S. Rawat and M. Sharma wrote the manuscript.

Submitted: 1 August 2021

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

# Supplemental material

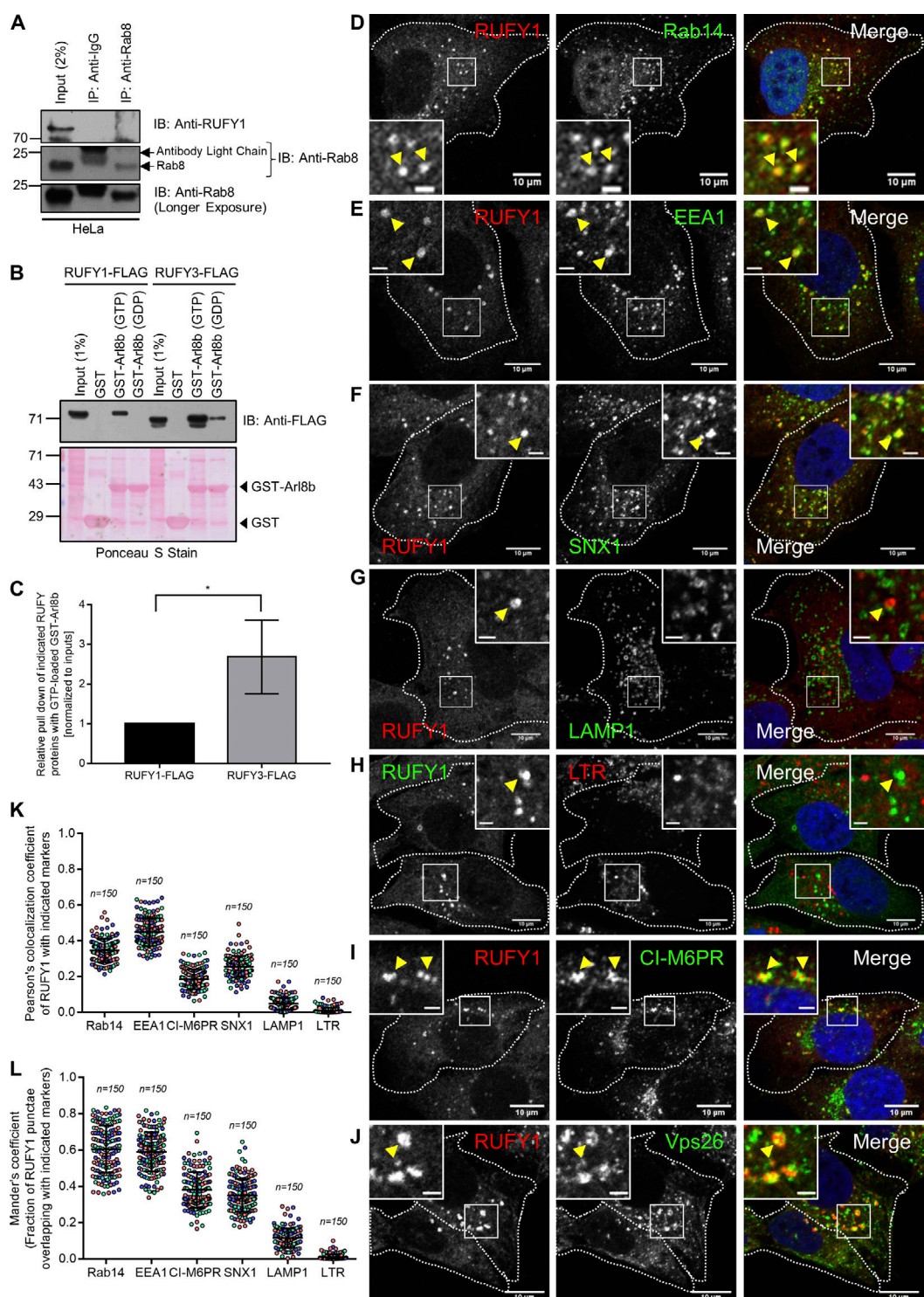

Figure S1. **RUFY1 and RUFY3 interact with Arl8b, and RUFY1 localize to compartments positive for early/recycling endocytic markers. (A)** HeLa cell lysates were immunoprecipitated with anti-Rab8 antibodies bound to Protein A/G beads. The precipitates were immunoblotted (IB) with the indicated antibodies. **(B)** Recombinant GST and GST-Arl8b proteins immobilized on glutathione-coated-agarose beads were loaded with either GTP or GDP and then incubated with HEK293T cell lysates expressing RUFY1-FLAG or RUFY3-FLAG. The precipitates were IB with an anti-FLAG antibody and Ponceau S staining was done to visualize the purified proteins. **(C)** Densitometric analysis of RUFY1 and RUFY3 pulldown (normalized to input signals) using GTP-loaded GST-Arl8b. The values plotted are the mean ± SD from three independent experiments (*P < 0.05; unpaired two-tailed $t$ test). **(D–J)** Representative confocal micrographs of HeLa cells immunostained for endogenous RUFY1 and various endocytic markers (D) Rab14, (E) EEA1, (F) SNX1, (G) LAMP1, (H) Lysotracker Red (LTR), (I) CI-M6PR and (J) Vps26. In the inset, the arrowhead marks the colocalized pixels. Bars: (main) 10 µm; (insets) 2 µm. **(K and L)** Pearson's and Mander's colocalization coefficient quantification of endogenous RUFY1 with various indicated markers. The values plotted are the mean ± SD from three independent experiments. Experiments are color-coded, and each dot represents the individual data points from each experiment. The total number of cells analyzed is indicated on the top of each data set. Source data are available for this figure: SourceData FS1.

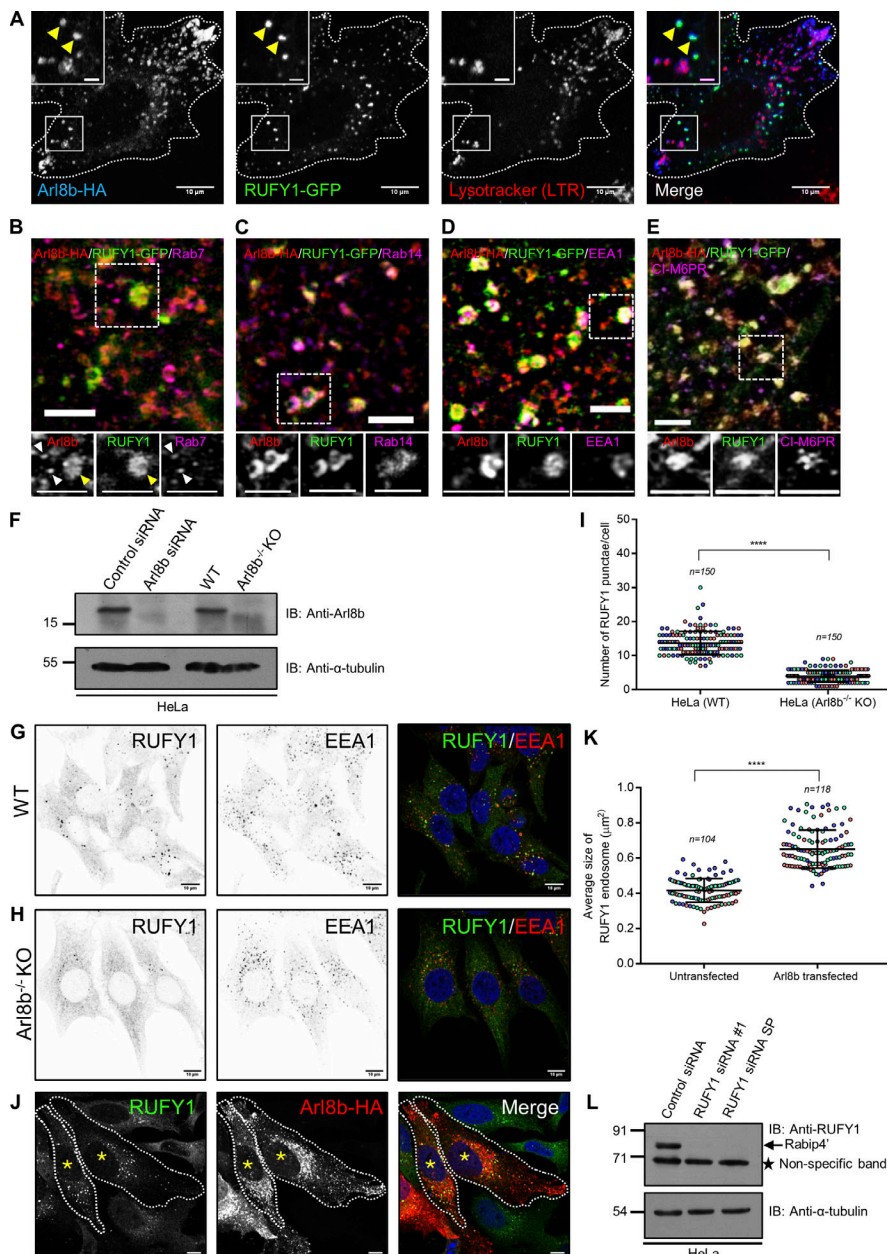

Figure S2.  **RUFY1 and Arl8b colocalize on early/recycling endosomes and Arl8b regulates RUFY1 membrane localization. (A)** Representative confocal micrograph of HeLa cells co-transfected with Arl8b-HA and RUFY1-GFP and incubated with Lysotracker Red (100 nM) for 2 h, followed by PFA fixation and immunostaining with anti-HA antibody. Bars: (main) 10 μm; (insets) 2 μm. **(B–E)** Structured illumination microscopy (SIM) of HeLa cells co-transfected with Arl8b-HA and RUFY1-GFP and immunostained for Arl8b using anti-HA antibodies and other endocytic markers as indicated. For B–E, arrowheads in the insets mark the colocalized pixels. The white arrowheads in the inset of Fig. S2 B mark Arl8b and Rab7 colocalized punctae, while the yellow arrowhead marks Arl8b and RUFY1 colocalized puncta. Bars: (main) 2 μm; (insets) 2 μm. **(F)** Control- and Arl8b-siRNA-treated HeLa cell lysates or lysates of WT- and Arl8b$^{-/-}$ knockout (KO)-HeLa cells were immunoblotted (IB) with anti-Arl8b antibody for assessing the knockdown efficiency. α-tubulin was used as the loading control. **(G and H)** Confocal images of wild-type (WT) and Arl8b$^{-/-}$ KO HeLa cells immunostained for endogenous RUFY1 and EEA1. Single-channel images of RUFY1 and EEA1 are represented as inverted images to facilitate understanding. Bars: 10 μm. **(I)** The graph shows the quantification of the number of RUFY1 punctae in WT and Arl8b$^{-/-}$ KO HeLa cells. The values plotted are the mean ± SD from three independent experiments. Experiments are color-coded, and each dot represents the individual data points from each experiment. The total number of cells analyzed is indicated on the top of each data set (****P < 0.0001; unpaired two-tailed t test). **(J)** Representative confocal micrographs of HeLa cells transfected with Arl8b-HA followed by immunostaining with anti-HA and anti-RUFY1 antibodies. An asterisk denotes the Arl8b-HA-expressing cells. Bars: 10 μm. **(K)** The graph shows the quantification of RUFY1 punctae size in Arl8b-HA transfected and surrounding untransfected cells. The values plotted are the mean ± SD from three independent experiments. Experiments are color-coded, and each dot represents the individual data points from each experiment. The total number of cells analyzed is indicated on the top of each data set (****P < 0.0001; unpaired two-tailed t test). **(L)** HeLa cell lysates from the indicated siRNA treatments were IB with anti-RUFY1 antibody (Santa Cruz Biotechnology [SCBT]; sc-398740) to assess the specificity of the antibody, and α-tubulin was used as the loading control. This antibody from SCBT recognizes the longer isoform of RUFY1 (Rabip4'; ~80 kD; marked by an arrow) but also shows a non-specific band at ~70 kD (marked by an asterisk) whose intensity is not reduced upon RUFY1 knockdown. Source data are available for this figure: SourceData FS2.

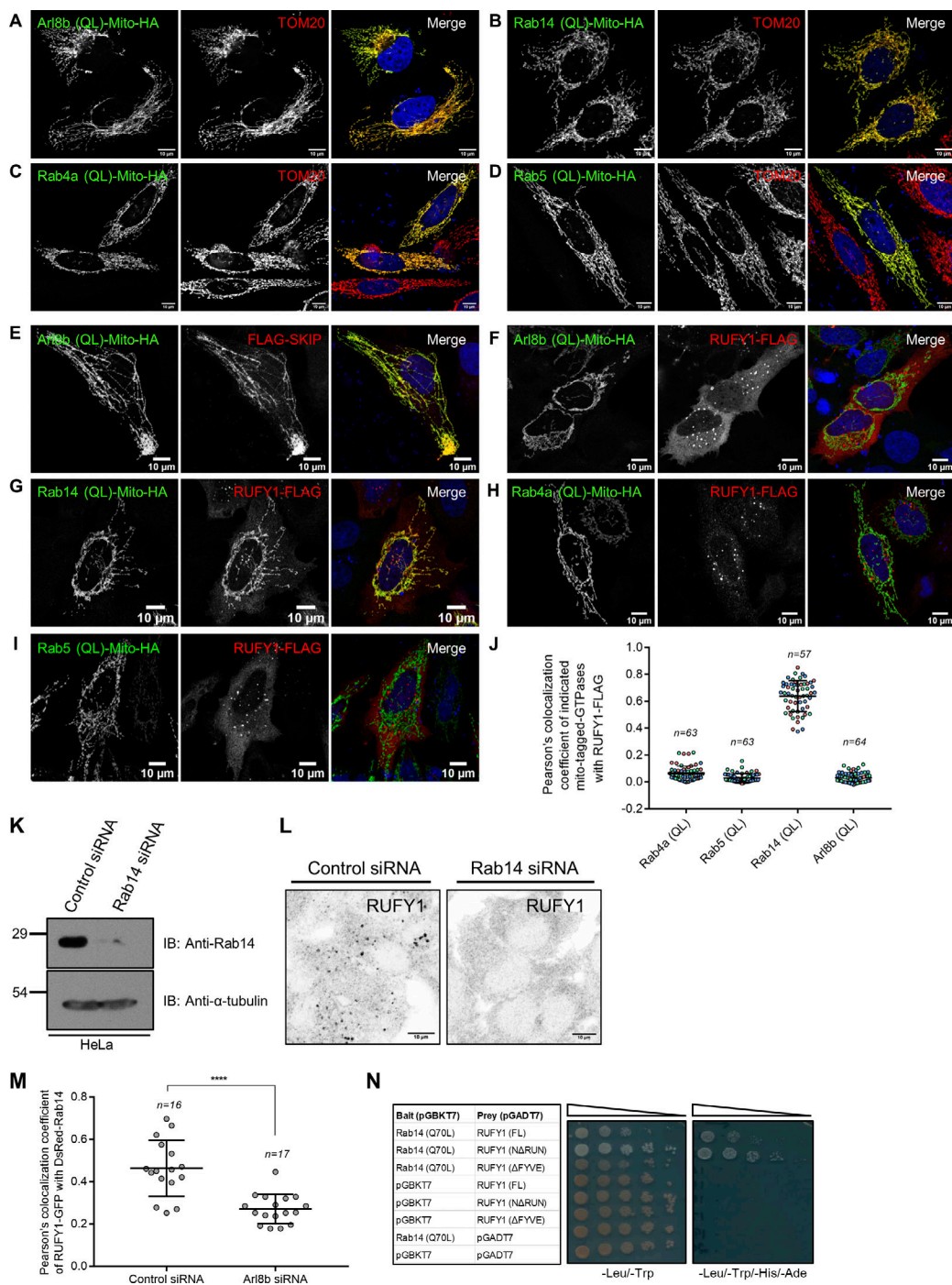

Figure S3. **Rab14, but not Arl8b, determines RUFY1 recruitment to membranes. (A–D)** Confocal micrographs of HeLa cells transfected with C-terminal Mito-HA-tagged GTP-locked versions of small G proteins, Arl8b (Q75L; A), Rab14 (Q70L; B), Rab4a (Q72L; C), and Rab5 (Q79L; D). Cells were fixed and immunostained with anti-HA and anti-Tom20 (mitochondrial marker) antibodies. Bars: 10 μm. **(E–I)** Confocal micrographs of HeLa cells co-transfected with Mito-tagged GTP-locked versions of small G proteins as described above and with either FLAG-SKIP (E) or RUFY1-FLAG (F–I). Cells were fixed and immunostained with anti-HA and anti-FLAG antibodies. Bars: 10 μm. **(J)** Pearson's colocalization coefficient quantification of RUFY1-FLAG and Mito-HA-tagged GTP-locked versions of small G proteins-Rab4a, Rab5, Rab14, and Arl8b. The values plotted are the mean ± SD from three independent experiments. Experiments are color-coded, and each dot represents the individual data points from each experiment. The total number of cells analyzed is indicated on the top of each data set. **(K)** Control- and Rab14-siRNA-treated HeLa cell lysates were immunoblotted (IB) with anti-Rab14 antibody for assessing the knockdown efficiency. α-tubulin was used as the loading control. **(L)** Representative confocal micrographs of HeLa cells treated with either control or Rab14 siRNA followed by immunostaining for endogenous RUFY1. Bars: 10 μm. **(M)** Pearson's colocalization coefficient (PCC) was quantified for RUFY1-GFP and DsRed-tagged-Rab14 signals in either control or Arl8b siRNA-treated HeLa cells. PCC was calculated from a single frame of live-cell imaging videos. The values plotted are the mean ± SD from two independent experiments with 16–17 cells analyzed per treatment (****P < 0.0001; unpaired two-tailed *t* test). **(N)** Yeast two-hybrid assay. The indicated yeast co-transformants were spotted (five-fold serial dilution) on -Leu/-Trp (non-selection) and -Leu/-Trp/-His/-Ade (selection) media plates to confirm viability and interactions, respectively. Source data are available for this figure: SourceData FS3.

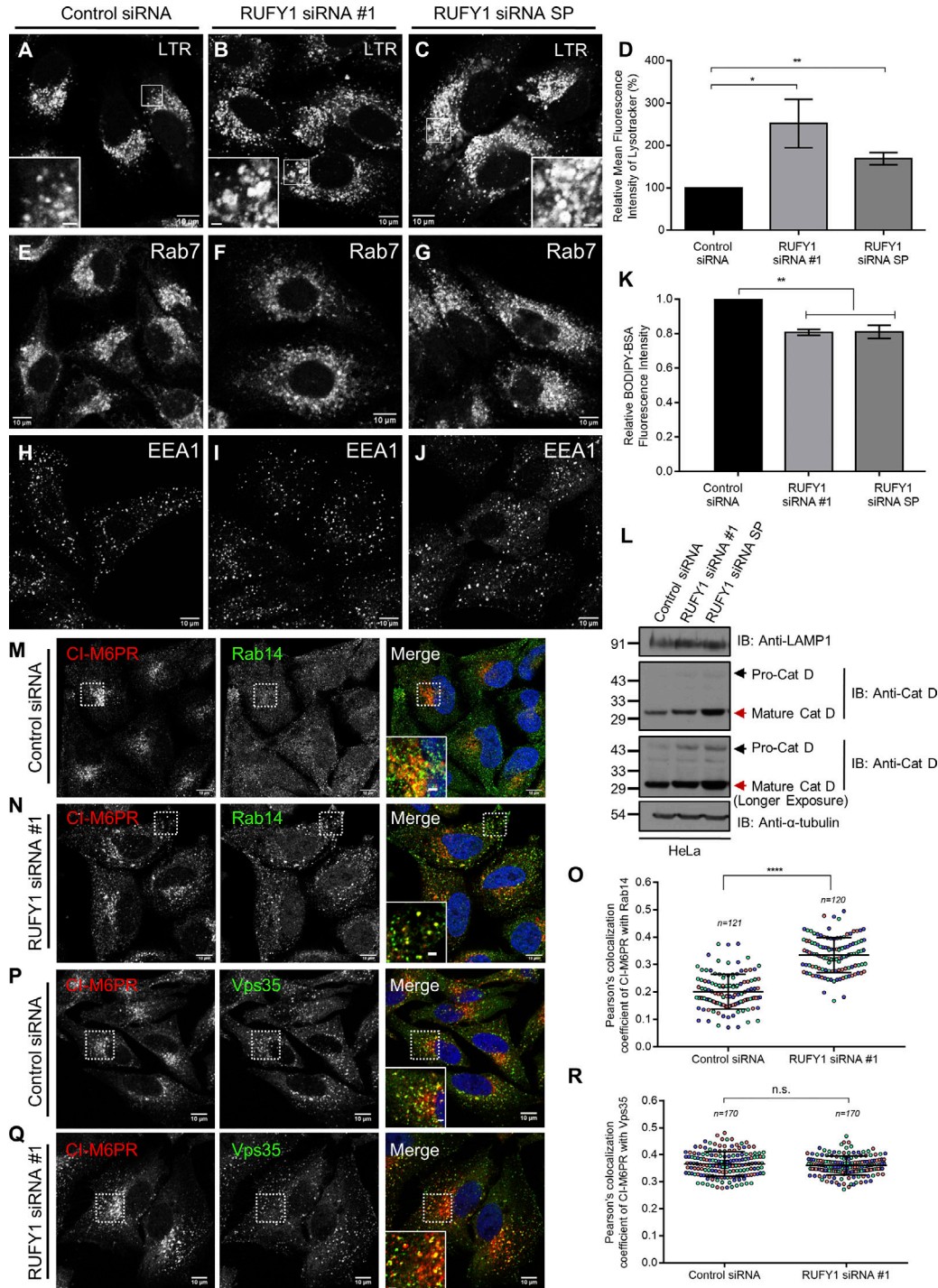

Figure S4. **RUFY1 depleted cells show features of lysosome dysfunction and enhanced colocalization of CI-M6PR and Rab14. (A–C)** Representative confocal images of HeLa cells treated for 60 h with the indicated siRNAs and incubated for 2 h with Lysotracker Red (LTR) before fixation. Bars: (main) 10 μm; (insets) 2 μm. **(D)** Measurement of the fold change in LTR intensity in HeLa cells treated with the indicated siRNAs and analyzed by flow cytometry. The values plotted are the mean ± SD from three independent experiments (**P < 0.01; *P < 0.05; unpaired two-tailed t test). **(E–J)** Confocal micrographs of HeLa cells treated with the indicated siRNAs followed by immunostaining with anti-Rab7 or anti-EEA1 antibodies. Bars: 10 μm. **(K)** HeLa cells were treated with the indicated siRNAs and subjected to BODIPY-BSA uptake for 7 h. The BODIPY-BSA fluorescence was analyzed by flow cytometry. The values plotted are the mean ± SD from three independent experiments (**P < 0.01; unpaired two-tailed t test). **(L)** Western blot analysis of pro-cathepsin D (Cat D), mature Cat D, and LAMP1 levels in either control or RUFY1 siRNA-treated HeLa cells. α-tubulin was used as a loading control. **(M, N, P, and Q)** Representative confocal images of HeLa cells treated with control or RUFY1 siRNA, followed by immunostaining with anti-CI-M6PR and anti-Rab14 antibodies (M and N) or anti-CI-M6PR and anti-Vps35 antibodies (P and Q). Bars: (main) 10 μm; (insets) 2 μm. **(O and R)** The Pearson's colocalization coefficient of CI-M6PR with Rab14 (O) or CI-M6PR and Vps35 (R) was measured in HeLa cells treated with either control or RUFY1 siRNA. The values plotted are the mean ± SD from three independent experiments. Experiments are color-coded, and each dot represents the individual data points from each experiment. The total number of cells analyzed is indicated on the top of each data set (****P < 0.0001; n.s., not significant; unpaired two-tailed t test). Source data are available for this figure: SourceData FS4.

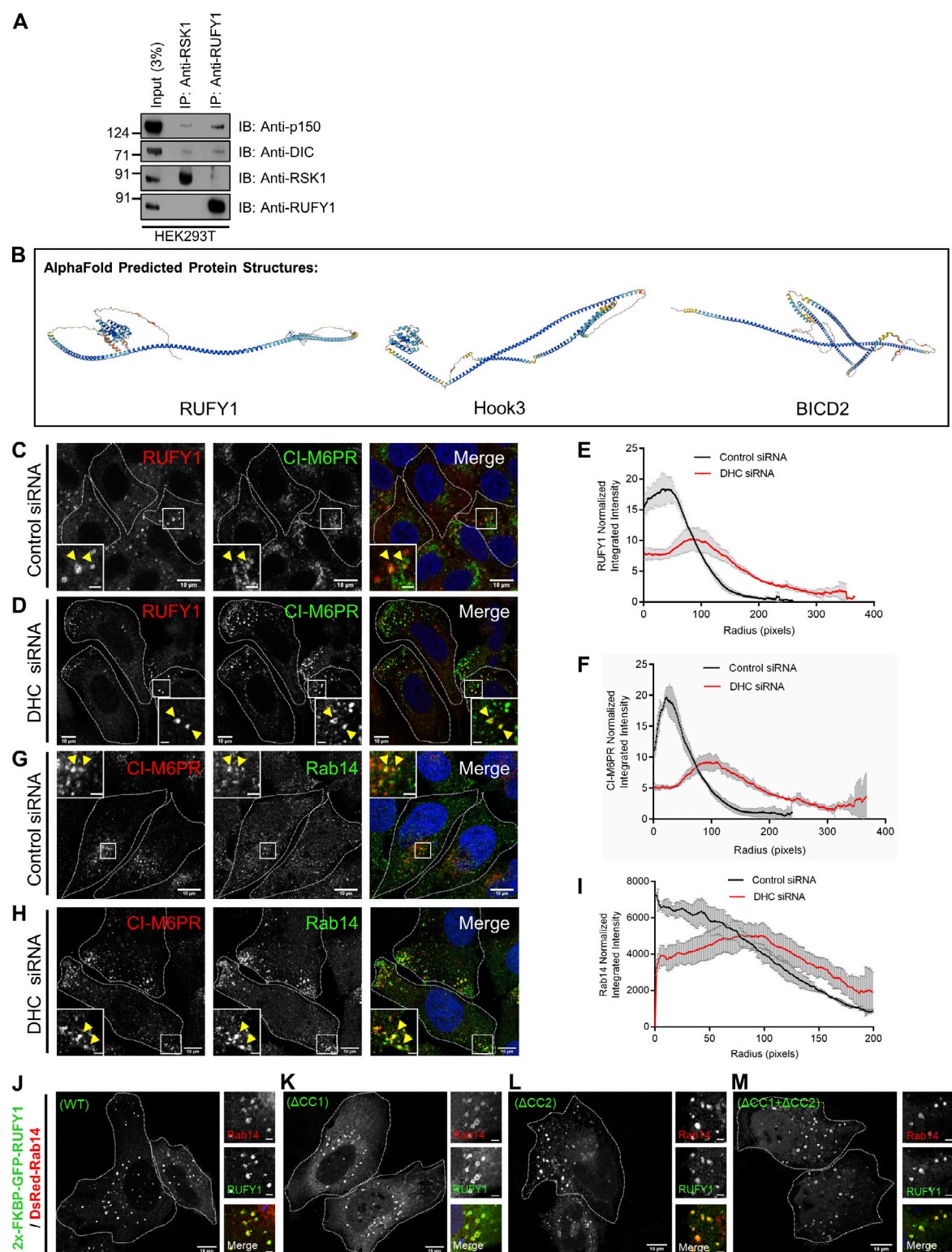

Figure S5.  **Perinuclear positioning of RUFY1, CI-M6PR, and Rab14 is dependent on dynein. (A)** HEK293T cell lysates were immunoprecipitated with either anti-RSK1 (used as a control) or anti-RUFY1 antibodies bound to Protein A/G beads. The precipitates were immunoblotted (IB) with the indicated antibodies. **(B)** AlphaFold prediction of structures for human proteins-RUFY1, HOOK3 and BICD2. **(C, D, G, and H)** Representative immunofluorescence images of HeLa cells treated with either control or Dynein Heavy Chain (DHC) siRNA, followed by immunostaining with anti-RUFY1 and anti-CI-M6PR antibodies (C and D) or anti-Rab14 and anti-CI-M6PR antibodies (G and H). The inset indicates colocalized pixels. Bars: (main) 10 µm; (insets) 2 µm. **(E, F, and I)** The graph shows a radial profile plot of RUFY1 (E), CI-M6PR (F), and Rab14 (I) intensity in HeLa cells treated with either control or DHC siRNA. The values plotted are the mean ± SD from three independent experiments with 25 cells (E and F) and 15–20 cells (I) analyzed per experiment. **(J–M)** Representative confocal images of HeLa cells co-transfected with plasmids expressing 2x-FKBP-GFP-RUFY1 (WT or mutants as labeled) and DsRed-Rab14. Bars: (main) 10 µm; (insets) 2 µm. Source data are available for this figure: SourceData FS5.

Video 1. **Time-lapse imaging of control and Arl8b siRNA-treated HeLa cells expressing RUFY1-GFP and DsRed-Rab14.** The control siRNA video is captured at 0.23 frames per second with no time interval between the frames, and the Arl8b siRNA video is captured at 0.31 frames per second with no time interval between the frames. The movies are shown at 4 frames/s (the total number of frames displayed is 50).

Video 2. **Time-lapse SIM imaging of HeLa cells expressing mCherry-Rabip4 (shorter isoform) and GFP-SNX1.** The video is captured at 0.78 frames per second with no time interval between the frames. The movie is shown at 7 frames/s (the total number of frames displayed is 40). The white arrows indicate the association and dynamics of GFP-SNX1 tubules with mCherry-Rabip4 endosomes.

Video 3. **Example of a processive track traversed by a CD8α-M6PR endosome in particle-tracking analysis.** Live-cell imaging of HeLa cells expressing CD8α-M6PR to track the movement of surface-labeled CD8α-M6PR receptors from the plasma membrane towards the Golgi. The video was captured at 1.28 frames per second with a 20-millisecond interval between the frames (total number of frames traversed by endosome = 47) and analyzed with particle tracking software. The track of endosomes is represented by the color red, and the selected endosome is marked inside a blue square. The movie is shown at 7 frames/s.

Video 4. **Example of a non-processive track traversed by a CD8α-M6PR endosome in particle-tracking analysis.** Live-cell imaging of HeLa cells expressing CD8α-M6PR to track the movement of surface-labeled CD8α-M6PR receptors from the plasma membrane towards the Golgi. The video was captured at 1.28 frames per second with a 20-millisecond interval between the frames (total number of frames traversed by endosome = 246) and analyzed with particle tracking software. The track of endosomes is represented by the color red, and the selected endosome is marked inside a blue square. The movie is shown at 7 frames/s.

Video 5. **Example of a diffusive track traversed by a CD8α-M6PR endosome in particle-tracking analysis.** Live-cell imaging of HeLa cells expressing CD8α-M6PR to track the movement of surface-labeled CD8α-M6PR receptors from the plasma membrane towards the Golgi. The video was captured at 1.28 frames per second with a 20-millisecond interval between the frames (total number of frames traversed by endosome = 169) and analyzed with particle tracking software. The track of endosomes is represented by the color red, and the selected endosome is marked inside a blue square. The movie is shown at 7 frames/s.

**Provided online are Table S1, Table S2, and Table S3. Table S1 is a list of RUFY1 interacting proteins identified from HEK293T cell lysates by TAP-pulldown assay followed by mass spectrometric analysis. Table S2 is a list of DNA constructs and primer sequences used in this study. Table S3 is a list of antibodies used in this study.**

