## [Peer Review File · The Journal of Cell Biology]

RUFY1 binds Arl8b and mediates endosome-to-TGN CI-M6PR retrieval for cargo sorting to lysosomes

Shalini Rawat, Dhruva Chatterjee, Rituraj Marwaha, Gitanjali Charak, Gaurav Kumar, Shrestha Shaw, Divya Khatter, Sheetal Sharma, Cecilia de Heus, Nalan Liv, Judith Klumperman, Amit Tuli, and Mahak Sharma

Corresponding Author(s): Mahak Sharma, Indian Institute of Science Education and Research Mohali

Review Timeline:

Submission Date:	2021-08-01
Editorial Decision:	2021-09-14
Revision Received:	2022-08-26
Editorial Decision:	2022-09-22
Revision Received:	2022-09-27

Monitoring Editor: Erika Holzbaur

Scientific Editor: Andrea Marat

Transaction Report:

DOI: <https://doi.org/10.1083/jcb.202108001>

September 14, 2021

Re: JCB manuscript #202108001

Dr. Mahak Sharma
Indian Institute of Science Education and Research Mohali
Biological Sciences
Sector-81, Knowledge City
SAS Nagar
Mohali, PUNJAB 140306
India

Dear Dr. Sharma,

Thank you for submitting your manuscript entitled "RUFY1 binds Arl8b and mediate retrograde transport of CI-M6PR to maintain lysosomal homeostasis". Your work was reviewed by two experts in the field, both of whom found the question of interest. However, both found the work in its present form falls short of the impact expected for a paper in JCB, as key mechanistic information is still lacking. Both referees suggest that further characterization of the interaction with dynein would be required to increase the overall impact of the advance. In addition, both reviewers raised significant technical concerns that must be addressed prior to publication.

Although your manuscript is intriguing, I feel that the points raised by the reviewers are more substantial than can be addressed in a typical revision period. If you wish to expedite publication of the current data, it may be best to pursue publication at another journal. Our journal office will transfer the reviews at your request, including to our partner journals Life Science Alliance, Journal of Cell Science, or Molecular Biology of the Cell.

We hope that the detailed comments provided will help guide the further development of this interesting work. If you are able to further develop mechanistic insights, as well as address the specific technical concerns, then JCB would be willing to reconsider this work, a process which would include further consultation with experts in the field. If you would like to resubmit this work to JCB, please contact the journal office to discuss an appeal of this decision or you may submit an appeal directly through our manuscript submission system. Please note that priority and novelty would be reassessed at resubmission.

Regardless of how you choose to proceed, we hope that the comments below will prove constructive as your work progresses. We would be happy to discuss the reviewer comments further once you've had a chance to consider the points raised in this letter. You can contact the journal office with any questions, cellbio@rockefeller.edu or call (212) 327-8588.

Thank you for thinking of JCB as an appropriate place to publish your work.

Sincerely,

Erika Holzbaur
Monitoring Editor

Andrea L. Marat
Senior Scientific Editor

Journal of Cell Biology

Reviewer #1 (Comments to the Authors (Required)):

In this manuscript, Rawat et al document a role for the RUN domain-containing protein RUFY1 (aka Rabip4/4') in regulating the delivery of lysosomal enzymes from endosomes to late endosomes/ lysosomes. Rapip4/4' (the name that they use throughout the paper, despite using RUFY1 in the title) has been well shown to regulate early endosomal dynamics, in part by binding to Rab4, Rab14, and Rab5. Here the authors first provide evidence that Rabip4/4', via its RUN domain, binds to the GTP-bound form of the small GTPase Arl8b, which is typically thought of as localized to late endosomes/lysosomes, on a subset of early endosomal tubulovesicular structures marked by RAB14 and EEA1. They show that knockdown or knockout of Arl8b results in a loss of Rabip4/4' stable association with endosomes, indicating the importance of this interaction. They then focus on the effect of Rapip4/4' depletion on lysosomal structure and activity, showing that it results in lysosomal enlargement, modest increase in pH, increase in cathepsin D content and secretion, displacement of mannose 6-phosphate receptor to endosomes and impaired

transport of newly synthesized cathepsin Z to late endosomes/ lysosomes. Finally, several data suggest that the latter results from a failure of dynein-dependent transport to the perinuclear region via a proposed direct Rabip4/4' interaction with dynein light chain. The authors conclude that Rabip4/4' is an adaptor for maturing endosomes to dynein-driven microtubule transport to facilitate endosomal maturation.

The manuscript is replete with a ton of data on Rabip4/4' function, and with a few exceptions described below, the data are very well done, nicely quantified, and convincing. The RUSH experiment showing that Rabip4/4' depletion interferes with Cathepsin Z trafficking to late endosomes/ lysosomes in Fig. 7 is particularly nice, and the increased secretion of endogenous pro-cathepsin D supports the physiological significance of the mistrafficking. All in all, with a few additions of key controls and toning down of some of the conclusions, the manuscript makes a solid contribution to the literature.

However, while the data are largely convincing and very well done, it is not clear whether the findings represent a substantial advance in our understanding of the control of endosomal dynamics. Perhaps the most surprising finding here is that Arl8b localizes to a subset of early endosomes in addition to late endosomes/ lysosomes, and that it participates in the recruitment and/or activation of Rabip4/4' for retrograde trafficking on microtubules. The authors do not explain how Arl8b participates in the recruitment of Rabip4/4' for ultimate dynein recruitment on these structures but of PLEKHM2 to late endosomes/ lysosomes for ultimate kinesin recruitment, or how Arl8b and Rab14 cooperate in Rabip4/4' recruitment (e.g. is Arl8b needed to support Rab14 recruitment? Is coincidence detection of both Rab14 and Arl8b required for recruitment?). Most of the rest of the paper, although mostly well done, could largely be predicted from previous characterizations of Rabip4/4'. Indeed, many of the interactions described in this paper had been identified before, and the function of the most critical interaction - that of dynein to Rabip4/4' - is not sufficiently defined to allow for a solid conclusion regarding its physiological significance. Even if this experiment were improved, it is not clear if the advance is of sufficient interest to readers of JCB.

Below are specific concerns about some of the experiments and their presentation.

Major concerns

1. While the data in Fig. S6, Fig. 8D-K, and Fig. 9A-H are convincing that Rabip4' drives motility to the perinuclear region and that this requires dynein, the reliance on dynein is not surprising and the results do not necessarily support the conclusion that a dynein/Rabip4' interaction mediates this localization. To conclude this, the authors would need to identify the dynein binding site on Rabip4', mutagenize it, and show that this construct no longer mediated perinuclear clustering. Absent that, it is not possible to conclude that the interaction mediates the phenotype. Any perinuclear movement would be expected to rely on dynein/ dynactin.
2. Many of the biochemical experiments lack suitable negative controls, and thus the conclusions drawn from them are not well justified. For example:
 - Fig. 1C lacks a negative control of a large protein that does not interact with GST-Arl8b. This concern is partially mitigated by Figure 3 in which the deltaRUN version does not bind to Arl8b, but in that case we do not know if the deltaRUN version folds properly. Moreover, a more comforting negative control than GST alone would be the GDP-bound form of Arl8b (relative to GTPgammaS-bound Arl8b). The same applies to Fig. S1B.
 - Fig. 1D lack a suitable negative IP control. Anti-IgG (to something absent in the lysate) is not a suitable negative control for a co-IP experiment; an antibody to another endogenous protein, such as Arf1 or a negative control Rab, would be more suitable.
 - In Fig. 3D, a lower exposure of the anti-His IB input lanes should be shown to ensure that the T34N variant of His-Arl8b is expressed at similar levels to the WT and Q75L variants. If not, then the experiment should be repeated with more T34N to be convincing.
 - Fig. 8A, B lack suitable negative controls in much the same way as described for Fig. 1C and D.
3. Fig. 4G,H - the authors have not yet shown us that Rabip4'-GFP behaves like endogenous Rabip4' in the absence of Arl8b, and in fact the cell shown with the Arl8b siRNA has equivalent Rabip4'-GFP localization to puncta as the control siRNA cell. This suggests that Rabip4' overexpression or its conjugation to GFP alters its properties, and raises concerns about the interpretation of the FRAP experiment.
4. In Fig. 5D, which siRNA was rescued with resistant construct? This needs to be better labeled in the figure and the legend. If it was the SP, then the rescue efficiency does not look to be significant. Note, the authors should consider a different way of quantifying the data to take into account the substantial variability in LAMP-positive labeling area in the control siRNA cells; the lack of SD indicators in the control belies clear variability in the images. Finally, a representative image of the rescue should be shown in this set of figures.
5. Fig. 5H is a strange experiment, and the conclusions are not justified. How does the cell "lose" rhodamine signal upon degradation of rhodamine-EGF? The signal should be retained (although perhaps released in a TCA soluble form). This result might rather reflect an increase in recycling rather than a decrease in the rate of degradation.

6. Table S1: The authors should comment on the lack of identification of other known Rabip4/4' interactors - particularly Arl8 or Rab14, documented in this manuscript - among the hits found in the proteomics. Note, although they "identify" AP-3 among the hits, it looks like only a single peptide from the AP3B1 chain and none from any other subunits; this would not be considered a solid "hit". The authors also need to explain the format of the results - particularly the "Quant (SPC)" column; what are the 3 numbers shown?

7. Although the RUSH experiment in Figure 7 is beautiful and convincing, the linkage of lysosomal trafficking to the Arl8b association in the beginning of the paper would be strengthened if the cargo at the later time points was trapped in the Rab14+/Arl8b+ compartments described in Figures 1-2. Is this true?

Minor concerns:

8. In the legend to Fig. 1B, siRNA #1, #2 and SP must be defined.

9. The conclusion from Fig. S1C-K should be that Rabip4/4' localizes to a subset of Rab14-positive early endosomes, not "to Rab14-positive early/sorting endosomes", as it appears that there are a substantial number of Rab14-containing structures that lack Rabip4/4'.

10. In Figure 1F, the authors presumably show endogenous Arl8b labeling, but it is surprising that so much of it accumulates at the cell periphery; this is normally a feature of overexpressed Arl8b. Is this a typical observation?

11. In the text on page 9, the authors mistakenly state that Figure S2B shows a population of Arl8b-positive structures that is positive for Rab7 and negative for Rabip4'; rather, the figure shows the opposite population.

12. The legend to Fig. S3A should include a description of the color coding used in "conservation" and "quality", as well as of the definitions of "conservation", "quality" and "occupancy".

13. The conclusion from Figures 3 and S3 on page 11, the end of the first paragraph, is premature; the conclusion that "Arl8b regulates Rabip4' endosomal localization" is not firmly shown until Figure 4.

14. Although not a major point, the increased size and brightness of Rabip4/4'-positive structures in cells overexpressing Arl8b-HA in Fig. S4D is not terribly convincing without quantification.

15. It is confusing that the lower band in Fig. 1B corresponds to Rabip4 (i.e. disappears with siRNA), but the lower band in Figure 4E/ S4E is non-specific (i.e. does not disappear with siRNA). Were two different antibodies used in these experiments? If not, please explain the difference between these two experiments.

16. The authors conclude from the data in Figure 7 that Rabip4/4' siRNA causes a delay in cathepsin Z delivery to lysosomes, but they never show a time point in which delivery is observed. Is it actually a delay or mistargeting? Either a later time point should be shown to confirm the delay or the conclusion should be modified. Also in this figure, it should be clarified whether the different time points are from the same cell imaged over time or from different cells at different time points.

17. Video 1 is not very informative.

18. It is surprising in Fig. 9E-H that non-processive tracks are increased upon Rabip4/4' if only dynein association is impaired. Is the association of kinesin with endosomes also impaired upon Rabip4/4' depletion?

19. The title should indicate that Rabip4/4' is an alternative name for RUFY1. While this reviewer understands the deference to the original name for the protein throughout the text, RUFY1 is a more sensible modern name given the homology with the other RUFY proteins and in my view would be much cleaner to refer to throughout the paper.

Reviewer #2 (Comments to the Authors (Required)):

Rawat et al find an unexpected role for the GTPase Arl8b in endosomal/Rab14-dependent trafficking to the TGN via an interaction with RUFY1. Some of the main findings make this study potentially interesting for the readership of JCB: First, it places some of Arl8b's function within the Rab14-endosomal compartment and upstream of lysosomes, shedding some light into the specificity of these GTPases in the endolysosomal pathway. Second, it suggests that the RUFY family of proteins may be dynein activating adaptors critical for activating the motility of dynein dependent cargos. Understanding how dynein's cargo selectivity is achieved in cells is a critical question in the field. Overall, the experiments are well-executed. However, a few key findings (and some minor ones) need to be investigated further to be considered for acceptance into JCB:

(1) The possibility of RUFY being a bona fide coiled coil activating adaptor for dynein is very interesting. The gold standard

experiment for this is to reconstitute the dynein transport complex (DDX) with the activating adaptor followed by in vitro motility assays on microtubules. I understand that these reconstitution experiments are beyond the scope of the current study, and thus, definitively proving that RUFY1 is an activating adaptor is not possible. However, I suggest the following structure/function experiments in the mitochondria relocalization assays (Figure 8D-K) and coimmunoprecipitation assays (Figure 8 A,B) to put RUFY1 in context of the growing list of dynein adaptors and solidify the interaction between RUFY1/dynein:

- a. Are all three coiled coils (CCs) necessary and sufficient? For example, HOOK3 only requires N-terminal region CCs and HOOK domains; C-terminal interactions are dispensable. It would be interesting to know if the N-terminus of RUFY1 (~1-500) are all that is required for its interaction with dynein and sufficient for rapamycin-induced mitochondrial translocation.
 - b. Specifically deleting the RUN domain in this context would also be informative.
- (2) What happens to Rab14 localization in DHC siRNA treated cells (extension of Figure S7A-C)? The prediction is that it should also be mis-localized since there is such an extensive overlap between Rab14 and Rabip4/4'.
- (3) The robustness of the Arl8b/RUFY1 interaction is critical for this study. There are a couple of experiments that would strengthen the Arl8b/RUFY1 connection:
- a. The authors should validate the Arl8b knockdown/knockout cells by immunofluorescence and/or Western blot. It is possible that some of the overlapping Rab14/RUFY1 puncta are still on residual Arl8b vesicles (partial knockdown/knockouts)
 - b. The endogenous colocalization of Arl8b and RUFY1 would be ideal. I understand that good antibodies can be tough to come by, so at the very least, the authors should comment on this. Perhaps using a different epitope tag (Arl8b-FLAG or myc) to corroborate the main findings.
- (4) The literature suggests that Rabip4' plays multiple roles in the endolysosomal pathway.
- a. How do the authors reconcile the findings demonstrating that Rabip4' regulates lysosome distribution (Ivan et al., 2012)? A comment in the discussion should be made here.
 - b. Since RUFY1 can interact with Rab4-marked early endosomes via its FYVE domain (Yamamoto et al., 2010), the model in Figure 9 should reflect these findings and put them into context in the discussion.
 - c. Based on this model, shouldn't there be a subset of Rab14/Arl8b puncta that do not overlap with Rab4? This should be tested.
- (5) The photobleaching experiment in Figure 4G-I is confusing. Shouldn't the most relevant populations of Rabip4'GFP coated vesicles for this analysis be the motile ones (which makes it nearly impossible for photobleaching experiments)? Also, I am confused by the interpretation of the results. If Rabip4' recovers quickly in the Arl8b knockdown, shouldn't this mean that Arl8b is dispensable for binding to membranes because it is MORE stably associated because presumably the new recovered pool of RUFY1 is coming from the cytosol, not other membranes?

Other points:

- (1) In Figure S2B, it would be helpful to point out the two populations with arrows.

Response to Reviewers Comments:

We thank the reviewers for their valuable comments and suggestions pertaining to our manuscript. Below we provide a point-by-point response (shown in blue) to how we have addressed the concerns raised by the reviewers (shown in black).

Reviewer #1 (Comments to the Authors (Required)): The manuscript is replete with a ton of data on Rabip4/4' function, and with a few exceptions described below, the data are very well done, nicely quantified, and convincing. The RUSH experiment showing that Rapip4/4' depletion interferes with Cathepsin Z trafficking to late endosomes/ lysosomes in Fig. 7 is particularly nice, and the increased secretion of endogenous pro-cathepsin D supports the physiological significance of the mistrafficking. All in all, with a few additions of key controls and toning down of some of the conclusions, the manuscript makes a solid contribution to the literature.

However, while the data are largely convincing and very well done, it is not clear whether the findings represent a substantial advance in our understanding of the control of endosomal dynamics. Perhaps the most surprising finding here is that Arl8b localizes to a subset of early endosomes in addition to late endosomes/ lysosomes, and that it participates in the recruitment and/or activation of Rabip4/4' for retrograde trafficking on microtubules. The authors do not explain how Arl8b participates in the recruitment of Rabip4/4' for ultimate dynein recruitment on these structures but of PLEKHM2 to late endosomes/ lysosomes for ultimate kinesin recruitment, or how Arl8b and Rab14 cooperate in Rabip4/4' recruitment (e.g. is Arl8b needed to support Rab14 recruitment? Is coincidence detection of both Rab14 and Arl8b required for recruitment?).

Response: We appreciate the reviewer's insightful questions. To address the mechanism by which Arl8b regulates RUFY1 membrane localization, we have added new data to the manuscript demonstrating that while Arl8b recruits its known effector SKIP/PLEKHM2 to membranes (Mito-fusion Arl8b (Q75L) recruits SKIP to mitochondria), it is insufficient for RUFY1 recruitment on membranes (**Fig. S3E-F**). We show that of the four small G proteins reported to bind to RUFY1, i.e., Rab14, Rab4, Rab5 and Arl8b, only Rab14 is able to recruit RUFY1 on membranes (**Fig. S3F-I**).

Surprisingly, we found that even when Rab14 was overexpressed in Arl8b knockdown cells, a significant proportion of RUFY1 remained cytosolic, despite Rab14's endosomal distribution (**see Video S1 and Fig. S3M**). This was reflected in Arl8b-depleted cells by a reduced co-immunoprecipitation of RUFY1 with Rab14 (**Fig. 4G-H**). Following that, the possibility that Arl8b regulates RUFY1 interaction with Rab14 was investigated. To test this, the three proteins were recombinantly isolated, and RUFY1 was incubated with GTP-loaded Rab14 in the presence of increasing amounts of His-tagged Arl8b protein. The pulldown of RUFY1 with Rab14 increased with increasing amounts of Arl8b, although no such trend was observed in the presence of His-tagged Rab7a (**Fig. 4I**), another G protein on late endosomes and lysosomes that does not colocalize with RUFY1 (**Fig. 2E**).

Interestingly, deletion of the RUN domain-containing region (NΔRUN RUFY1; internal deletion of residues 147–270 a.a.) enhanced RUFY1 interaction with Rab14 in a yeast two-hybrid assay (**Fig. S3N**). These findings have led us to hypothesize that Arl8b binding to the RUN domain likely relieves RUFY1 autoinhibition and promotes RUFY1 and Rab14 interaction, resulting in RUFY1 stable membrane localization. Interestingly, Arl8b is shown to play a similar role in relieving SKIP autoinhibition for binding to the anterograde motor kinesin-1 (Kaplan KT and Bonifacino JS, *Current Biology* 2021).

Most of the rest of the paper, although mostly well done, could largely be predicted from previous characterizations of Rabip4/4'. Indeed, many of the interactions described in this paper had been identified before, and the function of the most critical interaction - that of dynein to Rabip4/4' - is not sufficiently defined to allow for a solid conclusion regarding its physiological significance. Even if this experiment were improved, it is not clear if the advance is of sufficient interest to readers of JCB.

Response: We respectfully disagree with the reviewer's assessment of the work's novelty and significance. While RUFY1's interaction with Rab4 and Rab14 was known (Yamamoto H et al., *Molecular Biology of the Cell* 2010), the function of RUFY1 on these recycling endosomes was unclear based on previous research. Here, we report that RUFY1 is a dynein adaptor on these Rab14-positive sorting/recycling endosomes that mediates the retrieval of CI-M6PR towards the TGN and, in turn, the delivery of lysosomal hydrolases. RUFY1 interacts with Arl8b on these specific subsets of endosomes, and Arl8b-binding is required for RUFY1 endosomal localization (**Fig. 4**). We have now added new evidence towards an understanding of how RUFY1 interacts with dynein-the role of the first two coiled-coil regions of RUFY1 in engaging the dynein-dynactin complex (**Fig. 9**). Furthermore, we found that RUFY1 binding to dynein is required for CI-M6PR sorting towards the TGN, as a dynein-binding-defective RUFY1 mutant was unable to mediate CI-M6PR retrieval from endosomes to the TGN (**Fig. 10F-D**).

Below are specific concerns about some of the experiments and their presentation.

Major concerns

1. While the data in Fig. S6, Fig. 8D-K, and Fig. 9A-H are convincing that Rabip4' drives motility to the perinuclear region and that this requires dynein, the reliance on dynein is not surprising and the results do not necessarily support the conclusion that a dynein/Rabip4' interaction mediates this localization. To conclude this, the authors would need to identify the dynein binding site on Rabip4', mutagenize it, and show that this construct no longer mediated perinuclear clustering. Absent that, it is not possible to conclude that the interaction mediates the phenotype. Any perinuclear movement would be expected to rely on dynein/ dynactin.

Response: To address the reviewer's comment, we created different RUFY1 mutants with varying lengths of the coiled-coil (CC) region or internal deletions of either the first (CC1), second (CC2) or both coiled coil (CC21+CC2) regions (**Fig. 9A**). The reason for

focusing on the CC regions is that, like other activating dynein adaptors such as BICD2 and Hook3, RUFY1 has an extended long coiled-coil region of 300 amino acids. Previous research has demonstrated that the long coiled-coil segment (350 Å) of activating dynein adaptor proteins runs parallel to the dynactin filament and serves as a binding surface for dynein tails (Urnavicius L et al., *Science* 2015; Urnavicius L et al., *Nature* 2018; Grotjahn DA et al., *Nat Struct Mol Biol.* 2018). Our findings reveal that the first two coiled-coil regions of RUFY1 are important for the dynein-dynactin engagement, as revealed by the reduced mitochondrial clustering of RUFY1 mutants lacking this region (**Fig. 9F-H** and **9J**). Here, we noted that the RUFY1 mutant lacking the second coiled-coil (Δ CC2) showed a stronger defect in mitochondrial clustering (**Fig. 9G** and **9J**). We have further confirmed our observations via a GST pulldown assay that shows reduced interaction of RUFY1 mutants lacking either the first (Δ CC1) or the second (Δ CC2) or both coiled-coil regions (Δ CC1+ Δ CC2) with the dynein subunit-LIC1 (**Fig. 9K-L**). Finally, we have shown that a dynein binding-defective RUFY1 mutant (RUFY1 (Δ CC2)) was unable to rescue CI-M6PR steady state distribution (**Fig. 10F-I**), suggesting that dynein-binding is crucial for RUFY1 function in regulating CI-M6PR retrieval from endosomes to TGN.

2. Many of the biochemical experiments lack suitable negative controls, and thus the conclusions drawn from them are not well justified. For example:

- Fig. 1C lacks a negative control of a large protein that does not interact with GST-Arl8b. This concern is partially mitigated by Figure 3 in which the deltaRUN version does not bind to Arl8b, but in that case we do not know if the deltaRUN version folds properly. Moreover, a more comforting negative control than GST alone would be the GDP-bound form of Arl8b (relative to GTPgammaS-bound Arl8b). The same applies to Fig. S1B.

Response: We appreciate the reviewer's comments and have replaced the experiment in **Fig. 1C** with a new experiment in which GTP- and GDP-loaded Arl8b were used as bait for RUFY1 pulldown. Only the GTP-loaded form of Arl8b was able to pull down RUFY1, as shown in revised **Fig. 1C**. A comparable control has been added to **Fig. S1B**.

- Fig. 1D lack a suitable negative IP control. Anti-IgG (to something absent in the lysate) is not a suitable negative control for a co-IP experiment; an antibody to another endogenous protein, such as Arf1 or a negative control Rab, would be more suitable.

Response: We have added the control mentioned by the reviewer. As shown in **Fig. S1A**, Rab8 (used as a control small G protein) was not able to co-immunoprecipitate RUFY1 from HeLa cell lysates.

- In Fig. 3D, a lower exposure of the anti-His IB input lanes should be shown to ensure that the T34N variant of His-Arl8b is expressed at similar levels to the WT and Q75L variants. If not, then the experiment should be repeated with more T34N to be convincing.

Response: We thank the reviewer for this critical comment. To address this comment, we have repeated new sets of experiments, displaying a lower exposure blot from one of them (**Fig. 3D**). We have also densitometrically quantified these blots, as shown in **Fig. 3E**.

• Fig. 8A, B lack suitable negative controls in much the same way as described for Fig. 1C and D.

Response: We've added the control (IP with anti-RSK1 antibody) mentioned by the reviewer and presented a new experiment in **Fig. S5A**. We show that the co-immunoprecipitation of dynein subunit DIC and p150^{glued} was substantially more with RUFY1 than with RSK1. Fig. 8B from the previous version has been replaced with a new figure demonstrating that RUFY1 interacts with the C-terminal region of LIC1 via its RUN domain-containing region, since the binding of the RUFY1 (Δ RUN) mutant to LIC1 was significantly reduced as compared to RUFY1 (WT) (see new **Fig. 8B**). Also, as shown in **Fig. 8C**, RUFY1 did not interact with an LIC1 F447A/F448A (FFAA) mutant in which conserved phenylalanine residues in the C-terminal amphipathic helix of LIC1 that are needed for interaction with dynein adaptors were mutated.

3. Fig. 4G,H - the authors have not yet shown us that Rabip4'-GFP behaves like endogenous Rabip4' in the absence of Arl8b, and in fact the cell shown with the Arl8b siRNA has equivalent Rapip4'-GFP localization to puncta as the control siRNA cell. This suggests that Rapip4' overexpression or its conjugation to GFP alters its properties, and raises concerns about the interpretation of the FRAP experiment.

Response: In light of reviewer 2's comment that motile endosomes (the relevant population here) are not suitable for FRAP, we have decided not to include the FRAP experiment in this revised version of the manuscript. Nonetheless, we demonstrated in **Fig. 6I-J, Fig. 10G and 10I** that the RUFY1-GFP construct can rescue the CI-M6PR localization defect observed in RUFY1 depleted cells, implying that this construct can replace endogenous protein function.

4. In Fig. 5D, which siRNA was rescued with resistant construct? This needs to be better labeled in the figure and the legend. If it was the SP, then the rescue efficiency does not look to be significant. Note, the authors should consider a different way of quantifying the data to take into account the substantial variability in LAMP-positive labeling area in the control siRNA cells; the lack of SD indicators in the control belies clear variability in the images. Finally, a representative image of the rescue should be shown in this set of figures.

Response: We sincerely apologize for this oversight. The rescue construct is intended to correct the phenotype caused by RUFY1 siRNA #1. This information has been added to the figure labels (**Fig. 5D-E and 5O**). We previously demonstrated the fold change in lysosome size by normalizing it to control. We have now shown the same quantification as a super plot with the average lysosome area/cell plotted for control, RUFY1 knockdown, and rescue construct (against siRNA #1)-transfected cells from three

independent experiments (Fig. 5E). The rescue construct representative image has been added (Fig. 5D). We'd also like to mention that the lysosome area in control and RUFY1 depletion has also been quantified from the higher resolution images obtained using immuno-electron microscopy (Fig. 5F-G).

5. Fig. 5H is a strange experiment, and the conclusions are not justified. How does the cell "lose" rhodamine signal upon degradation of rhodamine-EGF? The signal should be retained (although perhaps released in a TCA soluble form). This result might rather reflect an increase in recycling rather than a decrease in the rate of degradation.

Response: We thank the reviewer for bringing this to our attention, and we agree that our interpretation of rhodamine signal loss was incorrect in that it monitors EGFR degradation. We have now quantified the corrected total cell fluorescence (CTCF) of the EGF receptor at various time points after EGF stimulation. Our findings show that RUFY1 knockdown significantly delays EGFR degradation (Fig. 5I). With the reduced degradation of another endocytic cargo-BODIPY-BSA, we conclude that RUFY1 depletion impairs the cargo degradative function of lysosomes (Fig. S4K). The reduced delivery of pro-cathepsins to the lysosome observed in RUFY1 knockdown cells explains these cells' impaired lysosomal degradation ability (Fig. 7).

6. Table S1: The authors should comment on the lack of identification of other known Rabip4/4' interactors - particularly Arl8 or Rab14, documented in this manuscript - among the hits found in the proteomics. Note, although they "identify" AP-3 among the hits, it looks like only a single peptide from the AP3B1 chain and none from any other subunits; this would not be considered a solid "hit". The authors also need to explain the format of the results - particularly the "Quant (SPC)" column; what are the 3 numbers shown?

Response: We appreciate the reviewer bringing this to our attention. We have incorporated reviewer comments on the lack of identification of Arl8b or Rab14 in the TAP-pulldown in manuscript text (line # 427-429) and clarified the meaning of "Quant (SPC)" in the figure legend of **Supplementary Table I**. SPC stands for "spectral peptide count," which is the total number of peptides found in three separate experiments for a given protein (the number of peptides found in each experiment is listed in the column and separated by a bar).

7. Although the RUSH experiment in Figure 7 is beautiful and convincing, the linkage of lysosomal trafficking to the Arl8b association in the beginning of the paper would be strengthened if the cargo at the later time points was trapped in the Rab14+/ Arl8b+ compartments described in Figures 1-2. Is this true?

Response: We thank the reviewer for bringing this interesting observation to our attention. The RUSH assay revealed that cathepsin Z exit from the Golgi appears to be delayed (Fig. 7B-J). This observation is consistent with our findings that the cathepsin Z receptor, CI-M6PR, is delayed in its retrieval back to the TGN in RUFY1-depleted cells (Fig. 6M-Q). To address this comment, we measured and found that CI-M6PR

colocalization with Rab14 is indeed increased in RUFY1-depleted cells (Fig. S4M-O). Thus, RUFY1 regulates CI-M6PR retrieval from a Rab14-positive compartment to the TGN.

Minor concerns:

8. In the legend to Fig. 1B, siRNA #1, #2 and SP must be defined.

Response: We appreciate the reviewer bringing this to our attention. The figure legend has been revised and the sequences of RUFY1 siRNA #1, #2, and the catalog number of RUFY1 SMARTpool are described in the material and method section.

9. The conclusion from Fig. S1C-K should be that Rabip4/4' localizes to a subset of Rab14-positive early endosomes, not "to Rab14-positive early/sorting endosomes", as it appears that there are a substantial number of Rab14-containing structures that lack Rapip4/4'.

Response: We appreciate the reviewer bringing this to our attention. We have made the necessary changes to the text.

10. In Figure 1F, the authors presumably show endogenous Arl8b labeling, but it is surprising that so much of it accumulates at the cell periphery; this is normally a feature of overexpressed Arl8b. Is this a typical observation?

Response: Yes, this is typical. Please see the images below for examples of HeLa cells immunostained for endogenous Arl8b.

Typical localization of Arl8b in HeLa cells. HeLa cells seeded on coverslips were fixed and immunostained for endogenous Arl8b (shown in green), and the nucleus was stained using DAPI. Different field images are shown to depict typical Arl8b localization in the cells. Bars: 10 µm.

11. In the text on page 9, the authors mistakenly state that Figure S2B shows a population of Arl8b-positive structures that is positive for Rab7 and negative for Rabip4'; rather, the figure shows the opposite population.

Response: We sincerely apologize for the error. We have now corrected the text and added arrowheads to the image to show the two different populations of Arl8b (**Fig. S2B**).

12. The legend to Fig. S3A should include a description of the color coding used in "conservation" and "quality", as well as of the definitions of "conservation", "quality" and "occupancy".

Response: Due to the addition of several new experiments, we have omitted the previous Fig. S3A from this version of the manuscript. The original Fig. S3A showed the alignment between the RUN domains of PLEKHM1, PLEKHM2, and RUFY1 RUN domains.

13. The conclusion from Figures 3 and S3 on page 11, the end of the first paragraph, is premature; the conclusion that "Arl8b regulates Rabip4' endosomal localization" is not firmly shown until Figure 4.

Response: We agree with the reviewer's suggestion and have made the necessary changes to the text. This heading now comes in the result section where we describe **Fig. 4**.

14. Although not a major point, the increased size and brightness of Rabip4/4'-positive structures in cells overexpressing Arl8b-HA in Fig. S4D is not terribly convincing without quantification.

Response: We agree with the reviewer and have now included the quantification as **Fig. S2K**, and the corresponding image is shown in **Fig. S2J**.

15. It is confusing that the lower band in Fig. 1B corresponds to Rapip4 (i.e. disappears with siRNA), but the lower band in Figure 4E/ S4E is non-specific (i.e. does not disappear with siRNA). Were two different antibodies used in these experiments? If not, please explain the difference between these two experiments.

Response: Yes, in **Fig. 1B**, anti-RUFY1 antibody from Abcam (catalog #: ab241080) was used, while in **Fig. 4E** and **S2L**, anti-RUFY1 antibody from Santa Cruz Biotechnology (SCBT; catalog #: sc-398740) was used. When we performed the **Fig. 4E** experiments, the Abcam antibody was not working and produced a lot of background, so we used the SCBT antibody instead.

We have performed siRNA followed by western blotting to confirm the specificities of these antibodies (**Fig. 1B** and **Fig. S2L**). We have also included details of the antibodies used in the figure legends and material and method section.

16. The authors conclude from the data in Figure 7 that Rapip4/4' siRNA causes a delay in cathepsin Z delivery to lysosomes, but they never show a time point in which delivery is observed. Is it actually a delay or mistargeting? Either a later time point should be

shown to confirm the delay or the conclusion should be modified. Also in this figure, it should be clarified whether the different time points are from the same cell imaged over time or from different cells at different time points.

Response: We appreciate the reviewer's excellent suggestion. We have now repeated these experiments and included longer time points of chase in the RUSH assay, up to 320 min (**Fig. 7B-J**). In RUFY1 depleted cells, cathepsin Z colocalization to lysosomes is observed after 260 min (**Fig. 7I and 7J**), whereas in control cells, such colocalization is observed between 120 and 180 min (**Fig. 7D and 7J**). We conclude that RUFY1 depletion causes a delay rather than a complete block in cathepsin delivery to lysosomes. **Fig. 7** shows representative images from different cells at every time point, and this information is mentioned in the figure legend.

17. Video 1 is not very informative.

Response: As a similar experiment is shown in **Fig. 10B**, we agree with the reviewer's suggestion and have removed Video 1.

18. It is surprising in Fig. 9E-H that non-processive tracks are increased upon Rabip4/4' if only dynein association is impaired. Is the association of kinesin with endosomes also impaired upon Rabip4/4' depletion?

Response: We appreciate the reviewer's thoughtful and important question. Fig. 9E (now **Fig. 10D**) depicts particle tracking of newly endocytosed CD8 α -M6PR vesicles, which typically move in a retrograde direction towards the TGN, and thus the motility is primarily dynein-driven. After 30 minutes of chase, the majority of CD8 α -M6PR vesicles are in the perinuclear TGN region. We do not understand why non-processive tracks were more prevalent in RUFY1-depleted cells rather than vesicles moving in an anterograde direction. One explanation could be that anterograde motors, such as Kif16b, which binds to Rab14 and is expected to be on this endosome (Ueno H et al., *Dev Cell*. 2011), are not in their active configuration due to a lack of active motor adaptors. We are pursuing these questions in our future studies.

19. The title should indicate that Rabip4/4' is an alternative name for RUFY1. While this reviewer understands the deference to the original name for the protein throughout the text, RUFY1 is a more sensible modern name given the homology with the other RUFY proteins and in my view would be much cleaner to refer to throughout the paper.

Response: We agree with the reviewer's suggestion and have replaced Rabip4/4' in the manuscript text with RUFY1.

Reviewer #2 (Comments to the Authors (Required)):

Rawat et al find an unexpected role for the GTPase Arl8b in endosomal/Rab14-dependent trafficking to the TGN via an interaction with RUFY1. Some of the main findings make this study potentially interesting for the readership of JCB: First, it places some of Arl8b's

function within the Rab14-endosomal compartment and upstream of lysosomes, shedding some light into the specificity of these GTPases in the endolysosomal pathway. Second, it suggests that the RUFY family of proteins may be dynein-activating adaptors critical for activating the motility of dynein dependent cargos. Understanding how dynein's cargo selectivity is achieved in cells is a critical question in the field. Overall, the experiments are well executed. However, a few key findings (and some minor ones) need to be investigated further to be considered for acceptance into JCB:

(1) The possibility of RUFY being a bona fide coiled coil activating adaptor for dynein is very interesting. The gold standard experiment for this is to reconstitute the dynein transport complex (DDX) with the activating adaptor followed by in vitro motility assays on microtubules. I understand that these reconstitution experiments are beyond the scope of the current study, and thus, definitively proving that RUFY1 is an activating adaptor is not possible. However, I suggest the following structure/function experiments in the mitochondria relocalization assays (Figure 8D-K) and coimmunoprecipitation assays (Figure 8 A,B) to put RUFY1 in context of the growing list of dynein adaptors and solidify the interaction between RUFY1/dynein:

a. Are all three coiled coils (CCs) necessary and sufficient? For example, HOOK3 only requires N-terminal region CCs and HOOK domains; C-terminal interactions are dispensable. It would be interesting to know if the N-terminus of RUFY1 (~1-500) are all that is required for its interaction with dynein and sufficient for rapamycin-induced mitochondrial translocation.

b. Specifically deleting the RUN domain in this context would also be informative.

Response: We thank the reviewer for these extremely useful suggestions, and in response, we created different RUFY1 mutants with varying lengths of the coiled-coil (CC) region or internal deletions of either the first (CC1), second (CC2), or both coiled-coil (CC21+CC2) regions. The results in **Fig. 9** show that reducing the coiled-coil region impairs RUFY1's ability to interact with dynein subunit-LIC1 and mediate dynein-dependent mitochondrial clustering that was analyzed via the FRB-FKBP heterodimerization approach. RUFY1 lacking the first (Δ CC1), or the second (Δ CC2), or both first and second coiled-coil (Δ CC1+ Δ CC2) regions (residues 300-500 a.a.) exhibit a significant decrease in their ability to promote mitochondrial clustering, as well as show significantly weaker interaction with LIC1 dynein subunit, as compared to RUFY1 (WT) (**Fig. 9F-H and 9J-L**). Here, we noted that the RUFY1 mutant lacking the second coiled-coil (Δ CC2) showed a stronger defect in mitochondrial clustering (**Fig. 9G and 9J**). This mutant was employed to rescue the CI-M6PR localization defect in RUFY1 knockdown cells. As shown in **Fig. 10F-I**, the dynein binding-defective RUFY1 mutant (RUFY1 (Δ CC2)) was unable to rescue CI-M6PR steady state distribution, and CI-M6PR remained stuck in endosomes in these cells, suggesting that RUFY1-dynein-dynactin interaction is required for CI-M6PR retrograde transport.

We also found that the RUFY1 RUN domain-containing region (1-300 amino acids) is required for its binding to the LIC1 (**Fig. 8B**). Deletion of the RUN domain-containing region reduced RUFY1's ability to engage LIC1 and to mediate perinuclear

organelle clustering (**Fig. 9I-J**), suggesting that both the LIC1 binding site and the coiled-coil regions are required for functional engagement of the dynein-dynactin complex. Indeed, LIC1 binding in dynein activating adaptors has been shown to be important for the processive motility of cargo by the dynein-dynactin complex (Lee IG et al., *Nat Commun.* 2020). Excitingly, recent research has demonstrated that RUFY3 and RUFY4 are Arl8 effectors that localize to late endosomes and lysosomes and interact with the dynein-dynactin complex to facilitate dynein-dependent retrograde motility of lysosomes (Keren-Kaplan et al., *Nat Commun.* 2022; Kumar et al., *Nat Commun.* 2022). We hypothesize that the RUFY family of proteins are a new class of activating dynein cargo adaptors that interact with Arl8b on distinct membranes (i.e., recycling endosomes (for RUFY1 and RUFY2) or lysosomes (for RUFY3 and RUFY4)) and mediate processive motility of the target compartments.

(2) What happens to Rab14 localization in DHC siRNA treated cells (extension of Figure S7A-C)? The prediction is that it should also be mis-localized since there is such an extensive overlap between Rab14 and Rabip4/4'.

Response: Yes, the reviewer's prediction is correct. Rab14, like RUFY1, was dramatically shifted towards the cell periphery after dynein heavy chain knockdown, indicating that normal Rab14 distribution in the perinuclear region is dependent on dynein. The endogenous Rab14 localization in control and dynein heavy chain siRNA treated cells has now been added as **Fig. S5G-I**.

(3) The robustness of the Arl8b/RUFY1 interaction is critical for this study. There are a couple of experiments that would strengthen the Arl8b/RUFY1 connection:

a. The authors should validate the Arl8b knockdown/knockout cells by immunofluorescence and/or Western blot. It is possible that some of the overlapping Rab14/RUFY1 puncta are still on residual Arl8b vesicles (partial knockdown/knockouts)

Response: We apologize for not including the Arl8b knockdown efficiency and Arl8b knockout (KO) status in the previous version and gratefully acknowledge the reviewer for this critical comment. We have now included these critical validations in our revised manuscript (**Fig. S2F**). We still observe RUFY1 punctae in Arl8b gene knockout HeLa cells (Arl8b^{-/-} KO), which were confirmed to be completely lacking Arl8b expression, (Marwaha R et al., *Journal of Cell Biology* 2017) ruling out the possibility of residual Arl8b expression in these cells. The identity of the RUFY1 endosomes that persist in Arl8b^{-/-} KO HeLa cells is an intriguing and important question that we are investigating in our current and future research.

b. The endogenous colocalization of Arl8b and RUFY1 would be ideal. I understand that good antibodies can be tough to come by, so at the very least, the authors should comment on this. Perhaps using a different epitope tag (Arl8b-FLAG or myc) to corroborate the main findings.

Response: The endogenous localization of both RUFY1 and Arl8b was included in the original manuscript (shown as **Fig. 1F**), and the colocalization was quantified in **Fig. 1J**. The figure legend and description state that this is endogenous staining.

(4) The literature suggests that Rabip4' plays multiple roles in the endolysosomal pathway.

a. How do the authors reconcile the findings demonstrating that Rabip4' regulates lysosome distribution (Ivan et al., 2012)? A comment in the discussion should be made here.

Response: This important suggestion has been included in the result section (line # 304-311). In RUFY1 knockdown cells, Ivan V et al., (*PLoSOne* 2012) found a peripheral distribution of lysosomes. Although we observed a modest dispersal of the perinuclear late endosomal/lysosomal pool (as shown in **Fig. 5B-C**; **Fig. S4B-C** and **S4F-G**), however, unlike Ivan et al., we did not observe a significant shift of lysosomes towards the cell periphery. HEK293T cells, which were used in previous research, exhibit a different RUFY1 isoform expression pattern than HeLa cells (see **Fig. 1B**), in addition to the morphological differences between the two cell types, which could account for this difference.

b. Since RUFY1 can interact with Rab4-marked early endosomes via its FYVE domain (Yamamoto et al., 2010), the model in Figure 9 should reflect these findings and put them into context in the discussion.

Response: We appreciate the reviewer's helpful suggestion and have updated our model to include Rab4 (**Fig. 10J**).

c. Based on this model, shouldn't there be a subset of Rab14/Arl8b puncta that do not overlap with Rab4? This should be tested.

Response: In experiments where we have expressed the epitope-tagged versions of RUFY1, Rab14, and Rab4, we find that there is a strong overlap between RUFY1, Rab14, and Rab4. Nevertheless, the reviewer is correct and we do observe several smaller punctae of Rab4 typically located in the cell periphery that do not colocalize with RUFY1 or Rab14. We are showing this data below, as it was not possible to include it in the main and supplementary figures because of a lack of space. In summary, we find that RUFY1 localizes to a subpopulation of Rab4/Rab14 endosomes that are located in perinuclear/juxtannuclear space (likely reflective of their association with dynein) and are also comparatively larger than Rab14-alone vesicles. Pearson's colocalization coefficient values reflect a modestly higher overlap of RUFY1 with Rab14 (0.6 ± 0.1) versus with Rab4 (0.45 ± 0.12), which is in agreement with the essential role of Rab14 in driving RUFY1 membrane recruitment (**Fig. S3G**).

Rab14/RUFY1 endosomes overlap with Rab4: HeLa cells expressing the indicated proteins were fixed and processed for immunofluorescences staining with anti-FLAG antibody. Inset 1 shows the Rab4 endosomes negative for Rab14 and RUFY1, and inset 2 shows the endosomes positive for Rab4, Rab14, and RUFY1. Bars: (main) 10 μm ; (insets) 2 μm .

Pearson's colocalization coefficients of RUFY1 with Rab14 and Rab4. The values plotted are the mean \pm S.D. from three independent experiments. Experiments are color-coded, and each dot represents the individual data points from each experiment. The total number of cells analyzed is indicated on the bottom of each data set (**** $p < 0.0001$; Student's t-test).

(5) The photobleaching experiment in Figure 4G-I is confusing. Shouldn't the most relevant populations of Rab4/GFP coated vesicles for this analysis be the motile ones (which makes it nearly impossible for photobleaching experiments)? Also, I am confused by the interpretation of the results. If Rab4 recovers quickly in the Arl8b knockdown, shouldn't this mean that Arl8b is dispensable for binding to membranes because it is MORE stably associated because presumably the new recovered pool of RUFY1 is coming from the cytosol, not other membranes?

Response: We agree with the reviewer that FRAP is not suitable to analyze the highly motile RUFY1 punctae. We have removed the FRAP assay from this revised version and instead used other approaches to determine the role of Arl8b in regulating RUFY1 localization. Using a knockdown approach to target Arl8b, Rab14, Rab4 and Rab5 to mitochondria (the three Rab proteins have been previously shown to interact with RUFY1), we show that Rab14 recruits RUFY1 to endosomal membranes, while Arl8b does not drive RUFY1 to membranes (**Fig. S3A-J**).

Surprisingly, even when Rab14 was overexpressed in Arl8b knockdown cells, a significant proportion of RUFY1 remained cytosolic, despite Rab14's endosomal distribution (see **Video S1 and Fig. S3M**). This was reflected in Arl8b-depleted cells by a reduced co-immunoprecipitation of RUFY1 and Rab14 (**Fig. 4G-H**). Following that, the possibility that Arl8b regulates RUFY1 interaction with Rab14 was investigated. To test this, the three proteins were recombinantly isolated, and RUFY1 was incubated with GTP-loaded Rab14 in the presence of increasing amounts of His-tagged Arl8b protein. The pulldown of full-length RUFY1 with Rab14 increased with increasing amounts of Arl8b, although no such increase was observed in the presence of His-tagged Rab7a, another G protein on late endosomes and lysosomes that does not colocalize with RUFY1 (**Fig. 4I**).

Interestingly, deletion of the RUN domain-containing region (N Δ RUN RUFY1 with an internal deletion of residues 147-270 a.a.) enhanced RUFY1 interaction with Rab14 in a yeast two-hybrid assay (**Fig. S3N**). These findings have led us to hypothesize that Arl8b binding to the RUN domain likely relieves RUFY1 autoinhibition and promotes RUFY1 and Rab14 interaction, resulting in RUFY1 stable membrane localization. Interestingly, Arl8b is shown to play a similar role in relieving SKIP autoinhibition for binding to the anterograde motor kinesin-1 (Kaplan KT and Bonifacino JS, *Current Biology* 2021). While we have removed the FRAP data from this version of the manuscript, the results from the FRAP assay did indicate a higher mobility of RUFY1 between the membrane and cytosol upon Arl8b knockdown that can be explained by less stable binding to Rab14.

Other points:

(1) In Figure S2B, it would be helpful to point out the two populations with arrows.

Response: We thank the reviewer for this useful suggestion and have now added arrowheads in the inset to show the two different populations of Arl8b.

September 22, 2022

RE: JCB Manuscript #202108001R-A

Dr. Mahak Sharma
Indian Institute of Science Education and Research Mohali
Biological Sciences
Sector-81, Knowledge City
SAS Nagar
Mohali, PUNJAB 140306
India

Dear Dr. Sharma,

Thank you for submitting your revised manuscript entitled "RUFY1 binds Arl8b and mediates endosome-to-TGN retrieval of CI-M6PR for cargo sorting to lysosomes". We would be happy to publish your paper in JCB pending final revisions necessary to meet our formatting guidelines (see details below). In your final revision, please be sure to address the reviewers' final comments and concerns.

A. MANUSCRIPT ORGANIZATION AND FORMATTING:

Submission of a paper that does not conform to JCB guidelines will delay the acceptance of your manuscript.

- 1) Text limits: Character count for Articles is < 40,000, not including spaces. Count includes abstract, introduction, results, discussion, and acknowledgments. Count does not include title page, figure legends, materials and methods, references, tables, or supplemental legends. * Please carefully edit your final text for length and clarity.
- 2) Figures limits: Articles may have up to 10 main text figures.
- 3) Figure formatting: Scale bars must be present on all microscopy images, including inset magnifications. Molecular weight or nucleic acid size markers must be included on all gel electrophoresis.
- 4) Statistical analysis: Error bars on graphic representations of numerical data must be clearly described in the figure legend. The number of independent data points (n) represented in a graph must be indicated in the legend. Statistical methods should be explained in full in the materials and methods. For figures presenting pooled data the statistical measure should be defined in the figure legends. Please also be sure to indicate the statistical tests used in each of your experiments (either in the figure legend itself or in a separate methods section) as well as the parameters of the test (for example, if you ran a t-test, please indicate if it was one- or two-sided, etc.). Also, if you used parametric tests, please indicate if the data distribution was tested for normality (and if so, how). If not, you must state something to the effect that "Data distribution was assumed to be normal but this was not formally tested."
- 5) Abstract and title: The abstract should be no longer than 160 words and should communicate the significance of the paper for a general audience. The title should be less than 100 characters including spaces. Make the title concise but accessible to a general readership.

We suggest the following edits to your title: " RUFY1 binds Arl8b and mediates endosome-to-TGN CI-M6PR retrieval for cargo sorting to lysosomes"

Please also remove the use of possessive, for example "RUFY1's coiled-coil region.." in the abstract, should be edited to "The coiled-coil region of RUFY.."

- 6) Materials and methods: Should be comprehensive and not simply reference a previous publication for details on how an experiment was performed. Please provide full descriptions in the text for readers who may not have access to referenced manuscripts.
- 7) Please be sure to provide the sequences for all of your primers/oligos and RNAi constructs in the materials and methods. You must also indicate in the methods the source, species, and catalog numbers (where appropriate) for all of your antibodies.

Please also indicate the acquisition and quantification methods for immunoblotting/western blots.

8) Microscope image acquisition: The following information must be provided about the acquisition and processing of images:

- a. Make and model of microscope
- b. Type, magnification, and numerical aperture of the objective lenses
- c. Temperature
- d. Imaging medium
- e. Fluorochromes
- f. Camera make and model
- g. Acquisition software
- h. Any software used for image processing subsequent to data acquisition. Please include details and types of operations involved (e.g., type of deconvolution, 3D reconstitutions, surface or volume rendering, gamma adjustments, etc.).

10) Supplemental materials: There are strict limits on the allowable amount of supplemental data. Articles may have up to 5 supplemental figures. Please also note that tables, like figures, should be provided as individual, editable files. A summary of all supplemental material should appear at the end of the Materials and methods section.

13) ORCID IDs: ORCID IDs are unique identifiers allowing researchers to create a record of their various scholarly contributions in a single place. At resubmission of your final files, please consider providing an ORCID ID for as many contributing authors as possible.

Please note that JCB now requires authors to submit Source Data used to generate figures containing gels and Western blots with all revised manuscripts. This Source Data consists of fully uncropped and unprocessed images for each gel/blot displayed in the main and supplemental figures. Since your paper includes cropped gel and/or blot images, please be sure to provide one Source Data file for each figure that contains gels and/or blots along with your revised manuscript files. File names for Source Data figures should be alphanumeric without any spaces or special characters (i.e., SourceDataF#, where F# refers to the associated main figure number or SourceDataFS# for those associated with Supplementary figures). The lanes of the gels/blots should be labeled as they are in the associated figure, the place where cropping was applied should be marked (with a box), and molecular weight/size standards should be labeled wherever possible.

B. FINAL FILES:

-- Cover images: If you have any striking images related to this story, we would be happy to consider them for inclusion on the journal cover. Submitted images may also be chosen for highlighting on the journal table of contents or JCB homepage carousel.

Images should be uploaded as TIFF or EPS files and must be at least 300 dpi resolution.

****It is JCB policy that if requested, original data images must be made available to the editors. Failure to provide original images upon request will result in unavoidable delays in publication. Please ensure that you have access to all original data images prior to final submission.****

****The license to publish form must be signed before your manuscript can be sent to production. A link to the electronic license to publish form will be sent to the corresponding author only. Please take a moment to check your funder requirements before choosing the appropriate license.****

Thank you for this interesting contribution, we look forward to publishing your paper in Journal of Cell Biology.

Sincerely,

Erika Holzbaur
Monitoring Editor

Andrea L. Marat
Senior Scientific Editor

Journal of Cell Biology

Reviewer #1 (Comments to the Authors (Required)):

The revised manuscript by Rawat et al has been substantially improved compared to the originally submitted manuscript. The authors have added a great deal of new data that not only address the previously raised concerns, but that also now provide mechanistic insight into the relative roles of RAB14 and ARL8B in RUFY1 recruitment to early endosomes, evidence for direct binding of the RUFY1 RUN domain to the active form of ARL8B, and solid mechanistic evidence of a requirement for RUFY1 in dynein-dependent cargo transport of the cation-independent mannose 6-phosphate receptor (CI-M6PR) from peripheral endosomes to the TGN region. The latter is responsible for lysosomal defects in cells depleted of RUFY1. Additional controls have also been added to strengthen the preliminary experiments in the original submission. All together, these additions have greatly strengthened the manuscript. The data now well support most of the conclusions, and the paper will make an important contribution to our understanding of ARL8B functions in compartments other than lysosomes and a better generic understanding of the function of RUFY family proteins.

Several minor concerns with the text and one of the figures need to be attended to, but these should be easily addressed.

1. The new data that Rabip4' and Rabip4 are pulled down by GST-Arl8b-GTP but not the -GDP form and that endogenous Rabip4' coimmunoprecipitates with Arl8b but not Rab8 are very nice. In the text on line 118, Rab8 should be identified as the "another cellular protein" used as a negative control.
2. In the text on line 127, the authors should reconsider their interpretation about relative affinities of RUFY1 and RUFY3 for GST-Arl8b-GTP based on the experiment shown. This was not done with purified proteins, and in the cell lysate one does not know (a) the relative abundance of RUFY1 vs. RUFY3 or (b) whether other proteins, e.g. endogenous Arl8b, might compete with binding for the GST fusion protein. The result simply shows that a lower fraction of endogenous RUFY1 can be pulled down by the GST fusion protein.
3. Supplemental Figure S3 seems to be central to the major findings in the paper. It would be worth considering rearranging some of the less important panels in the main figures to make room for this figure as a main figure.

4. Text line 306: Structures labeled by LAMP1, RAB7, or LysoTracker do not appear to be more scattered in RUFY1-depleted cells than controls. Unless the point can be made quantitatively, the sentence should be removed. On lines 310-311, the authors must specify "sorting endosomes" labeled by EEA1; recycling endosomes will not label for EEA1, but they are still "early endosomes".
5. Figure 5I and associated text: It is never mentioned in the text or the figure legend that the assay for EGFR degradation is loss of signal by immunofluorescence microscopy. This needs to be clearly stated in both places.
6. Text lines 329-331: The elevation of lysosomal enzyme expression during lysosomal dysfunction reflects mTORC1 inactivation and consequent activation of TFEB and TFE3; the authors could cite one of many reviews that address this point.
7. Text lines 374-378: In explaining the CD8alpha-CI-M6PR reporter, it needs to be clearly stated that the reporter is a chimera containing the luminal domain of CD8alpha and the cytoplasmic domain of CI-M6PR (and whichever component is the source of the transmembrane domain). The conclusion at the end of this paragraph is overstated; the data do not support that RUFY1 is essential for CI-M6PR exit, only that it "contributes to CI-M6PR exit" or "is required for optimal CI-M6PR exit".
8. In Figure 10F-H, it is difficult to see the potential overlap of CI-M6PR with Giantin because of (a) the use of blue color, which is very faint, to pseudocolor the Giantin-positive structures, and (b) the use of the green color, which is very bright, to identify the transfected cells. This figure would be much more useful if (a) the transfected cells were indicated only by the delimiting line (which is already shown in the current figure), (b) Giantin positive structures were pseudocolored green instead of blue, and (c) a magnified inset were shown in the perinuclear region of one of the rescued cells. As is, the figure does not seem to support the quantification shown, but this may be due to its presentation.
9. Use of terms such as "To prove that..." must be altered to "To test whether".
10. Overall, the text could be substantially shortened by tightening the language.

Reviewer #2 (Comments to the Authors (Required)):

Rawat et al investigate the role of Arl8b in endosomal to TGN cargo transport. They find that Arl8b interacts with a RUN-domain containing protein RUFY1, which is required for normal CI-M6PR retrieval back to the TGN. The authors also identify a tripartite interaction between Rab14/Arl8b/RUFY1; Arl8b is required to facilitate the interaction between Rab14 and RUFY1. The authors solidified these major findings very convincingly with new biochemical and cellular data. Finally, the authors suggested that the RUFY family is a new potential class of dynein adaptors important for retrograde cargo retrieval back to the TGN. It is still possible that other known activating adaptors are influencing RUFY dependent recruitment of dynein (on the same vesicles as RUFY). However, the authors show good evidence that RUFY1's role in cargo retrieval relies on an interaction with dynein, and they do beautiful work to characterize this interaction with dynein both in vitro and in cells. I believe that this study, along with the two others that were recently published, raise interesting questions about the selectivity of cargo transport/sorting via these different RUFY family members, potentially through dynein.

The authors adequately addressed my previous concerns and the paper has been drastically improved in all areas. I did have a minor point: Figure 9A panel should be re-worked to make it easier on the reader. The domains should be the same size across every construct, and the relative size of the protein should scale appropriately. For example, the 300 aa construct should be 3/5's the size of the 500 aa construct.

Response to Reviewers Comments:

We appreciate the reviewers' positive comments and insightful suggestions regarding our revised manuscript. The following is a point-by-point response (shown in blue) to the changes suggested by the reviewers (shown in black).

Reviewer #1 (Comments to the Authors (Required)):

The revised manuscript by Rawat et al has been substantially improved compared to the originally submitted manuscript. The authors have added a great deal of new data that not only address the previously raised concerns, but that also now provide mechanistic insight into the relative roles of RAB14 and ARL8B in RUFY1 recruitment to early endosomes, evidence for direct binding of the RUFY1 RUN domain to the active form of ARL8B, and solid mechanistic evidence of a requirement for RUFY1 in dynein-dependent cargo transport of the cation-independent mannose 6-phosphate receptor (CI-M6PR) from peripheral endosomes to the TGN region. The latter is responsible for lysosomal defects in cells depleted of RUFY1. Additional controls have also been added to strengthen the preliminary experiments in the original submission. All together, these additions have greatly strengthened the manuscript. The data now well support most of the conclusions, and the paper will make an important contribution to our understanding of ARL8B functions in compartments other than lysosomes and a better generic understanding of the function of RUFY family proteins.

Several minor concerns with the text and one of the figures need to be attended to, but these should be easily addressed.

1. The new data that Rabip4' and Rabip4 are pulled down by GST-Arl8b-GTP but not the -GDP form and that endogenous Rabip4' coimmunoprecipitates with Arl8b but not Rab8 are very nice. In the text on line 118, Rab8 should be identified as the "another cellular protein" used as a negative control.

Response: We appreciate the reviewer's recommendation. We have made the required modifications to the text (line # 104-105).

2. In the text on line 127, the authors should reconsider their interpretation about relative affinities of RUFY1 and RUFY3 for GST-Arl8b-GTP based on the experiment shown. This was not done with purified proteins, and in the cell lysate one does not know (a) the relative abundance of RUFY1 vs. RUFY3 or (b) whether other proteins, e.g. endogenous Arl8b, might compete with binding for the GST fusion protein. The result simply shows that a lower fraction of endogenous RUFY1 can be pulled down by the GST fusion protein.

Response: We appreciate the reviewer's insightful comments. The text has been modified accordingly (line # 112-115).

3. Supplemental Figure S3 seems to be central to the major findings in the paper. It would be worth considering rearranging some of the less important panels in the main figures to make room for this figure as a main figure.

Response: We appreciate the reviewer's recommendation. We attempted to move the Mito-ID experiment from Figure S3 to Figure 4, but this would disrupt the manuscript's flow and also affect Figure S2. Therefore, we have chosen not to move the Figure S3 panels.

4. Text line 306: Structures labeled by LAMP1, RAB7, or LysoTracker do not appear to be more scattered in RUFY1-depleted cells than controls. Unless the point can be made quantitatively, the sentence should be removed. On lines 310-311, the authors must specify "sorting endosomes" labeled by EEA1; recycling endosomes will not label for EEA1, but they are still "early endosomes".

Response: We concur with the reviewer's recommendation and have made the necessary modifications to the text (line # 288-294).

5. Figure 5I and associated text: It is never mentioned in the text or the figure legend that the assay for EGFR degradation is loss of signal by immunofluorescence microscopy. This needs to be clearly stated in both places.

Response: We appreciate the reviewer drawing our attention to this. We have made the required modifications to the text (line # 309-310) and figure legend (line # 727-730).

6. Text lines 329-331: The elevation of lysosomal enzyme expression during lysosomal dysfunction reflects mTORC1 inactivation and consequent activation of TFEB and TFE3; the authors could cite one of many reviews that address this point.

Response: We have included a suitable reference to TFEB in the text of the manuscript (line # 314).

7. Text lines 374-378: In explaining the CD8 α -CI-M6PR reporter, it needs to be clearly stated that the reporter is a chimera containing the luminal domain of CD8 α and the cytoplasmic domain of CI-M6PR (and whichever component is the source of the transmembrane domain). The conclusion at the end of this paragraph is overstated; the data do not support that RUFY1 is essential for CI-M6PR exit, only that it "contributes to CI-M6PR exit" or "is required for optimal CI-M6PR exit".

Response: We appreciate the reviewer's insightful comments on these textual statements. The text has been amended accordingly (line # 357-358 and 367-368).

8. In Figure 10F-H, it is difficult to see the potential overlap of CI-M6PR with Giantin because of (a) the use of blue color, which is very faint, to pseudocolor the Giantin-positive structures, and (b) the use of the green color, which is very bright, to identify the transfected cells. This figure would be much more useful if (a) the transfected cells were indicated only by the delimiting line (which is already shown in the current figure), (b) Giantin positive structures were pseudocolored green instead of blue, and (c) a magnified inset were shown in the perinuclear region of one of the rescued cells. As is, the figure does not seem to support the quantification shown, but this may be due to its presentation.

Response: We appreciate the reviewer's excellent suggestion regarding the representation of this figure, and we have modified Figure 10F-H accordingly. The Giantin (Golgi marker) labeling is now displayed as the green channel. The image does not depict the GFP channel; rather, the rescue construct transfected cells are outlined by a boundary.

In addition, we added insets to the images. Indeed, the revision suggested by the reviewer was essential and helpful for comprehending the experiment depicted in this figure.

9. Use of terms such as "To prove that..." must be altered to "To test whether".

Response: We have replaced the word “prove” with “to test whether” (line # 515).

10. Overall, the text could be substantially shortened by tightening the language.

Response: We appreciate the reviewer's recommendation and have reduced the article's character count by shortening the introduction and discussion sections.

Reviewer #2 (Comments to the Authors (Required)):

Rawat et al investigate the role of Arl8b in endosomal to TGN cargo transport. They find that Arl8b interacts with a RUN-domain containing protein RUFY1, which is required for normal CI-M6PR retrieval back to the TGN. The authors also identify a tripartite interaction between Rab14/Arl8b/RUFY1; Arl8b is required to facilitate the interaction between Rab14 and RUFY1. The authors solidified these major findings very convincingly with new biochemical and cellular data. Finally, the authors suggested that the RUFY family is a new potential class of dynein adaptors important for retrograde cargo retrieval back to the TGN. It is still possible that other known activating adaptors are influencing RUFY dependent recruitment of dynein (on the same vesicles as RUFY). However, the authors show good evidence that RUFY1's role in cargo retrieval relies on an interaction with dynein, and they do beautiful work to characterize this interaction with dynein both in vitro and in cells. I believe that this study, along with the two others that were recently published, raise interesting questions about the selectivity of cargo transport/sorting via these different RUFY family members, potentially through dynein.

The authors adequately addressed my previous concerns and the paper has been drastically improved in all areas. I did have a minor point: Figure 9A panel should be re-worked to make it easier on the reader. The domains should be the same size across every construct, and the relative size of the protein should scale appropriately. For example, the 300 aa construct should be 3/5's the size of the 500 aa construct.

Response: We appreciate the reviewer's excellent suggestion regarding the representation of this figure, and we have replaced the Figure 9A schematic to reflect the reviewer's recommendation. The domains are now of equal size, and the lengths of the various mutants reflect the size of the mutant proteins. This was a very useful modification to the panel, which improved the accuracy of the schematic.